# In vitro production of cat-restricted *Toxoplasma* pre-sexual stages

Ana Vera Antunes[1,5], Martina Shahinas[1,5], Christopher Swale[1,5], Dayana C. Farhat[1], Chandra Ramakrishnan[2], Christophe Bruley[3], Dominique Cannella[1], Marie G. Robert[1], Charlotte Corrao[1], Yohann Couté[3], Adrian B. Hehl[2], Alexandre Bougdour[1], Isabelle Coppens[4] & Mohamed-Ali Hakimi[1✉]

Sexual reproduction of *Toxoplasma gondii*, confined to the felid gut, remains largely uncharted owing to ethical concerns regarding the use of cats as model organisms. Chromatin modifiers dictate the developmental fate of the parasite during its multistage life cycle, but their targeting to stage-specific cistromes is poorly described[1,2]. Here we found that the transcription factors AP2XII-1 and AP2XI-2 operate during the tachyzoite stage, a hallmark of acute toxoplasmosis, to silence genes necessary for merozoites, a developmental stage critical for subsequent sexual commitment and transmission to the next host, including humans. Their conditional and simultaneous depletion leads to a marked change in the transcriptional program, promoting a full transition from tachyzoites to merozoites. These in vitro-cultured pre-gametes have unique protein markers and undergo typical asexual endopolygenic division cycles. In tachyzoites, AP2XII-1 and AP2XI-2 bind DNA as heterodimers at merozoite promoters and recruit MORC and HDAC3 (ref. 1), thereby limiting chromatin accessibility and transcription. Consequently, the commitment to merogony stems from a profound epigenetic rewiring orchestrated by AP2XII-1 and AP2XI-2. Successful production of merozoites in vitro paves the way for future studies on *Toxoplasma* sexual development without the need for cat infections and holds promise for the development of therapies to prevent parasite transmission.

*Toxoplasma*, the cause of the global zoonotic infection toxoplasmosis, presents a multifaceted life cycle with distinctive stages (Fig. 1a). Much is understood about its fast-growing tachyzoites and semi-dormant bradyzoites, but its sexual reproduction, confined to felid guts, is less explored. Each stage has a distinctive transcriptional signature and switching between them is regulated by intricate transcriptional cascades in which covalent and noncovalent epigenetic mechanisms act as driving forces[3,4]. The chromatin modifiers MORC and HDAC3, key players in gene silencing[1,2], act as critical checkpoints for sexual commitment, and when conditionally depleted or inhibited in tachyzoites, they trigger broad activation of chronic and sexual gene expression[1,2]. Previous studies overlooked their combined presence in nucleosomal telomeric repeats[1], suggesting a secondary role in maintaining genome stability through the formation of telomeric heterochromatin (Extended Data Fig. 1a). MORC detachment from chromosome ends disrupts subtelomeric gene silencing (Extended Data Fig. 1b), causing telomere dysfunction, mitotic bypass and aberrant polyploid zoite accumulation (Extended Data Fig. 1c), reminiscent of aneuploidy in human cancer[5]. In MORC-depleted parasites, disorganized telomeres may in turn disrupt gene-level transcriptional regulation, leading to misguided sexual development. An alternative way to explore the modus operandi of MORC involves its partners, the apetala proteins[1]

(AP2; Extended Data Fig. 2a), which are considered important regulators of life-cycle transitions in all apicomplexan species[3,6]. This study underscores the role of two AP2 repressors in coordinating the expression of stage-specific genetic programs, and thus controlling *Toxoplasma* merogony.

## AP2-mediated merozoite gene silencing

In the 1970s, *Toxoplasma*'s sexual cycle was partially studied in infected kittens through meticulous examination of the ultrastructure of the pre-gametes zoites and sexual dimorphic stages in the intestinal lining of *Felis catus*[7–11]. Merozoites—the initiators of the sexual cycle—have a unique transcriptional profile[12,13], but their study has been difficult owing to the lack of specific markers. The only recognized marker so far, GRA11b, is used to track the development of this stage in the gut of *Toxoplasma*-infected cats[14] (Fig. 1b). For a more in-depth exploration of merogony, we identified three potential merozoite-specific proteins with gene expression profiles mirroring those of GRA11b and produced matching antibodies. Notably, antibodies to TGME49_273980 showed robust reactivity with feline merozoites (Fig. 1b), but not with tachyzoites or bradyzoites, whether converted in vitro by overexpressing BFD1 (ref. 15) or present in tissue cysts in mouse brain (Extended Data Fig. 2b,c).

[1]Institute for Advanced Biosciences (IAB), Team Host-Pathogen Interactions and Immunity to Infection, INSERM U1209, CNRS UMR5309, University Grenoble Alpes, Grenoble, France. [2]Institute of Parasitology, University of Zurich, Zurich, Switzerland. [3]University Grenoble Alpes, CEA, INSERM, UA13 BGE, CNRS, CEA, Grenoble, France. [4]Department of Molecular Microbiology and Immunology, Johns Hopkins University Bloomberg School of Public Health and Malaria Research Institute, Baltimore, MD, USA. [5]These authors contributed equally: Ana Vera Antunes, Martina Shahinas, Christopher Swale. ✉e-mail: mohamed-ali.hakimi@inserm.fr

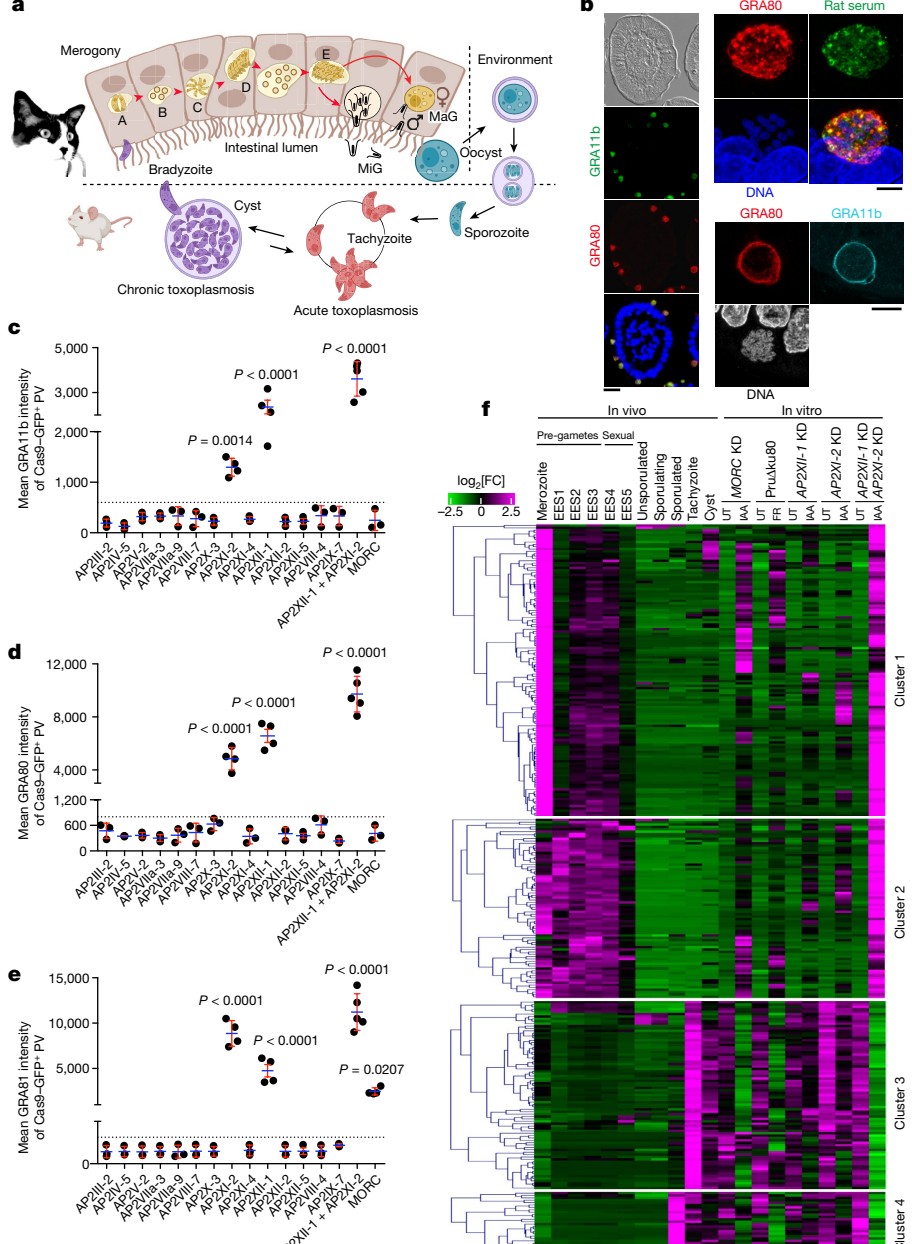

**Fig. 1 | Dual depletion of AP2XI-2 and AP2XII-1 induces merozoite-specific gene expression. a**, *T. gondii* has a multistage life cycle. The enteroepithelial cycle begins when a cat ingests tissue cysts, initiating asexual replication of merozoites (morphotypes A–E) leading to production of macro-gametes (MaG) and micro-gametes (MiG). These form oocysts that sporulate in an oxygen-rich environment. After ingestion, released sporozoites develop into tachyzoites causing acute infection. Later, owing to host immunity, they convert into bradyzoites forming tissue cysts. Created with BioRender.com. **b**, IFA images of infected cat intestine using GRA80 and GRA11b antibodies and rat serum; nuclei were counterstained with 4',6-diamidino-2-phenylindole (DAPI). Scale bars, 25 µm (left column of images), 2 µm (top right four images) and 5 µm (bottom right three images). **c–e**, Expression of the merozoite markers GRA11b (*TGME49_237800*; **c**), GRA80 (*TGME49_273980*; **d**) and GRA81 (*TGME49_243940*; **e**) was quantified in tachyzoites in which MORC, 1 of 14 MORC-associated AP2 proteins or AP2XII-1 and AP2XI-2 were genetically disrupted. Cas9–GFP measures genetic disruption efficacy (Extended Data Fig. 2f). Horizontal bars represent mean ± s.d. of vacuolar proteins intensity

from $n = 3$ (**c**,**d**) and $n = 4$ (**e**) independent experiments ($n = 50$ GFP⁺ vacuoles per dot). The *P* values were determined using one-way analysis of variance (ANOVA) and Tukey's test. PV, parasitophorous vacuole. **f**, Differential expression analysis was carried out with DESeq2 on raw rRNA-subtracted data, with Benjamini–Hochberg correction (*P* value threshold of 0.05). In the IAA (24 h)-induced double-KD parasites, 295 genes exhibited upregulation ($\log_2[FC] > 2$) and 195 genes exhibited downregulation ($\log_2[FC] < -1$), compared with untreated (UT) parasites. The heat map uses mean-centred data and *k*-means clustering (Pearson correlation) using iDEP.96, and shows $\log_2$-transformed data. Hierarchical clustering grouped genes and samples to elucidate expression patterns across different in vivo stages—merozoites, enteroepithelial stages (EESs) 1–5, tachyzoites, sporozoites and cysts—as documented in previous studies[12,13,16]. We also examined these patterns in the context of in vitro MORC KD and HDAC3 inhibition with FR235222 (FR; ref. 1). Transcript abundance is shown as $\log_2[FC]$ based on the log-transformed mean transcripts per million kilobases (TPM) values from three biological replicates. Magenta indicates upregulation, and green indicates downregulation.

This protein, together with GRA11b, is localized in the vacuolar space and on parasitophorous vacuole membranes, and has the typical characteristics of a dense granule protein; it is hereafter referred to as GRA80

(Fig. 1b). We were unable to produce antibodies compatible with immunofluorescence assay (IFA) for TGME49_243940, but the epitope-tagged protein shows typical features of a dense granule protein (GRA81) that

accumulates in the vacuolar space after MORC depletion[1] (Extended Data Fig. 2d). Another protein, GRA82 (TGME49_277230), co-occurs with GRA11b in in vivo schizonts specifically (Extended Data Figs. 2e and 3d).

Using these new markers, we investigated which MORC- and HDAC3-associated AP2 is responsible for repressing merozoite-specific gene expression in tachyzoites. In a CRISPR loss-of-function screen, inactivation of only *AP2XI-2* or *AP2XII-1* among the 14 MORC-associated AP2 proteins resulted in a significant increase in the expression level of GRA11b, GRA80 and GRA81–HA to varying degrees (Fig. 1c–e and Extended Data Fig. 2f). This suggests a common or overlapping role of these two transcription factors. To examine the genetic relationship between AP2XI-2 and AP2XII-1, we knocked out both genes simultaneously and observed a synergistic increase in the expression level of all three merozoite markers that exceeded threefold the levels observed in the individual knockouts (Fig. 1c–e). By contrast, *MORC* knockout led to only a low level of expression of GRA81–HA (Fig. 1e). AP2XI-2 and AP2XII-1 are essential for tachyzoite proliferation (Extended Data Fig. 2a), hindering the study of merogony with knockouts. To investigate this further, we used the minimal auxin-inducible degron system (mAID; Supplementary Table 1) and transiently knocked down each AP2 factor individually or simultaneously (Extended Data Fig. 2g). Single knockdowns (KDs) of AP2XI-2 and AP2XII-1 had limited effects on merozoite marker expression, as less than 25% of vacuoles were co-labelled with GRA11b and GRA80 (Extended Data Fig. 3b). However, simultaneous KD resulted in efficient merozoite differentiation with more than 98% of vacuoles showing both markers after 48 h (Extended Data Fig. 3a,b). By contrast, MORC-depleted vacuoles did not exhibit a significant level of coexpression of GRA11b and GRA80 (Extended Data Fig. 3b). Additionally, KD of three genes encoding other MORC-associated AP2 proteins (AP2VII-3a, AP2VIII-4 and AP2VIII-7) failed to induce merozoite marker expression (Extended Data Fig. 3c).

Notably, GRA82 showed delayed but significant coexpression with GRA11b in 98% of vacuoles depleted of both AP2XII-1 and AP2XI-2, 48 h after 3-indoleacetic acid (IAA) addition. This suggests a potential temporal regulation or association with a specific merozoite morphotype (Extended Data Fig. 3d,e). The in vitro pre-sexual parasite population that emerged following the acute depletion of AP2XI-2 and AP2XII-1 is homogeneous and does not express the typical markers of tachyzoites (for example, GRA2; Extended Data Fig. 3f,g) and bradyzoites (for example, BCLA (ref. 1), BAG1 and DBA; Extended Data Fig. 3h–j). In comparison, accumulation of Shield-protected BFD1 (ref. 15) induced bradyzoite differentiation in more than 95% of parasites (Extended Data Fig. 3h–j), without the expression of merozoite markers (Extended Data Figs. 2b and 3b). In contrast to the AP2 double-KD mutant, MORC-depleted parasites showed asynchronous development[1] with vacuoles expressing either bradyzoite (BCLA⁺) or merozoite (GRA81⁺) markers in a mutually exclusive manner (Extended Data Fig. 3k).

To gain a more complete understanding of in vitro merozoite differentiation beyond the limited perspective offered by GRA11b and GRA80 co-staining, we carried out RNA sequencing (RNA-seq) on all KD strains to investigate genome-wide transcriptional changes resulting from individual or simultaneous depletion of AP2XI-2 and AP2XII-1. Analysing the RNA-seq data using DESeq2, we identified 490 differentially expressed transcripts (fold change (FC) threshold of ≥8 and *P* value < 0.05), including 295 upregulated genes and 195 downregulated genes when both AP2XI-2 and AP2XII-1 were depleted (Fig. 1f and Supplementary Table 2).

Hierarchical clustering analysis revealed that the co-depletion of AP2XI-2 and AP2XII-1 resulted in gene expression profiles reminiscent of those observed in vivo in enteroepithelial stages (EESs)[12,13,16]. Their acute degradation triggered the induction of pre-gamete-specific genes at different developmental stages (clusters 1 and 2), while concurrently repressing a subset of tachyzoite-specific genes (clusters 3 and 4; Fig. 1f). These transcriptional changes mirror the gene expression profile observed in merozoites in the cat intestine[12,13]. Using principal component analysis, we observed consistent clustering of biological

replicates within each treatment, indicating excellent reproducibility. In addition, the samples showed substantial clustering based on genetic background with significant separation between single-KD and double-KD samples (PC1 = 58%), suggesting synergistic regulation of gene expression by AP2XI-2 and AP2XII-1 in *Toxoplasma* (Extended Data Fig. 4a). DESeq2 analysis revealed that depletion of both AP2 factors was found to be necessary to upregulate 65% (194/295) of the identified genes (FC ≥ 8; *P* value < 0.05; Extended Data Fig. 4b). By contrast, single KDs of AP2XI-2 and AP2XII-1 resulted in expression of a lower proportion of genes, 9% and 13.5%, respectively (Extended Data Fig. 4b). This transcriptional trend also extends to tachyzoite-specific genes, whose repression is quantitatively more pronounced when AP2 proteins are simultaneously depleted (Extended Data Fig. 4c).

Changes in mRNA levels in response to IAA treatment translated into changes in protein abundance. Principal component analysis showed consistent clustering of biological replicates within each condition, indicating high reproducibility (Extended Data Fig. 4d). Proteomic analysis revealed robust changes in 18% of the parasite proteins (*n* = 3,020 detected; log₂[FC] ≥ 1; *P* value ≤ 0.01), with a highly polarized response to the merozoite stage. IAA-treated parasites exhibited increased expression levels of 276 proteins associated with pre-gamete stages, whereas 285 tachyzoite proteins were suppressed (Extended Data Fig. 4e and Supplementary Table 3). Overall, the RNA and protein expression patterns of in vitro merozoites mirrored those observed in their enteroepithelial counterparts[12,13].

## AP2 depletion causes stage conversion

The process of invasion of *Toxoplasma* tachyzoites has been thoroughly examined, revealing the cryptic functions of organelle-resident proteins[17]. Micronemes secrete adhesin proteins (MICs) for attachment, facilitating motility and invasion. Rhoptries release neck (RON) and bulb (ROP) proteins, which interact with MICs to breach the host cell membrane and form the parasitophorous vacuole. Dense granules (GRA) release proteins involved in intravacuolar function, such as the tubulovesicular network, and act as effectors at the parasitophorous vacuole membrane and beyond, manipulating host signalling[18]. Most MIC, ROP and GRA proteins secreted by tachyzoites and bradyzoites are absent in merozoites isolated from cats[12,13,19,20]. In vitro, we also observed substantial changes in the expression of many known MIC, ROP and GRA proteins after the addition of IAA, providing evidence for a developmental switch (Fig. 2a–c and Supplementary Table 2). Specifically, MIC complexes that play important roles in tachyzoites (for example, MIC2 and AMA1) were repressed, whereas merozoite- and bradyzoite-specific MICs (for example, MIC17a, MIC17b, MIC17c and AMA2) were markedly induced. Notably, the reassortment of organelle-resident proteins seems to be highly specific, as MICs restricted to sporozoites (SporoAMA1) were not induced in response to IAA (Fig. 2a).

The combined depletion of AP2XI-2 and AP2XII-1 silenced 80% of the 143 rhoptry proteins described or predicted by hyperLOPIT to be tachyzoite specific (Fig. 2b). These include the components of the tachyzoite complex of RON2, RON4, RON5 and RON8 as well as ROP16, ROP18 and ROP5, which function as effectors to protect parasites from host cell-autonomous immune defences[18]. The number of ROP proteins reported to be exclusively specific to pre-gametes is rather limited (Fig. 2b). Among them, BRP1 stands out as the first protein described to be expressed in both merozoites and bradyzoites[21] (Fig. 2d). ROP26 is also expressed in both in vitro merozoites and BFD1-expressing bradyzoites (Fig. 2b,d and Extended Data Fig. 4f). These proteins, present in both stages, may facilitate the transition process from bradyzoites to merozoites.

When we examined GRA mRNA and protein levels in response to IAA treatment, we observed extensive and comparable shutdown of the tachyzoite program and activation of the merozoite program (Fig. 2c). Notably, the levels of core proteins of the tubulovesicular network,

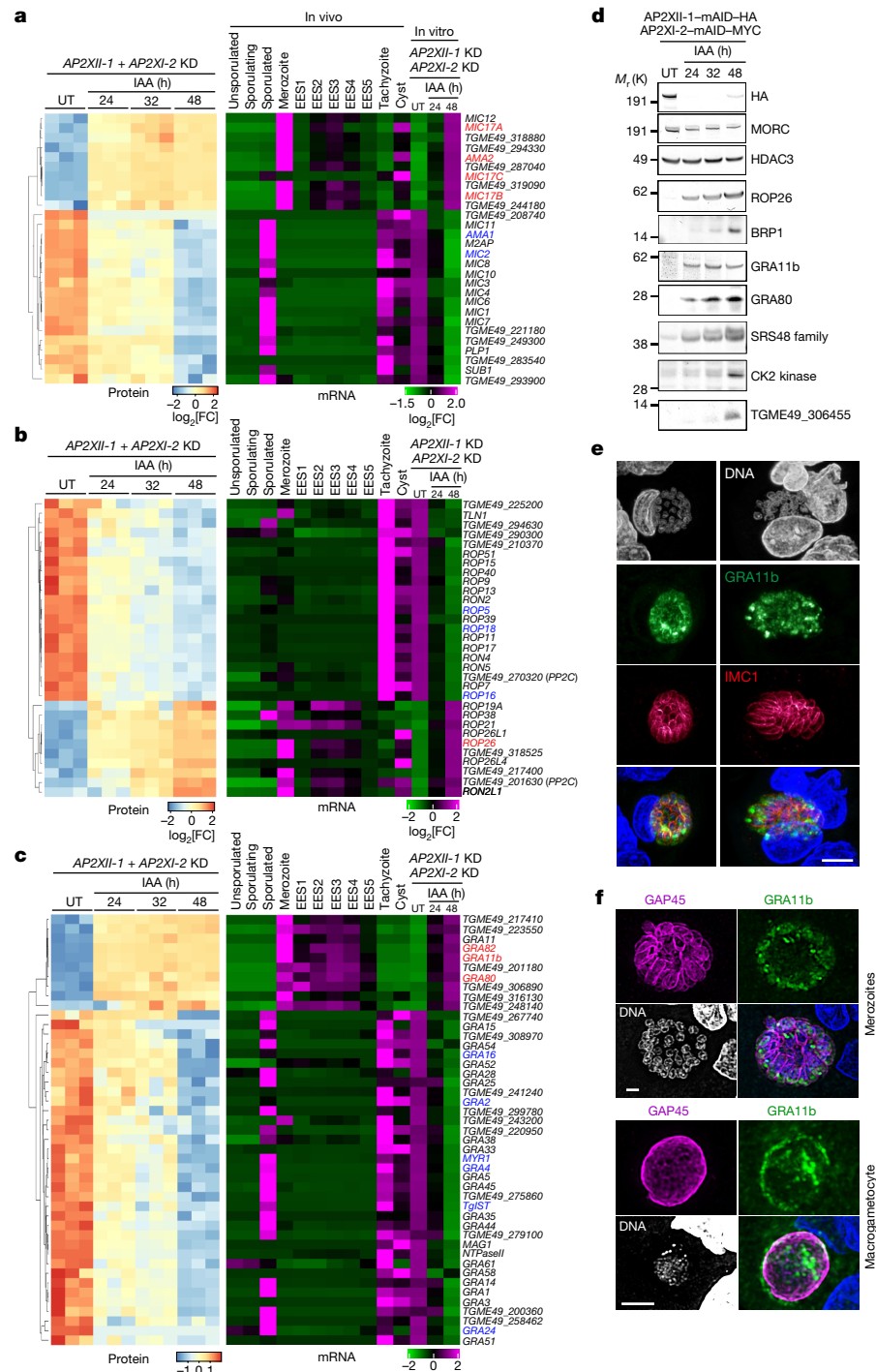

**Fig. 2 | Co-depletion of AP2XI-2 and AP2XII-1 causes rewiring of organellar proteomes specialized in invasion and host–parasite interaction. a–c**, Heat map showing hierarchical clustering analysis of selected microneme (**a**), rhoptry (**b**) and dense granule (**c**) mRNA transcripts and their corresponding proteins, which were significantly upregulated (log$_2$[FC] > 2; *P* value < 0.05) or downregulated (log$_2$[FC] < −1; *P* value < 0.05) following the simultaneous depletion of AP2XII-1 and AP2XI-2. The abundance of these transcripts is presented across different in vivo stages—merozoites, EES1–EES5 stages, tachyzoites, sporozoites and cysts—as documented in previous studies[12,13,16]. Pertinent examples of tachyzoite and merozoite stage genes are highlighted in blue and red, respectively. Analysis parameters are those of Fig. 1f. **d**, Time-course western blot analysis of protein expression levels after depletion of

AP2XII-1–mAID–HA and AP2XI-2–mAID–MYC. Samples were collected at the indicated time points after addition of IAA and probed with antibodies to HA, MORC, HDAC3, rhoptry proteins (ROP26 and BRP1), dense granule proteins (GRA11b and GRA80), the SRS48 family, a CK2 kinase (encoded by *TGME49_307640*) and the protein encoded by *TGME49_306455*. The experiment was repeated three times and a representative blot is shown. **e,f**, Maximum-intensity projection of a confocal microscopy *z*-stack from a meront in infected small intestine of a kitten. Antibodies to GRA11b (green) mark the dense granules and those to IMC1 (red) or GAP45 (magenta) mark the IMC of individual merozoites. Nuclei were counterstained with DAPI. Scale bars, 5 μm (**e**) and 2 μm (**f**).

including the GRA2 archetype, declined significantly in in vitro merozoites over time (Extended Data Fig. 3f,g). The MYR-dependent effectors GRA16, GRA24 or TgIST also showed a comparable decrease (Fig. 2c). By contrast, confirmed GRA markers of pre-gametes (for example, GRA11b (ref. 14), GRA80, GRA81 and GRA82) were induced in response to IAA (Fig. 2c,d and Extended Data Fig. 3a,b,d,e).

During the transition from tachyzoite to merozoite, surface proteins on the zoite also undergo significant restructuring, including the SAG-related surface (SRS) protein family[12,13]. Compared to tachyzoites, merozoites express a broader range of SRS proteins (Extended Data Fig. 4g)—for example, SRS48 (Fig. 2d) and SRS59—which may contribute to gamete development and fertilization[12,13]. IAA treatment induces the expression of 90% of the known 88 SRS proteins, effectively mimicking the phenotypic features of in vivo merozoites, whereas suppressing tachyzoite-specific SRS proteins (Extended Data Fig. 4g). In addition, 29 of the 33 family A members, representing the major membrane-associated merozoite proteins, are expressed in vitro during this transition to pre-gametes (Extended Data Fig. 4h).

In vitro-produced merozoites are deficient in essential proteins necessary for tachyzoite functions, including motility, attachment and invasion. As a result, these zoites exhibited reduced infectivity in human fibroblasts, as indicated by a notable decrease in the number of lytic plaques compared to that of untreated parasites (Extended Data Fig. 5a).

## AP2-depleted zoites undergo merogony

In 1972, Dubey and Frenkel characterized pre-sexual stages at the cellular level and identified five morphological stages (A–E) that form sequentially during colonization of the cat intestinal epithelium before gamete formation[22] (Fig. 1a). These morphotypes can be differentiated on the basis of their distinct subcellular structures and nuclear content[23]. Using transmission electron microscopy and immunofluorescence, we tracked the development of these stages during in vitro merogony, including the dynamic behaviour of the inner membrane complex (IMC), a unique organelle essential for daughter cell formation during replication. During development in the cat intestine, the functions of various IMC proteins diverge from their assigned roles in tachyzoites[23]. For example, at the later stages of schizogony, IMC1 and IMC3 staining revealed daughter IMC, whereas IMC7 staining was restricted to the periphery of the mother cell[23]. In agreement with these findings, our results show that IMC1 but also GAP45 are valuable markers for tracking merozoite division during their development in the intestinal mucosa of cats (Fig. 2e,f).

As a first step, 24 h post-IAA addition, the nucleus of the mother cell undergoes several fission events while maintaining the nuclear envelope, leading to individualized nuclei (in even numbers, ranging from 4 up to 10; Fig. 3a). Concomitantly, single organelles such as the apicoplast and the Golgi apparatus expand and multiply to match the number of nuclei. Transversal cross-sections of the apicoplast (limited by four membranes) reveal its elongation and constriction, suggestive of replication by scission (Fig. 3b), which was also visualized by IFA with the ATrx1 antibody[24] (Extended Data Fig. 5b) and is in line with maternal inheritance of the apicoplast seen in meronts in the cat intestine[25]. Multiple Golgi complexes are formed at different sites of the nucleus, sometimes in opposing orientations, suggestive of de novo formation (Fig. 3c). The manner in which other organelles, such as secretory organelles, multiply is not yet known. At this stage, the multinucleated mother cell contains several sets of organelles randomly distributed throughout the cytoplasm. Despite the increase in size of the mother cell, the subpellicular IMC is still prominently present beneath the plasma membrane. The parasites exhibiting a characteristic ovoid shape with four and eight nuclei are morphologically related to the cryptic and early meronts, namely B and C morphotypes[22] (Fig. 3d and Extended Data Fig. 5c).

As a second step, new flattened vesicles of the IMC emerge in the mother cytoplasm, and progressively elongate allowing the sub-compartmentalization of organelles destined for each daughter cell (Fig. 3e). This process of internal budding of more than two daughter cells, referred to here as endopolygeny[26–30], differs from the tachyzoite division by endodyogeny in which the two daughter cells are generated symmetrically and in a synchronous manner in the mother cell (Fig. 3f). Alongside the expansion of daughter buds, the mother IMC and conoid undergo partial disassembly. Notably, rhoptries inside daughter cells are different in shape and electron density from mother rhoptries dispersed in the cytoplasm, suggesting de novo biogenesis of rhoptries (Fig. 3g), which can also be traced with ROP26 (Fig. 3h). This finding aligns with the observation that the bulbous end of the rhoptry in in vivo meronts remains spherical, in contrast to that in tachyzoites and bradyzoites[31].

In these multinucleated structures, daughter IMC could be identified with IMC1 staining, whereas GAP45 staining was restricted to the periphery of the mother cell (Fig. 3i and Extended Data Fig. 5d). Daughter cells become polarized with the formation of a conoid and apical distribution of micronemes, rhoptries, the apicoplast, and the Golgi apparatus (Fig. 4a and Extended Data Fig. 5e). At this stage, it is evident that the maternal conoid coexists with the newly formed conoids of the progeny, as shown by the labelling of apically methylated proteins (Extended Data Fig. 5f). After final assembly, the daughter cells emerge separately, wrapped by the plasma membrane of the mother cell (cortical or peripheral budding; Fig. 4b,c), forming fan-like structures as previously described in infected cat cells[26]. Compared to tachyzoites, these newly formed parasites are thinner and do not form a rosette-like structure within the parasitophorous vacuole but instead are aligned, with their apex facing the parasitophorous vacuole membrane (Extended Data Fig. 5g,h); these features are reminiscent of those of type-D-like merozoites arising in cat intestinal cells from meront entities at the onset of infection[28–30]. Notably, the parasitophorous vacuole membrane of merozoites also forms physical interactions with host ER and mitochondria, potentially for nutrient acquisition[32] (Extended Data Fig. 5h).

## In vitro merozoites mimic cat pre-gametes

Extending the IAA treatment for another 16 h reveals the presence of very large meronts containing numerous daughter cells in formation (Fig. 4d,e). These schizonts are detected in the same parasitophorous vacuole together with fully formed in vitro merozoites (Fig. 4e), the latter being very similar to their counterpart in the cat intestine (Fig. 2e,f). Mature polyploid meronts can be visualized by IMC7 staining on their surface, whereas fully formed merozoites were completely negative for IMC7 (Fig. 3d,h and Extended Data Fig. 5c), a phenotype that has also been observed in pre-gametes developing in the cat gut[23]. Notably, new pre-merozoite or merozoite-specific markers such as ROP26 and GRA80, respectively, distinguish the two morphotypic populations. ROP26 marks zoites undergoing schizogonic replication, whereas GRA11b and GRA80 are expressed exclusively in mature in vitro merozoites (Fig. 3h and Extended Data Fig. 6a). As merozoites undergo several cycles of endopolygeny, they acquire new distinct morphological features compared to first-generation merozoites (24 h post-IAA), probably type E (ref. 26). Some in vitro merozoites are sausage shaped, with a diameter of 1.5–1.8 µm, packed in the parasitophorous vacuole without any spatial organization (Extended Data Fig. 6b,c). These forms contain similar organelles found in tachyzoites, but they exhibit an extruded conoid (Extended Data Fig. 6d). Other parasitophorous vacuoles contain peripherally arranged parasites, leaving a large empty space (Extended Data Fig. 6e,f), reminiscent of schizont parasitophorous vacuoles formed in cat intestinal cells[26]. Notably, these parasites at the parasitophorous vacuole edge adopt two configurations: a very large cell body (trapezoid) with a diameter of up

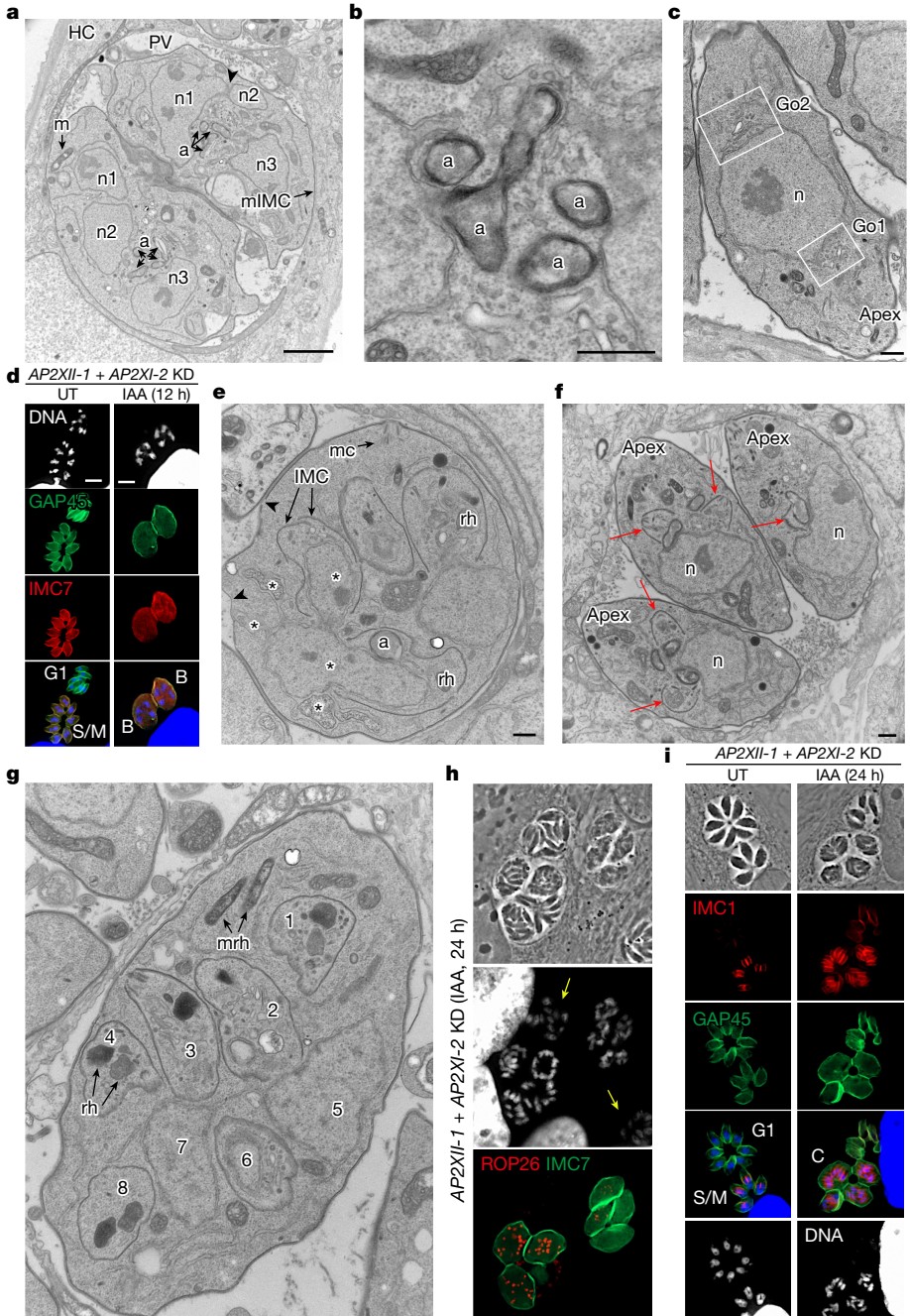

**Fig. 3 | Zoites depleted of AP2XI-2 and AP2XII-1 undergo endopolygeny with karyokinesis. a–c,e–g**, Electron micrographs of human foreskin fibroblast (HFF) cells infected with the RH (*AP2XII-1* and *AP2XI-2* KD) strain of *T. gondii*, untreated (**f**) or treated for 24 h (**a–c,e,g**) with IAA. **a**, Emphasis on karyokinesis with fission. n1 to n3, nuclear profiles; a, apicoplast; m, mitochondrion; mIMC, mother IMC; HC, host cell. The arrowhead highlights an area of nuclear fission. **b**, Emphasis on apicoplast multiplication by growth and scission. **c**, Emphasis on Golgi multiplication from either side of the nucleus (n). Go1 and Go2, two Golgi apparatus. **d**, IFA of tachyzoites (untreated) and zoites depleted of AP2XII-1 and AP2XI-2 (12 h post-IAA). GAP45 staining marks the mother cell and its progeny. IMC7 staining specifically marks the diploid (left) and polyploid (right) mother cell. Cells were stained with Hoechst DNA-specific dye. Type B meronts are marked in yellow. **e**, Emphasis on appearance and role

of the IMC segregating daughter buds in the mother cytoplasm. mc, mother conoid; rh, rhoptry. The arrowheads show areas devoid of the IMC and the asterisks highlight areas of nuclear fission. **f**, Emphasis on contrasting endodyogeny in tachyzoites (untreated condition). Two daughter buds formed apically and symmetrically (arrows). **g**, Emphasis on endopolygeny showing up to eight daughter buds and ultrastructure of rhoptries. mrh, mother rhoptry. **h,i**, Tachyzoites (untreated) and merozoites depleted of AP2XII-1 and AP2XI-2 (24 h post-IAA) were fixed and stained for ROP26 (*TGME49_209985*, in red) and IMC7 (green; **h**) or IMC1 (red) and GAP45 (green) along with Hoechst DNA-specific dye (**i**). Yellow arrows indicate IMC7⁻ mature merozoites, and type C meronts are shown. Scale bars, 2 μm (**a**), 500 nm (**b,c,e–g**), 5 μm (**d,i**) and 10 μm (**h**). G1, cell growth phase before DNA synthesis; S/M, DNA synthesis (S phase) and mitosis (M phase).

to 5 μm or a very thin and elongated shape (tubular) with a diameter of 200–250 nm (Extended Data Fig. 6g,h). The latter do not contain nuclei but mitochondrion profiles and ribosomes are observed. Their origin and formation remain to be determined but their abundance in parasitophorous vacuoles probably suggests a physiological relevance in the *Toxoplasma* life cycle.

Morphotypes A–E are difficult to study in vivo because they vary in size and shape and develop asynchronously in different regions of the digestive

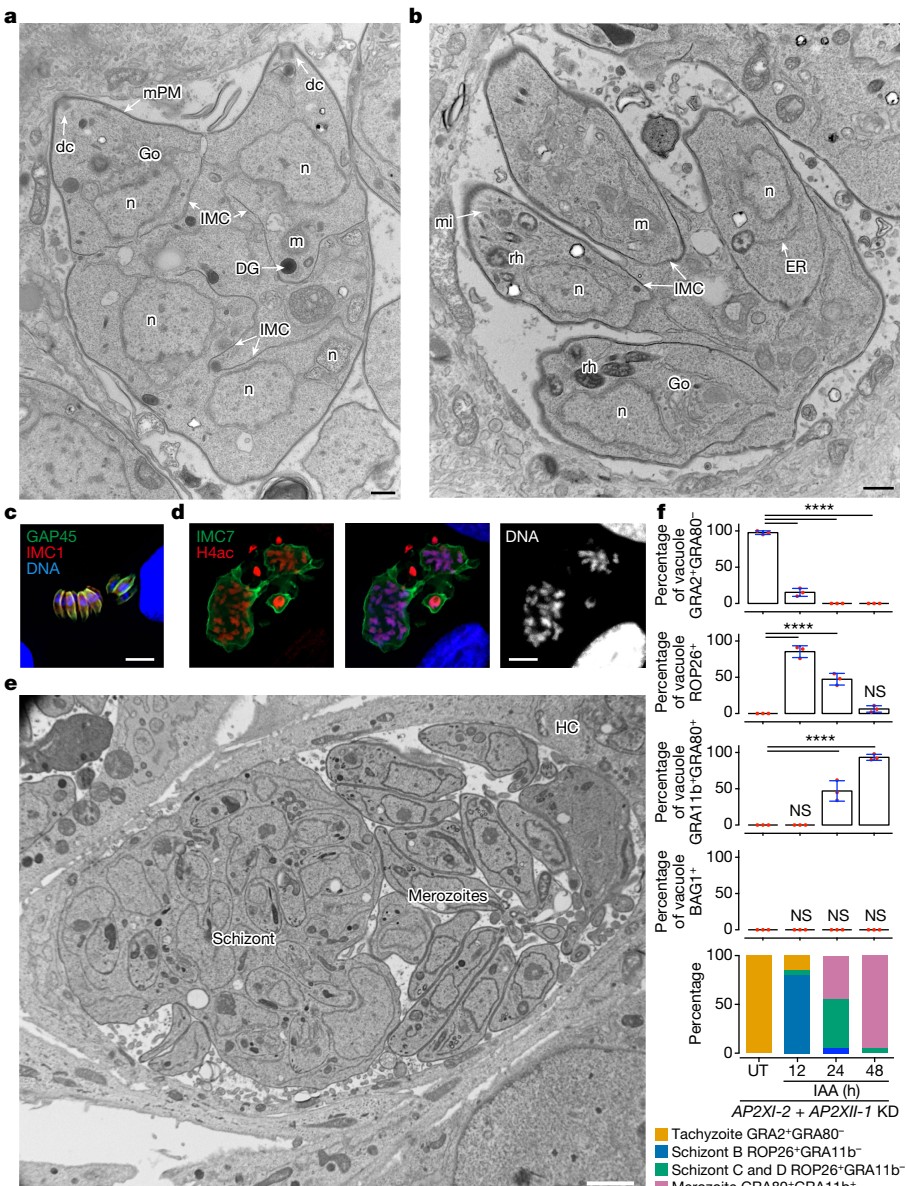

**Fig. 4 | In vitro-induced merogony typified by the emergence of multiple zoite stages. a,b,e**, Electron micrographs of RH (*AP2XII-1* and *AP2XI-2* KD)-infected HFFs treated for 24 h (**a,b**) or 48 h (**e**) with IAA. **a**, Emphasis on protruding daughters sharing the mother plasma membrane (mPM). DG, dense granule; dc, daughter conoid. **b**, Emphasis on daughter cell emergence. mi, microneme; ER, endoplasmic reticulum. **c**, Representative image of neatly aligned elongated merozoites and forming fan-like structures as they hatch from the mother cell. Mature merozoites are co-stained with GAP45 (green), IMC1 (red) and Hoechst DNA-specific dye (blue). **d**, Image of a giant schizont delineated by IMC7 (green) showing polyploidy (*n* = 16). The nuclear structure is co-stained with Hoechst DNA-specific dye and hyperacetylated histone H4

(H4ac; red). **e**, Emphasis on a large parasitophorous vacuole containing a mega meront with many daughter buds residing with merozoites in the same parasitophorous vacuole. **f**, Time-course analysis of marker expression following AP2XII-1 and AP2XI-2 co-depletion. Data represent mean ± s.d. of vacuole staining for GRA2+GRA80−, ROP26+, GRA11b+GRA80+ or BAG1+ from three experiments (*n* = 50 vacuoles per dot). Statistical evaluation was conducted using one-way ANOVA, followed by Tukey's multiple comparison test. Graph at bottom: longitudinal tracking of tachyzoites and developmental morphotypes within vacuoles post-IAA treatment, integrating stage-specific aforementioned markers and parasite nuclei count (Hoechst and histone staining). Scale bars, 500 nm (**a,b**), 5 μm (**c**), 10 μm (**d**) and 2 μm (**e**).

tract[19,20,26,31,33]. Here we were able to follow the initial steps of in vitro merogony using several stage-specific markers. Asynchronous nuclear division cycles were detected by DNA, histone or centrosome staining (Extended Data Fig. 7a–d). At 12-h post-IAA treatment, a significant proportion of the tachyzoite population transitioned into morphotype B (3–4 nuclei and ROP26+), reaching up to 75% (Fig. 4f). Within 24 h, morphotypes C and D (8–32 nuclei and ROP26+) and mononuclear merozoites (GRA11b+GRA80+) coexisted in culture (Fig. 4f). After 48 h, nearly 98% of the parasite population expressed merozoite markers (GRA11b+ and GRA80+), whereas typical tachyzoite (GRA2) or bradyzoite (BAG1) markers were absent

(Fig. 4f), aligning with our extensive transcriptome and proteome analyses.

## AP2 proteins bind to MORC and HDAC3

AP2XI-2 and AP2XII-1 probably synergize to suppress gene expression in tachyzoites, but their modus operandi is still enigmatic. Both proteins were originally found in a MORC pulldown along with HDAC3 in tachyzoites[1]. We confirmed their strong and specific association with MORC and HDAC3 by reverse immunoprecipitation combined with

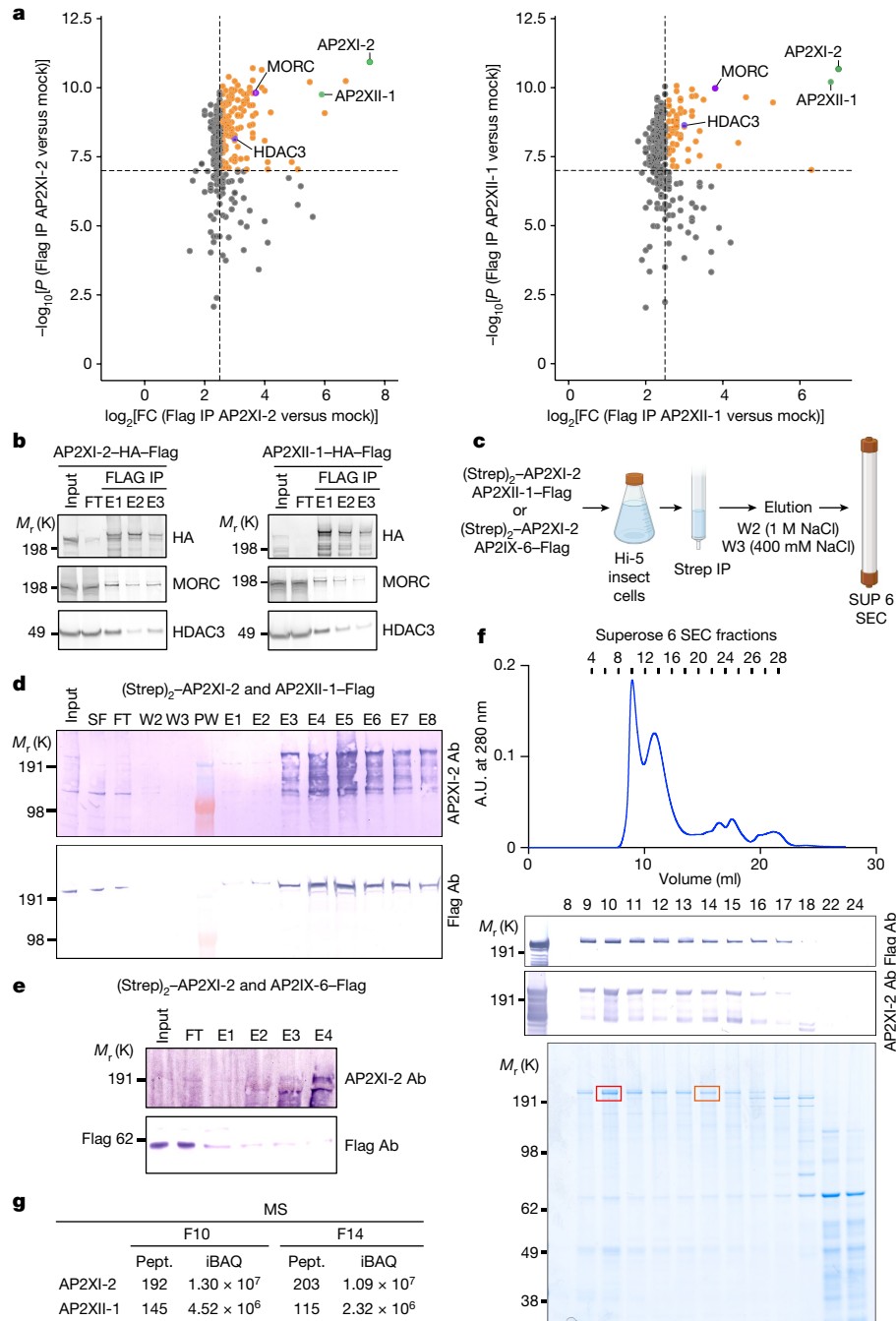

**Fig. 5 | AP2XII-1 and AP2XI-2 heterodimerize and interact with HDAC3 and MORC to form a repressive core complex. a**, Flag immunoprecipitation (IP) eluates in HFF cells infected with wild-type parasite (mock) or parasites stably expressing HA–Flag-tagged AP2XII-1 or AP2XI-2 protein were examined through label-free quantitative proteomics using MS (three replicates per condition) and are represented as a volcano plot. Each dot represents a protein. Black dashed lines show −log₁₀[*P* value] < 7 and log₂[FC] = 2.5 cutoffs; proteins above these thresholds are coloured in orange, with specific proteins in blue, purple and green as indicated. Raw data and a detailed statistical analysis are shown in Supplementary Table 4. **b**, Flag affinity eluates were analysed by western blot to detect MORC and HDAC3. This was repeated three times, and a representative blot is shown. **c**, Purification scheme for AP2XI-2–Strep and AP2XII-1–Flag or (Strep)₂–AP2XI-2 and AP2IX-6–Flag coexpressed in *Trichoplusia ni* (Hi-5) insect cells. Created with BioRender.com. **d**, Co-purification of AP2XII-1–Flag through the pulldown of (Strep)₂–AP2XI-2 by Strep-Tactin purification. **e**, AP2IX-6–Flag is not co-purified by (Strep)₂–

AP2XI-2. In **d**,**e**, (Strep)₂–AP2XI-2 was detected using an in-house anti-AP2XII-2 antibody (Ab) and AP2XII-1–Flag or AP2IX-6–Flag was detected using an anti-Flag antibody. Input, soluble fraction (SF), flow-through (FT), high-salt wash (W2) and final wash (W3) deposits were deposited using 8 µl whereas eluted fractions (E1 to E8), separated by the protein molecular weight marker (PW) were deposited using 15 µl. Experiments were carried out at least twice for **d**,**e**. **f**, Strep-Tactin XT-purified (Strep)₂–AP2XI-2 and AP₂XII-1–Flag proteins were fractionated on a Superose 6 Increase gel filtration column. Input (concentrated Strep-Tactin elution) and gel filtration fractions were separated by SDS–polyacrylamide gel electrophoresis and analysed by both western blots using anti-Flag and in-house anti-AP2XII-2 antibodies and colloidal blue staining. Fraction numbers are indicated at the top of the gel. A.U. at 280 nm, absorbance units at 280 nm. **g**, MS-based proteomic analyses of fraction (F) 10 and fraction 14. Number of identified peptides (Pept.) and intensity-based absolute quantification (iBAQ) values are indicated (Supplementary Table 5).

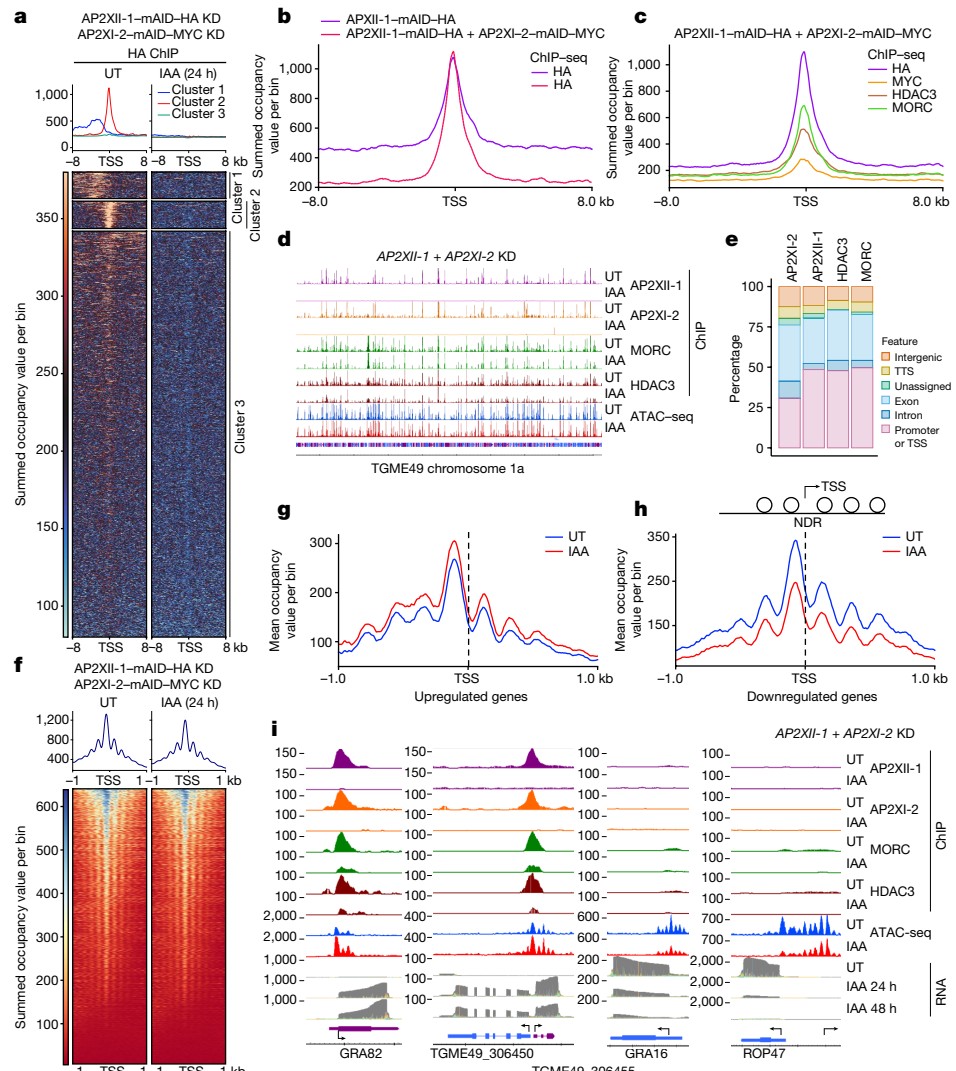

**Fig. 6 | Chromatin recruitment of HDAC3 and MORC by AP2XII-1 and AP2XI-2. a**, Heat map and profile analysis of the ChIP peak intensity for AP2XII-1 in parasites, either untreated or following co-depletion of AP2XII-1 and AP2XI-2. *k*-means clustering applied to the untreated sample shows a centred enrichment of AP2XII-1 at the TSS in genes from cluster 2. **b**, Profile comparison of AP2XII-1 ChIP peaks in untreated single- or double-KD strains, centred at the TSS (±8 kb) within cluster 2 genomic loci. **c**, Superposed profile plots indicate AP2XII-1, AP2XI-2, HDAC3 and MORC co-enrichment around the TSS of cluster 2 genes in the untreated state. The profile plots in **a**–**c** were generated by Deeptools using summed occupancies over a bin size of 10. **d**, Integrated Genome Browser view illustrates AP2XII-1, AP2XI-2, MORC and HDAC3 enrichment and ATAC–seq chromatin accessibility before and after simultaneous *AP2XII-1* and *AP2XI-2* KD. The *y* axis shows the read density. **e**, HOMER analysis reveals global distribution of significant peaks within genomic features for AP2XI-2, AP2XII-1, HDAC3 and MORC ChIP experiments. TTS, transcription termination site. **f**, Profile and heat maps of summed occupancy using bin sizes of 10 show Tn5 transposase accessibility (ATAC–seq) for parasite genes centred at the TSS (±1 kb) in untreated and IAA-treated samples. High read intensity is in blue, with average signal profiles plotted above. **g**,**h**, The Tn5 transposase accessibility plot predicts nucleosomal occupancy. Mean occupancy profiles with a bin size of 10 of untreated and IAA-treated ATAC–seq signals across downregulated (*n* = 226) and upregulated genes (*n* = 281) show a nucleosome-depleted region (NDR) at the TSS and enriched, phased mono-nucleosome fragments at surrounding regions. **i**, Integrated Genome Browser screenshots of representative merozoite and tachyzoite genes highlight ChIP–seq signal occupancy for HA, MYC, MORC and HDAC3, in untreated and post-depletion conditions. RNA-seq data at different induction times and Tn5 transposase accessibility profiles for both conditions are also shown.

mass spectrometry (MS)-based quantitative proteomic and western blot analyses using knock-in parasite lines expressing a Flag-tagged version of AP2XI-2 or AP2XII-1 (Fig. 5a,b). Each AP2 protein shows significant enrichment in the eluate of its corresponding counterpart (FC > 90; *P* value < 1.5 × 10⁻¹⁰), indicating their association within the same functional complex along with MORC and HDAC3 (Fig. 5a). Notably, this interaction is specific and exclusive to these two AP2 proteins, as no such association was observed with other AP2 proteins, or with other chromatin modifiers (Supplementary Table 4). HDAC3 and MORC exhibited comparatively lower levels of enrichment (FC of 8 and 14, respectively, Fig. 5a), suggesting that AP2XI-2 or AP2XII-1 proteins independently form heterodimers in cellular contexts. To further

test this hypothesis, we used baculoviruses to transiently coexpress epitope-tagged AP2XII-1–Flag and (Strep)₂–AP2XI-2 in insect cells, with AP2IX-6–Flag serving as an internal control (Fig. 5c). AP2XII-1 was purified by Strep-Tactin affinity chromatography, and the partnership between AP2XII-1 and AP2XI-2 was confirmed through western blot analysis (Fig. 5d), whereas no co-enrichment was detected with AP2IX-6 (Fig. 5e). Consistent with AP2XI-2 and AP2XII-1 being part of a heterodimer, our findings show these two proteins coelute in the same gel filtration fractions, in a MORC- and HDAC3-independent manner, as confirmed by MS-based proteomics (Fig. 5f,g and Supplementary Table 5). Many transcription factors, including apicomplexan AP2, were reported to form homodimers and heterodimers with different

partners that modulate DNA-binding specificity and affinity[34,35]. In this context, AP2XI-2 and AP2XII-1 probably bind cooperatively as a heterodimer to DNA to selectively and synergistically repress merozoite gene expression, and only their simultaneous depletion leads to achievement of the developmental program critical for merozoite formation.

## AP2XI-2 and AP2XII-1 limit chromatin access

To further explore gene repression by AP2XI-2 and AP2XII-1, we used chromatin immunoprecipitation with sequencing (ChIP–seq; GSE222819) to analyse their genome-wide distribution during conditional single or double KD. Simultaneously, we examined recruitment of their partners MORC and HDAC3 to chromatin under KD conditions. In the AP2XII-1-specific cistrome, we identified genes from cluster 2 that are exclusively expressed in pre-gametes and have a discrete and highly enriched peak centred on their transcription start site (TSS; Fig. 6a,b). Examining the co-occupancy in cluster 2, we observed a strong overlap between the binding sites of AP2XI-2 or AP2XII-1 cistromes in the untreated condition (Fig. 6c,d and Extended Data Fig. 8a) with approximately 30–50% of the peaks located at the promoter or TSS (Fig. 6e). Consistently, AP2XII-I and AP2XI-2 showed similar genome-wide occupancy when immunoprecipitated from single or double KD strains (Fig. 6b and Extended Data Fig. 8). HDAC3 and MORC are both enriched at AP2XII-I and AP2XI-2 peaks (Fig. 6c,d and Extended Data Fig. 8c,d). Addition of IAA triggers the acute release of AP2XI-2 and AP2XII-1 from chromatin and a concomitant reduction in HDAC3 and MORC occupancy at the TSS at cluster 2 genes, which is more pronounced in the context of double KD (Fig. 6d and Extended Data Fig. 8a,e).

AP2XI-2 and AP2XII-1 are expected to alter chromatin compaction and accessibility, a function attributed to their partners MORC and HDAC3. To investigate this assumption, we carried out assay for transposase-accessible chromatin with high-throughput sequencing (ATAC–seq; GSE222832), a robust and streamlined method for profiling chromatin accessibility[36]. At the genome level, there is a slight decrease in average accessibility between untreated and treated conditions (Fig. 6f) with most of the peaks located at the TSS (Extended Data Fig. 9a,b). However, when we plotted ATAC–seq data for the subset of downregulated and upregulated genes (as defined in Fig. 1f), the changes in occupancy were more pronounced and consistent with expected increase or decrease in accessibility of induced or repressed clusters, respectively (Fig. 6g,h). At the gene level, dynamic release of AP2XI-2 and AP2XII-1 from DNA induced by IAA resulted in a substantial decrease in MORC and HDAC3 enrichment, which enhanced local chromatin hyperaccessibility and led to a concomitant increase in target gene mRNA abundance, a pattern largely shared by representative merozoite genes (Fig. 6i and Extended Data Fig. 9c–e).

We next examined how AP2XII-1 degradation influences AP2XI-2 binding genome-wide. Initially, depletion of AP2XII-1 did not affect the nuclear localization or signal intensity of AP2XI-2 (Extended Data Fig. 10a). Notably, AP2XI-2 did not dissociate from chromatin post AP2XII-1 degradation (Extended Data Fig. 10b), indicating its ability to form independent homodimers and repress merozoite-specific genes (Extended Data Fig. 10c,d). At the transcriptional level, the persistence of homodimers on chromatin explains the sustained repression of merozoite gene expression when a single AP2 is depleted, with peak expression reached only following simultaneous KD (Fig. 1f and Extended Data Figs. 4c and 10d).

## Secondary regulators guide merogony

Some genes escaped direct regulation by AP2XI-2 and AP2XII-1 because they expressed increased levels of RNA and had hyperaccessible chromatin signatures after the addition of IAA but lacked the characteristic recruitment of MORC and HDAC3 to their promoters in the untreated state (for example, *PNP*; Extended Data Fig. 10e). This indirect regulation is also typical of tachyzoite genes that are repressed when AP2XI-2 and AP2XII-1 are depleted. They show a strong decrease in ATAC–seq signals after addition of IAA, but no apparent binding of the repressive MORC complex to their TSS (Fig. 6i and Extended Data Figs. 10f,g and 11a).

This suggests that AP2XI-2 and AP2XII-1 operate on gene expression through an indirect mechanism that is not reliant on their DNA-binding activities or their functional partners MORC and HDAC3. This transcriptional outcome may stem from secondary transcription factors that govern the establishment of specific predetermined transcriptional programs for distinct stages[1,4] (Extended Data Fig. 12). Supporting this hypothesis, our observations show that co-depletion of AP2XI-2 and AP2XII-1 leads to the activation of seven AP2 transcription factors and one C2H2 zinc finger transcription factor, all of which are subject to control by MORC and HDAC3 (Extended Data Fig. 11b–d).

## Discussion

Simultaneous KD of AP2XII-1 and AP2XI-2 efficiently triggers the pre-sexual transcriptional program in *Toxoplasma*, outperforming other methods. This study advances our understanding of pre-gamete biology, illustrating endopolygeny with karyokinesis as the meronts' preferred division mode, similar to the case for *Cystoisospora suis* but distinct from that for *Sarcocystis neurona*[29,37]. All predefined morphotypes (A–E), including type E expected to evolve into gametes, were observed. However, fully matured microgametocytes and macrogametocytes were not found, possibly owing to complex genetic requirements[38,39] or specialized metabolic conditions[40].

Converging pieces of evidence support the hypothesis that AP2XII-1 and AP2XI-2 are able to form homodimers but also heterodimers to silence merozoite genes in tachyzoites by binding to their promoters and recruiting MORC and HDAC3 (Extended Data Fig. 12a,b). MORC in turn forms dimers that topologically entrap DNA loops[41], leading to chromatin condensation that limits DNA accessibility to transcription factors and suppresses gene expression[42]. AP2XII-1 and AP2XI-2 also control the expression of secondary AP2 proteins specific to pre-gametes in the tachyzoite (Extended Data Fig. 12c). Operating as downstream activators or repressors, these transcription factors have the potential to significantly influence developmental trajectories post-merogony (for example, sex determination as shown for *Plasmodium falciparum*[38,39]). Fine-tuning their activity in mature in vitro-cultured merozoites holds promise for functional gamete production and in vitro fertilization[43].

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

# Methods

## Parasites and human cell cultures

Primary HFFs (ATCC CCL-171) were cultured in Dulbecco's modified Eagle medium (Invitrogen) supplemented with 10% heat-inactivated fetal bovine serum (Invitrogen), 10 mM 4-(2-hydroxyethyl)-1-piperazine-ethanesulfonic acid (HEPES) buffer pH 7.2, 2 mM L-glutamine and 50 µg ml$^{-1}$ penicillin and streptomycin (Invitrogen). Cells were incubated at 37 °C and 5% $CO_2$. *Toxoplasma* strains used in this study and listed in Supplementary Table 1 were maintained in vitro by serial passage on monolayers of HFFs. Cultures were free of mycoplasma as determined by qualitative PCR.

## Reagents

The following primary antibodies were used in the immunofluorescence, immunoblotting and/or ChIP assays: rabbit anti-TgHDAC3 (Research Resource Identifier (RRID): AB_2713903), rabbit anti-TgGAP45 (gift from Prof. Dominique Soldati), mouse anti-HA tag (Roche, RRID: AB_2314622), rabbit anti-HA Tag (Cell Signaling Technology, RRID: AB_1549585), rabbit anti-mCherry (Cell Signaling Technology, RRID: AB_2799246), rabbit anti-Flag (Cell Signaling Technology, RRID: AB_2798687), mouse anti-MYC clone 9B11 (RRID: AB_2148465), H3K9me3 (Diagenode, RRID: AB_2616044), rabbit anti-acetyl-histone H4, pan (Lys5,8,12) (Millipore, RRID: AB_310270), rat anti-IMC7 and centrin 1 (gift from Prof. M. J. Gubbels), mouse anti-IMC1 (gift from Prof. G. E. Ward), mouse anti-AtRx antibody clone 11G8 (ref. 24) and mouse anti-GRA11b[14]. We have also raised homemade antibodies to linear peptides in rabbits corresponding to the following proteins: MORC_Peptide2 (C+SGAPIWTGERGSGA); AP2XI-2 (C+HAFKTRRTEAAT); TGME49_273980 or GRA80 (C+RPPWAPGAGPEN); TGME49_243940 or GRA81 (C+QKELAEVAQRALEN); TGME49_277230 or GRA82 (C+SDVNTEGDATVANPE); TGME49_209985 or ROP26 (CQETVQGNGETQL); SRS48 family (CKALIEVKGVPK); SRS59B or K (C+IHVPGTDSTSSGPGS); TGME49_314250 or BRP1 (C+QVKEGTKNNKGLSDK); TGME49_307640 or CK2 kinase (C+IRAQYHAYKGKYSHA); and TGME49_306455 (C+DGRTPVDRVFEE). They were manufactured by Eurogentec and used for immunofluorescence, immunoblotting and/or ChIP. Secondary immunofluorescent antibodies were coupled with Alexa Fluor 488 or Alexa Fluor 594 (Thermo Fisher Scientific). Secondary antibodies used in western blotting were conjugated to alkaline phosphatase (Promega).

## Auxin-induced degradation

Degradation of AP2XII-1–mAID–HA, AP2XI-2–mAID–HA and AP2XII-1–mAID–HA plus AP2XI-2–mAID–MYC was achieved with IAA (Sigma-Aldrich number 45533). A stock of 500 mM IAA dissolved in 100% ethanol at a ratio of 1:1,000 was used to degrade mAID-tagged proteins to a final concentration of 500 µM. The mock treatment consisted of an equivalent volume of 100% ethanol at a final concentration of 0.0789% (w/v). To monitor the degradation of mAID-tagged proteins, parasites grown in HFF monolayers were treated with auxin or ethanol alone for various time intervals at 37 °C. After treatment, parasites were collected and analysed by immunofluorescence or western blotting.

## Immunofluorescence microscopy

*Toxoplasma*-infected HFF cells grown on coverslips were fixed in 3% formaldehyde for 20 min at room temperature, permeabilized with 0.1% (v/v) Triton X-100 for 15 min, and blocked in phosphate-buffered saline (PBS) containing 3% (w/v) BSA. Cells were then incubated with primary antibodies for 1 h, followed by the addition of secondary antibodies conjugated to Alexa Fluor 488 or 594 (Molecular Probes). Nuclei were stained with Hoechst 33258 (2 µg ml$^{-1}$ in PBS) for 10 min at room temperature. After washing four times in PBS, coverslips were mounted on a glass slide with Mowiol mounting medium, and images were acquired with a fluorescence microscope ZEISS ApoTome.2 and processed with ZEN software (Carl Zeiss).

For IFA of in vivo stages in the cat, small intestines of infected kittens from a previous study[16] embedded in paraffin were sectioned to 3 µm and dried overnight at 37 °C. Deparaffinization was carried out first three times for 2 min in xylene, and then the sections were washed twice for 1 min in 100% ethanol and finally rehydrated sequentially for 1 min in 96% ethanol, and then 70% ethanol and water. For antigen retrieval, samples were boiled in a pressure cooker for 20 min in citrate buffer at pH 6.1 (Dako Target Retrieval Solution, S2369) and transferred to water. Cells were permeabilized in 0.3% Triton X-100 in PBS and blocked with fetal calf serum (FCS). Staining was carried out overnight at 4 °C using the following combinations: mouse anti-GRA11b[14] with rabbit anti-GRA80, anti-IMC1 or anti-GAP45 (the last two being gifts from Prof. Dominique Soldati) or rabbit anti-GRA80 with rat immune serum recognizing merozoite proteins in 20% FCS and 0.3% Triton X-100 in PBS. The samples were then washed and incubated with 1 µg ml$^{-1}$ DAPI, 20% FCS and 0.3% Triton X-100 in PBS and appropriate combinations of anti-rabbit Alexa Fluor 488, 555 or 594 (Invitrogen, A11070, A32794 or A11072) with anti-mouse Alexa Fluor 488, 594 or 647 (Invitrogen, A11017, A11005 or A21235) or anti-rat Alexa Fluor 488 (Invitrogen, A11006) with anti-rabbit Alexa Fluor 594 (Invitrogen, A11072) for 1 h at room temperature. After three washes, samples were mounted with Vectashield and imaged either with a Leica DMI 6000 B epi-fluorescence microscope or a Leica SP8 confocal microscope. Confocal images were deconvoluted using SVI Huygens Professional. Maximum-intensity projections were carried out using FIJI 2.9.1.

## Transmission electron microscopy

For ultrastructural observations, *Toxoplasma*-infected HFFs grown as monolayers on a 6-well dish were exposed to 500 µM IAA or ethanol solvent as described above before fixation 24 h or 40 h post-infection in 2.5% glutaraldehyde in 0.1 mM sodium cacodylate (pH 7.4) and processed as described previously[44]. Ultrathin sections of infected cells were stained with osmium tetraoxide before examination with a Hitachi 7600 electron microscope under 80 kV equipped with a dual AMT CCD camera system.

## Western blot

Immunoblot analysis of protein was carried out as described in ref. 45. Briefly, about 10$^7$ cells were lysed and sonicated in 50 µl lysis buffer (10 mM Tris-HCl, pH 6.8, 0.5% SDS (v/v), 10% glycerol (v/v), 1 mM EDTA and protease inhibitor cocktail). Proteins were separated using SDS–polyacrylamide gel electrophoresis, transferred by liquid transfer to a polyvinylidene fluoride membrane (Immobilon-P; EMD Millipore), and western blots were probed with the appropriate primary antibodies and alkaline phosphatase-conjugated or horseradish peroxidase-conjugated secondary goat antibodies. Signals were detected using NBT-BCIP (Amresco) or an enhanced chemiluminescence system (Thermo Scientific).

## Plasmid construction

The plasmids and primers used in this work for the genes of interest (GOIs) are listed in Supplementary Table 1. To construct the vector pLIC-GOI-HA-Flag, pLIC-GOI-mAID-HA or pLIC-GOI-mAID-(MYC)2, the coding sequence of the GOI was amplified with the primers LIC-GOI-Fwd and LIC-GOI-Rev using genomic *Toxoplasma* DNA as a template. The resulting PCR product was cloned into the vector pLIC-HF-dhfr or pLIC-mCherry-dhfr using the ligation-independent cloning (LIC) method. Specific guide RNA for the GOI, based on the CRISPR–cas9 editing method, was cloned into the plasmid pTOXO_Cas9-CRISPR[1]. Twenty-base oligonucleotides corresponding to specific GOIs were cloned using the Golden Gate strategy. Briefly, the primers GOI-gRNA-Fwd and GOI-gRNA-Rev containing the single guide RNA targeting the genomic sequence of the GOI were phosphorylated,

annealed and ligated into the pTOXO_Cas9-CRISPR plasmid linearized with BsaI, resulting in pTOXO_Cas9-CRISPR::sgGOI.

### *Toxoplasma* transfection

Parasite strains were electroporated with vectors in Cytomix buffer (120 mM KCl, 0.15 mM $CaCl_2$, 10 mM $K_2HPO_4$ and $KH_2PO_4$ pH 7.6, 25 mM HEPES pH 7.6, 2 mM EGTA, 5 mM $MgCl_2$) using a BTX ECM 630 machine (Harvard Apparatus). Electroporation was carried out in a 2-mm cuvette at 1,100 V, 25 Ω and 25 µF. Antibiotics (concentration) used were chloramphenicol (20 µM), mycophenolic acid (25 µg ml$^{-1}$) with xanthine (50 µg ml$^{-1}$), pyrimethamine (3 µM) or 5-fluorodeoxyuracil (10 µM) as needed. Stable transgenic parasites were selected with the appropriate antibiotic, cloned in 96-well plates by limiting dilution, and verified by immunofluorescence assay or genomic analysis.

### Chromatographic purification of Flag-tagged proteins

*Toxoplasma* extracts from RHΔku80 or PruΔku80 cells stably expressing HA–Flag-tagged AP2XII-1 or AP2XI-2 proteins, respectively, were incubated with anti-Flag M2 affinity gel (Sigma-Aldrich) for 1 h at 4 °C. Beads were washed with 10 column volumes of BC500 buffer (20 mM Tris-HCl, pH 8.0, 500 mM KCl, 20% glycerol, 1 mM EDTA, 1 mM dithiothreitol, 0.5% NP-40 and protease inhibitors). Bound polypeptides were eluted stepwise with 250 µg ml$^{-1}$ Flag peptide (Sigma-Aldrich) diluted in BC100 buffer. For size-exclusion chromatography, protein eluates were loaded onto a Superose 6 HR 10/30 column equilibrated with BC500. The flow rate was set at 0.35 ml min$^{-1}$, and 0.5-ml fractions were collected.

### MS-based quantitative analyses of parasite proteomes and AP2 interactomes

**Sample preparation.** For proteome-wide analyses, HFF cells were grown to confluence, infected with the RH (*AP2XII-1* and *AP2XI-2* KD) strain and treated with IAA for 24 h, 32 h and 48 h or mock-treated. Proteins were extracted using cell lysis buffer (Invitrogen). Three biological replicates were prepared and analysed for each condition. For characterization of HA–Flag-tagged AP2XII-1 or AP2XI-2 interactomes, eluted proteins were solubilized in Laemmli buffer. Three biological replicates were prepared for each bait protein and for the negative control. Proteins were stacked in the top of a 4–12% NuPAGE gel (Invitrogen) and stained with Coomassie blue R-250 (Bio-Rad) before in-gel digestion using modified trypsin (Promega, sequencing grade) as previously described[1].

**Nanoliquid chromatography coupled to MS/MS analyses.** The resulting peptides were analysed by online nanoliquid chromatography coupled to an MS/MS instrument (Ultimate 3000 RSLCnano and Q-Exactive HF, Thermo Fisher Scientific) using a 360-min gradient for proteome-wide analysis and a 200-min gradient for interactome characterization. For this, peptides were sampled on a 300 µm × 5 mm PepMap C18 precolumn and separated in a 200 cm µPAC column (PharmaFluidics) or a 75 µm × 250 mm C18 column (Aurora Generation 2, 1.7 µm, IonOpticks) for, respectively, proteome-wide and interactome analyses. MS and MS/MS data were acquired using Xcalibur software version 4.0 (Thermo Scientific).

**Protein identification and quantification.** Peptides and proteins were identified by Mascot (version 2.8.0, Matrix Science) through concomitant searches against the *T. gondii* database (ME49 taxonomy, version 58 downloaded from ToxoDB), the Uniprot database (*Homo sapiens* taxonomy) and a homemade database containing the sequences of classical contaminant proteins found in proteomic analyses (human keratins, trypsin and so on). Trypsin was chosen as the enzyme and two missed cleavages were allowed. Precursor and fragment mass error tolerances were set, respectively, at 10 and 20 ppm. Peptide modifications allowed during the search were: carbamidomethyl (C, fixed), acetyl

(protein amino terminus, variable) and oxidation (M, variable). The Proline software (version 2.2.0) was used for the compilation, grouping and filtering of the results (conservation of rank 1 peptides, peptide length ≥ 6 amino acids, false discovery rate of peptide-spectrum-match identifications < 1% and minimum of one specific peptide per identified protein group). Proline was then used to carry out an MS1 label-free quantification of the identified protein groups based on razor and specific peptides.

**Statistical analyses.** Statistical analyses were carried out using ProStaR[46]. Proteins identified in the reverse and contaminant databases or matched to human sequences were discarded. For proteome-wide analyses, only proteins identified by MS/MS in a minimum of two replicates of one condition and quantified in the three replicates of one condition were conserved. After log$_2$ transformation, abundance values were normalized using the variance-stabilizing normalization method, before missing-value imputation (structured least squares algorithm (SLSA) for partially observed values in the condition and DetQuantile algorithm for totally absent values in the condition). For comparison of each IAA-treated condition to the mock-treated condition, statistical testing was conducted with limma, whereby differentially expressed proteins were selected using a log$_2$[FC] cutoff of 1 and a *P*-value cutoff of 0.01, allowing one to reach a false discovery rate inferior to 5% according to the Benjamini–Hochberg estimator. Proteins found differentially abundant but identified by MS/MS in fewer than two replicates, and detected in fewer than three replicates, in the condition in which they were found to be more abundant were invalidated (*P* value = 1). Protein abundances measured in the four different conditions were also compared globally by ANOVA using Perseus; *q* values were obtained by Benjamini–Hochberg correction.

For interactome analysis, only proteins identified by MS/MS in the three replicates of one condition and proteins quantified with a minimum of five peptides were conserved. After log$_2$ transformation, abundance values were normalized by condition-wise median centring, before missing-value imputation (SLSA algorithm for partially observed values in the condition and DetQuantile algorithm for totally absent values in the condition). Statistical testing was conducted with limma, whereby differentially expressed proteins were selected using a *P*-value cutoff of 0.01 and FC cutoffs of 5 and 3, respectively, for comparison of each AP2 interactome with negative control and AP2 interactomes together, allowing one to reach a false discovery rate inferior to 1% according to the Benjamini–Hochberg estimator. The relative abundances of AP2-associated proteins were determined using the iBAQ metrics; only proteins with an iBAQ ratio of at least 0.1 in relation to the bait protein were considered.

### MS-based proteomic analyses of SEC fractions

Protein bands were excised from colloidal blue-stained gels (Thermo Fisher Scientific) before in-gel digestion using modified trypsin (Promega, sequencing grade) as previously described[1]. Resulting peptides were analysed by online nanoliquid chromatography coupled to MS/MS (UltiMate 3000 RSLCnano and Orbitrap Exploris 480, Thermo Scientific). Peptides were sampled on a 300 µm × 5 mm PepMap C18 precolumn and separated on a 75 µm × 250 mm C18 column (Aurora Generation 2, 1.6 µm, IonOpticks) using a 25-min gradient. MS and MS/MS data were acquired using Xcalibur version 4.0 (Thermo Scientific). Peptides and proteins were identified using Mascot (version 2.8.0) through concomitant searches against the *T. gondii* database (ME49 taxonomy, version 58 downloaded from ToxoDB), the Uniprot database (*T. ni* taxonomy) and a homemade database containing the sequences of classical contaminant proteins found in proteomic analyses (human keratins, trypsin and so on). Trypsin/P was chosen as the enzyme and two missed cleavages were allowed. Precursor and fragment mass error tolerances were set, respectively, at 10 and 20 ppm. Peptide modifications allowed during the search were: carbamidomethyl (C, fixed),

acetyl (protein N terminus, variable) and oxidation (M, variable). The Proline software (version 2.2.0) was used for the compilation, grouping and filtering of the results (conservation of rank 1 peptides, peptide length ≥ 6 amino acids, false discovery rate of peptide-spectrum-match identifications < 1% and minimum of one specific peptide per identified protein group). iBAQ values were calculated for each protein group in Proline using MS1 intensities of specific and razor peptides.

## ChIP coupled with Illumina sequencing

**ChIP.** HFF cells were grown to confluence and infected with KD strains as indicated in the figure legends. Collected intracellular parasites were crosslinked with formaldehyde (final concentration 1%) for 8 min at room temperature, and crosslinking was stopped by addition of glycine (final concentration 0.125 M) for 5 min at room temperature. The parasites were lysed in ice-cold lysis buffer A (50 mM HEPES KOH pH 7.5, 140 mM NaCl, 1 mM EDTA, 10% glycerol, 0.5% NP-40, 0.125% Triton X-100 and protease inhibitor cocktail) and after centrifugation, crosslinked chromatin was sheared in buffer B (1 mM EDTA pH 8.0, 0.5 mM EGTA pH 8.0, 10 mM Tris pH 8.0 and protease inhibitor cocktail) by sonication with a Diagenode Biorupter. Samples were sonicated for 16 cycles (30 s on and 30 s off) to achieve an average size of 200–500 base pairs. Sheared chromatin, 5% BSA, a protease inhibitor cocktail, 10% Triton X-100, 10% deoxycholate, magnetic beads coated with DiaMag protein A (Diagenode) and antibodies to epitope tags (HA or MYC) or the protein of interest (MORC or HDAC3) were used for immunoprecipitation. A rabbit IgG antiserum served as a control mock. After overnight incubation at 4 °C on a rotating wheel, chromatin–antibody complexes were washed and eluted from the beads using the iDeal ChIP–seq kit for transcription factors (Diagenode) according to the manufacturer's protocol. Samples were de-crosslinked by heating for 4 h at 65 °C. DNA was purified using the IPure kit (Diagenode) and quantified using Qubit Assays (Thermo Fisher Scientific) according to the manufacturer's protocol. For ChIP-seq, the purified DNA was used for library preparation and subsequently sequenced by Array-star (USA).

**Library preparation, sequencing and data analysis (Arraystar).** ChIP–seq libraries were prepared according to the Illumina protocol "Preparing Samples for ChIP Sequencing of DNA". For library preparation, 10 ng of DNA from each sample was converted to blunt-end phosphorylated DNA fragments using T4 DNA polymerase, Klenow polymerase and T4 polymerase (NEB); an 'A' base was added to the 3' end of the blunt-end phosphorylated DNA fragments using the polymerase activity of Klenow (Exo-Minus) polymerase (NEB); Illumina genomic adapters were ligated to the A-tailed DNA fragments; PCR amplification to enrich the ligated fragments was carried out using Phusion High Fidelity PCR Master Mix with HF Buffer (Finnzymes Oy). The enriched product of about 200–700 bp was excised from the gel and purified. For sequencing, the library was denatured with 0.1 M NaOH to generate single-stranded DNA molecules and loaded into flow cell channels at a concentration of 8 pM and amplified in situ using TruSeq Rapid SR cluster kit (number GD-402-4001, Illumina). Sequencing was carried out for 100 cycles on the Illumina HiSeq 4000 according to the manufacturer's instructions. For data analysis, after the sequencing platform generated the sequencing images, the stages of image analysis and base calling were carried out using Off-Line Base-caller software (OLB V1.8). After passing the Solexa CHASTITY quality filter, the clean reads were aligned to the *T. gondii* reference genome (TGME49) using BOWTIE V2 and then converted and sorted using Bam-tools. Aligned reads were used for peak calling of the ChIP-enriched peaks using MACS v2.2 with a cutoff *P* value of $10^{-4}$. For Integrated Genome Browser visualization and gene-centred analysis using Deep-tools, MACS2-generated bedgraph files were processed with the command 'sort -k1,1 -k2,2n 5_treat_pileup.bdg > 5_treat_pileup-sorted.bdg', and then converted using the BedGraphToBigWig program (ENCODE

project). The Deeptools analysis was generated using the command computeMatrix reference point, with the following parameters: –minThreshold 2, –binSize 10 and –averageTypeBins sum. Plotprofile or heat map was then used with *k*-means clustering when applicable. Inter-sample comparisons were obtained using the nf-core ChIP–seq workflow with standard parameters[47]. From this pipeline, HOMER (annotatePeaks) was used to analyse peak distribution relative to gene features. All of these raw and processed files can be found at GSE222819.

## RNA-seq and sequence alignment

Total RNAs were extracted and purified using TRIzol (Invitrogen) and RNeasy Plus Mini Kit (Qiagen). RNA quantity and quality were measured using a NanoDrop 2000 (Thermo Scientific). For each condition, RNAs were prepared from three biological replicates. RNA integrity was assessed by standard non-denaturing 1.2% TBE agarose gel electrophoresis. RNA-seq was carried out following standard Illumina protocols, by Novogene (Cambridge, UK). Briefly, RNA quantity, integrity and purity were determined using the Agilent 5400 Fragment Analyzer System (Agilent Technologies). The RNA quality number ranged from 7.8 to 10 for all samples, which was considered sufficient. mRNAs were purified from total RNA using poly-T oligonucleotide-attached magnetic beads. After fragmentation, the first-strand cDNA was synthesized using random hexamer primers. Then the second-strand cDNA was synthesized using dUTP, instead of dTTP. The directional library was ready after end repair, A-tailing, adapter ligation, size selection, USER enzyme digestion, amplification and purification. The library was checked with Qubit and real-time PCR for quantification and a bioanalyser for size distribution detection. Quantified libraries were pooled and sequenced on Illumina platforms, according to effective library concentration and data amount. The samples were sequenced on the Illumina NovaSeq platform (2 × 150 bp, strand-specific sequencing) and generated about 40 million paired-end reads for each sample. The quality of the raw sequencing reads was assessed using FastQC (https://www.bioinformatics.babraham.ac.uk/projects/fastqc/) and MultiQC. For quantification and normalization of the expression data, the FASTQ reads were aligned to the ToxoDB-49 build of the *T. gondii* ME49 genome using Subread version 2.0.1 with the following options: subread-align -d 50 -D 600 –sortReadsByCoordinates. Read counts for each gene were calculated using featureCounts from the Subread package. Differential expression analysis was conducted using DESeq2 and default settings within the iDEP.96 web interface[48]. Transcripts were quantified and normalized using TPMCalculator. The Illumina RNA-seq dataset generated during this study is available at the National Center for Biotechnology Information: BioProject number PRJNA921935.

## Nanopore direct RNA-seq

The mRNA library preparation followed the SQK-RNA002 kit (Oxford Nanopore)–recommended protocol; the only modification was the input mRNA quantity increased from 500 to 1,000 ng, and all other consumables and parameters were standard. Final yields were evaluated using the Qubit HS dsDNA kit (Thermo Fisher Scientific, Q32851) with minimum RNA preparations reaching at least 200 ng. For all conditions, sequencing was carried out on FLO-MIN106 flow cells using either a MinION MK1C or MinION sequencer. All datasets were subsequently base called (high-accuracy base calling) with a Guppy version higher than 5.0.1 with a *Q* score cutoff of >7. Long-read alignment was carried out by Minimap2 as previously described[49]. Sam files were converted to bam and sorted using Samtools 1.4. Alignments were converted and sorted using Samtools 1.4.1. For the three described samples, *Toxoplasma* aligned reads range between 600,000 and 800,000. The Nanopore direct RNA-seq dataset is available at the National Center for Biotechnology Information: BioProject number PRJNA921935.

## ATAC–seq

Intracellular tachyzoites (non-treated or IAA treated for 24 h) were prepared using HFF cell monolayers in a T175 format, which was freshly scraped, gently homogenized by pipetting and centrifuged at 500$g$. Before initiating the transposition protocol, the pellet was gently washed with warm Dulbecco's PBS (Life technologies) and resuspended in 500 µl of cold PBS + protease inhibitor (Diagenode kit). Nuclei preparation, permeabilization, Tn5 transposition and library preparation was carried out following precisely the Diagenode ATAC–seq kit protocol (C01080002). Nucleus permeabilization was carried out on an estimated 100,000 tachyzoites by diluting 10 µl of Dulbecco's cell suspension (from one T175 resuspended in 500 µl) in 240 µl of Dulbecco's PBS + protease inhibitor (1/25 dilution). From this dilution, 50 µl was then taken to carry out the transposition reaction. Of note, the permeabilization protocol used a 3-min 0.02% digitonin (Promega) exposure. Following the Tn5 reaction, libraries were amplified using the Diagenode 24 UDI kit 1 (ref 01011034) following standard protocol procedures. Libraries were multiplexed and sequenced on a single Novaseq6000 lane by Fasteris (Genesupport SA) using 2 × 50 cycles, generating on average 27 million reads. Demultiplexing of raw reads was performed by bcl2fastq V3, and trimming, quality control, alignment to the ME49 reference genome (using bwa2) and duplicate read merging (using Picard) were carried out by the nf-core ATAQ-SEQ pipeline[47]. For Integrated Genome Browser visualization and gene-centred analysis using Deeptools, Picard merged bam files were converted to bigWig file format using a bin size of 5 by bamCoverage (Deeptools). The Deeptools analysis was then generated using 'computeMatrix reference point', with the following parameters: –minThreshold 2,–binSize 10 and –averageTypeBins sum. Quantitative analysis of untreated versus 24-h IAA conditions was carried out by nf-core through a broad peak calling and annotation (MACS2) followed by HOMER (annotatePeaks) to analyse peak distribution relative to gene features. Reads were counted on annotated peaks by featureCounts and counts were processed by DeSeq2 to generate global statistical analysis of peak intensities between conditions using biological duplicates. All of these raw and processed files can be found at GSE222832.

## Gene synthesis for recombinant coexpression of *T. gondii* AP2XI-2 and AP2XII-I

Gene synthesis for all insect cell codon-optimized constructs was provided by GenScript. (Strep)$_2$–APXI-2 and AP2XII-1–Flag or AP2IX-6–Flag genes were cloned within the coexpression donor vector pFastBac dual, which accepts two constructs. The (Strep)$_2$–AP2XI-2 expression cassette was derived from the TGME49_310900 gene with a fused dual Strep tag and tobacco etch virus (TEV) site in the N terminus. AP2XII-1–Flag or AP2IX-6–Flag was derived from the full-length TGME49_218960 and TGME49_290180 genes, respectively, with an additional non-cleavable Flag tag on the carboxy terminus. The (Strep)$_2$–AP2XI-2 expression cassette was under the control of the polyhedrin promoter; the AP2XII-1–Flag or AP2IX-6–Flag was under the control of the P10 promoter.

## Generation of baculovirus

Bacmid cloning steps and baculovirus generation were carried out using EMBacY baculovirus (gift from I. Berger), which contains a yellow fluorescent protein reporter gene in the virus backbone. The established standard cloning and transfection protocols set up within the EMBL Grenoble eukaryotic expression facility were used. Although baculovirus synthesis (V0) and amplification (to V1) were carried out with SF21 cells cultured in SF900 III medium (Life Technologies), large-scale expression cultures were carried out with Hi-5 cells cultured in the protein-free ESF 921 insect cell culture medium (Expression System) and infected with 0.8–1.0% (v/v) of generation 2 (V1) baculovirus suspensions and collected 72 h after infection.

## (Strep)$_2$–AP2XI-2 and AP2XII-1–Flag or (Strep)$_2$–AP2XI-2 and AP2XIX-6–Flag expression and purification

For purification, three cell pellets of bout 500 ml of Hi-5 culture were each resuspended in 50 ml of lysis buffer (50 mM Tris (pH 8.0), 400 mM NaCl and 2 mM β-mercaptoethanol (BME)) in the presence of an anti-protease cocktail (Complete EDTA-free, Roche) and 1 µl of Benzonase (Merck Millipore, 70746). Lysis was carried out on ice by sonication for 3 min (30-s on/ 30-s off, 45° amplitude). After the lysis step, 10% of glycerol was added. Clarification was then carried out by centrifugation for 1 h at 16,000$g$ and 4 °C and vacuum filtration using 45-µm nylon filter systems (SteriFlip, Merck Millipore). Before purification, tetrameric avidin (Biolock, IBA Lifescience) was added to the clarified lysate (1/1,000 v/v), which was then batch incubated for 20 min with 3 ml of Strep-Tactin XT (IBA Lifescience). Following the incubation, the resin was retained on a glass column and washed three times using 6 ml of lysis buffer (W1), 6 ml of lysis buffer with NaCl content at 1 M (W2) and 6 ml of lysis buffer (W3). The elution was then carried out using 1× BXT buffer (IBA Lifescience) containing 50 mM biotin, 100 mM Tris pH 8 and 150 mM NaCl. This initial 1× solution was further supplemented with 300 mM NaCl, 2 mM BME and 10% glycerol. Following Strep-Tactin XT elution, the sample was concentrated to 500 µl using a 100-kDa concentrator (Amicon Ultra 4, Merck Millipore) injected on an ÄKTA pure FPLC using a Superose 6 Increase column 10/300 GL (Cytiva) running in 50 mM Tris pH 8, 400 mM NaCl and 1 mM BME.

## Mouse infection and experimental survey

Six-week-old NMRI, CD1 or BALB/c mice were obtained from Janvier Laboratories. Female mice were used for all studies. For intraperitoneal infection, tachyzoites were grown in vitro and extracted from host cells by passage through a 27-gauge needle, washed three times in PBS, and quantified with a haemocytometer. Parasites were diluted in Hank's balanced salt solution (Life), and mice were inoculated by the intraperitoneal route with tachyzoites of each strain (in 200 µl volume) using a 28-gauge needle. Animal euthanasia was completed in an approved CO$_2$ chamber. For immunolabelling on histological sections of the brains, the brains were removed from mice, entirely embedded in a paraffin wax block and cut into 5-µm-thick layers using a microtome.

## Statistics and reproducibility

Sample sizes were not predetermined and were chosen according to previous literature. Experiments were carried out in biological replicates and provided consistent statistically relevant results. No method of randomization was used. All experiments were carried out in independent biological replicates as stated for each experiment in the manuscript. All corresponding treatment and control samples from ChIP–seq and RNA-seq were processed at the same time to minimize technical variation. Investigators were not blinded during the experiments. Statistical significance was evaluated using $P$ values from unpaired two-tailed Student $t$-tests. Data are presented as the mean ± s.d. Significance was set to a $P$ value of <0.05. All of the micrographs shown are representatives from three independently conducted experiments, with similar results obtained.

## Ethics statement

Mouse care and experimental procedures were carried out under pathogen-free conditions in accordance with established institutional guidance and approved protocols from the Institutional Animal Care and Use Committee of the University Grenoble Alpes (APAFIS number 4536-2016031 017075121 v5). Animal experiments were carried out under the direct supervision of a veterinary specialist, and according to Swiss law and guidelines on Animal Welfare and the specific regulations of the Canton of Zurich under permit numbers 130/2012 and 019/2016, as approved by the Veterinary Office and the Ethics Committee of the Canton of Zurich (Kantonales Veterinäramt Zürich).

## Reporting summary

Further information on research design is available in the Nature Portfolio Reporting Summary linked to this article.

## Data availability

Nanopore and Illumina RNA-seq data that support the findings of this study have been deposited under the BioProject number PRJNA921935. The ChIP–seq and ATAC–seq data have been deposited to the Gene Expression Omnibus database under accession numbers GSE222819 and GSE222832, respectively. The MS proteomics data have been uploaded to the ProteomeXchange Consortium through the PRIDE partner repository with the dataset identifiers PXD039400 and PXD042658 for, respectively, the proteome-wide and interactome analyses. Processed proteomics data are available in Supplementary Table 3. Source data are provided with this paper.

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

**Acknowledgements** We are grateful to the developers of the ToxoDB.org Genome Resource. ToxoDB and EuPathDB are part of the National Institutes of Health National Institutes of Allergy and Infectious Diseases (NIH NIAID)-funded Bioinformatics Resource Center. We thank the technical staff of the Electron Microscopy Core Facility at the Johns Hopkins University School of Medicine Microscopy Facility; M. J. Gubbels for providing several antibodies, encompassing rat anti-IMC7 and centrin1; and S. Lourido for providing *ΔBFD1*/DD-BFD1-Ty[15]. This work was supported by MSD Avenir (Project LatentToxoDiag, DS-2022-0017), the Laboratoire d'Excellence (LabEx) ParaFrap (ANR-11-LABX-0024), the Agence Nationale pour la Recherche (Project ApiNewDrug, ANR-21-CE35-0010-01; Project ApiMORCing, ANR-21-CE15-0002-01; Project ToxoP53, ANR–19-CE15-0026) and Fondation pour la Recherche Médicale (FRM Equipe, EQU202103012571). I.C. was supported by an NIH grant (R01 AI060767). MS-based proteomic experiments were partially supported by Agence Nationale de la Recherche under projects ProFI (Proteomics French Infrastructure, ANR-10-INBS-08) and GRAL, a program from the Chemistry Biology Health (CBH) Graduate School of University Grenoble Alpes (ANR-17-EURE-0003).

**Author contributions** M.-A.H. supervised the research and coordinated the collaboration. A.V.A., M.S., C.S., D.C.F., A.B., M.G.R., D.C., C.C., A.B.H. and M.-A.H. designed, carried out and interpreted the experimental work. Y.C. and C.B. carried out the MS analyses. I.C. carried out transmission electron microscopy and interpreted the associated results. C.R. carried out confocal microscopy of infected cat intestines. M.-A.H. wrote the paper with editorial support from I.C. and comments from all other authors.

**Competing interests** The authors declare no competing interests.

**Additional information**
**Correspondence and requests for materials** should be addressed to Mohamed-Ali Hakimi.

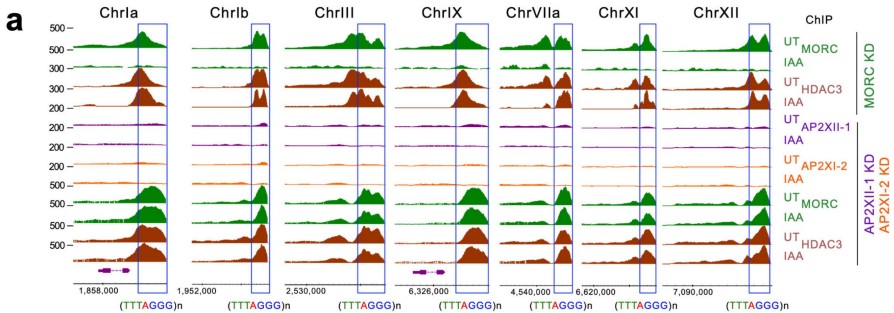

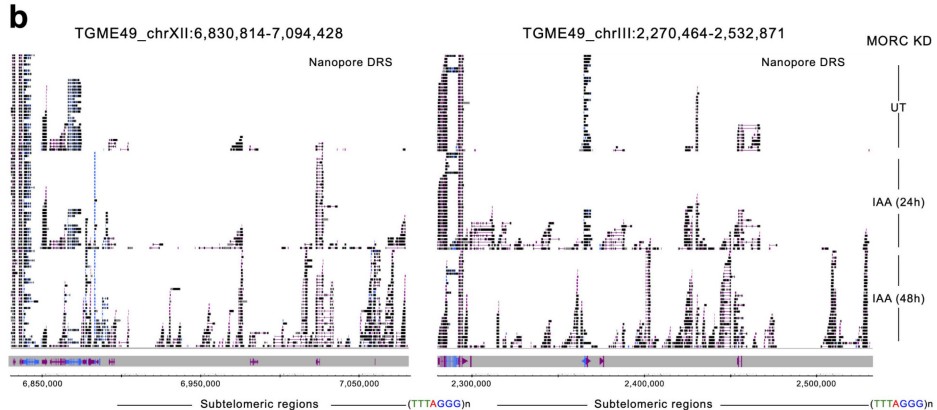

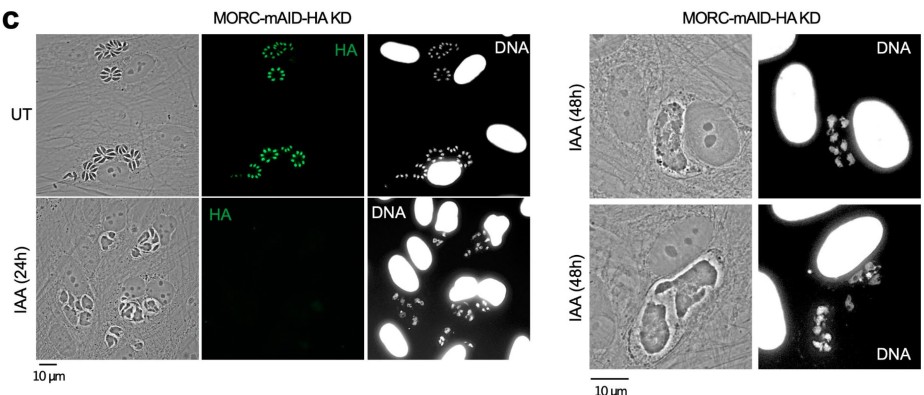

**Extended Data Fig. 1 | MORC binds to telomere and its depletion leads to subtelomeric non-coding RNAs reactivation and cell cycle disruption. a**, IGB views of MORC and HDAC3 ChIP-seq enrichment at chromosomal ends following MORC or AP2XII-1/AP2XI-2 depletion. Read density is on the y-axis, with telomeric repeats (TTTAGGG) marked. **b**, Nanopore direct RNA sequencing (DRS) read alignment of initially suppressed non-coding RNAs, observable post-MORC knockdown via IAA on the subtelomeric ends of chromosome III and XII. The y-axis shows read-depth. Positive strand reads are colored in magenta while negative strand reads are colored in blue. **c**, Expression levels of MORC over time are presented through IFA on cells infected with RH MORC–mAID–HA. Cells were fixed, permeabilized, and probed with HA antibodies (green) and Hoechst DNA-specific dye.

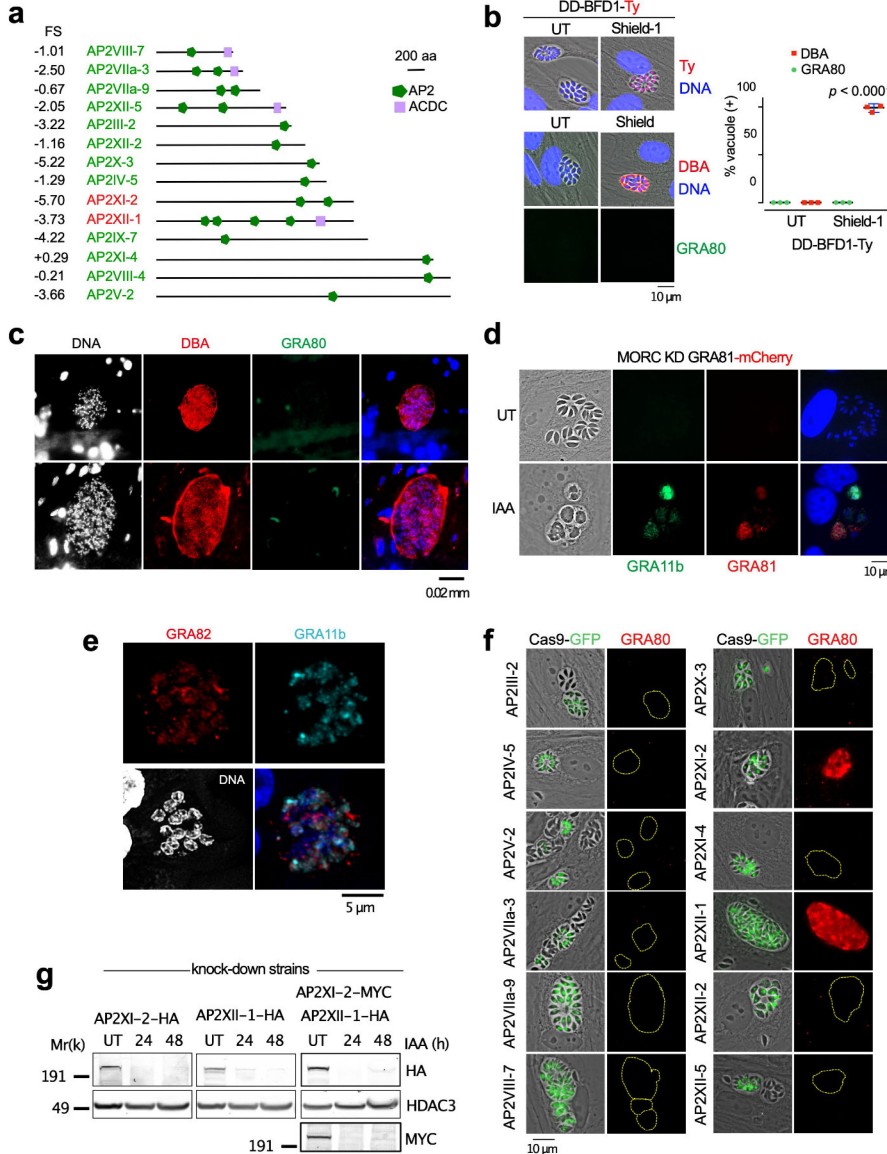

**Extended Data Fig. 2 | Evaluating stage-specific markers both in vitro and in vivo across varied genetic backgrounds. a**, Domain architectures of MORC AP2 partners, originally identified via immunoprecipitation and MS-proteomics, showcasing the AP2 (APETALA2) and ACDC (AP2-Coincident Domain primarily at the Carboxy-terminus) domains, as predicted by SMART and PFAM. **b**, Representative vacuoles of ΔBFD1/DD-BFD1-Ty parasites grown for 48 h with vehicle or 3 μM Shield-1, stained for Ty or DBA (red) and GRA80 (green). The graph on the right quantifies results (n = 50 vacuoles/dot), statistical analysis performed using one-tailed Student's t-test. Data are presented as mean values ± s.d. **c**, DBA staining (red) highlighted the glycosylated cyst wall in brain sections of mice chronically infected with ME49 type II strain. The sections were counterstained against GRA80 (green) and Hoechst DNA-specific dye. **d**, IFA on HFFs infected with RH MORC KD lineage harboring mCherry-tagged

GRA81 (*TGME49_243940*), a merozoite protein. UT and IAA-treated zoites were probed for GRA11b (green) and mCherry (red), and stained with Hoechst DNA-specific dye. **e**, Confocal microscopy of a meront in infected small intestine of a kitten co-stained with anti-GRA11b (cyan) and anti-GRA82 (red) antibodies, with DAPI employed for nuclear counterstaining. **f**, Representative images of intracellular parasites with disrupted AP2 genes due to transient CRISPR/Cas9 plasmid transfection. Cas9-GFP expression (green) indicates disruption, while merozoite marker GRA80 (red) is monitored in AP2 inactivated (GFP-positive) zoites. **g**, Expression levels of AP2XII-1 and AP2XI-2 in the single and double KD strains were monitored over time using Western blot. Post-IAA addition, samples were collected at the indicated time points and probed with HA, MYC, and HDAC3 antibodies. This was repeated three times, and a representative blot is displayed.

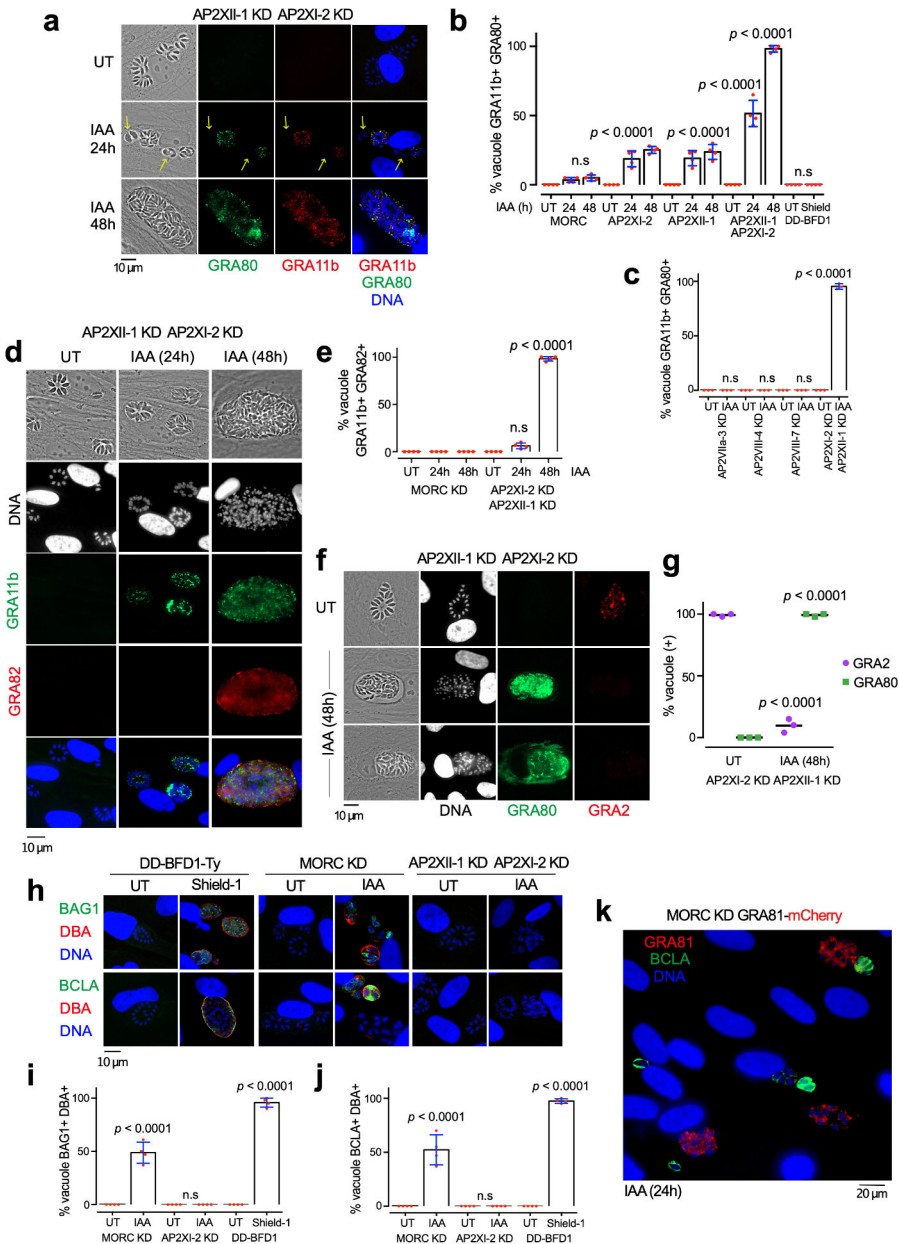

**Extended Data Fig. 3 | Quantification of stage-specific markers expression levels in the context of AP2XII-1/AP2XI-2 co-depletion, MORC depletion and BDF1 overexpression. a-b**, MORC and single or double AP2XII-1/AP2XI-2 KD parasites, untreated or IAA-treated (24 and 48 h), were compared with ΔBFD1/DD-BFD1-Ty parasites treated with vehicle or 3 µM Shield-1 (48 h). Parasites were stained with GRA11b (red) and GRA80 (green) antibodies, and Hoechst DNA-specific dye. Representative images of double KD vacuoles are shown on the left (**a**) the right graph (**b**) displays quantified results (n = 50 vacuoles/dot), analyzed using one-way ANOVA and Tukey's test. Data are presented as mean values ± s.d. **c**, Levels of merozoite markers after AP2VIIa-3, AP2VIII-4, and AP2VIII-7 depletion were compared with those after AP2XII-1/AP2XI-2 co-depletion (48 h). Displayed data represent mean ± s.d. of GRA11b(+)/GRA80(+) vacuole staining from three experiments (n = 50 vacuoles/dot). Statistics involved one-way ANOVA and Tukey's multiple comparison test. **d-e**, MORC and single or double AP2XII-1/AP2XI-2 KD parasites, untreated or IAA-treated (24 and 48 h) were co-stained with GRA11b (red) and GRA82 (green) antibodies. Representative images of double KD vacuoles are shown on the left (**d**) the right graph (**e**) displays quantified results (n = 50 vacuoles/dot), analyzed using one-way ANOVA and Tukey's test. Data are presented as mean values ± s.d. **f-g**, Representative vacuoles of AP2XII-1/ AP2XI-2 KD parasites, untreated or IAA-treated (48 h) (**f**), stained for GRA2 (red) and GRA80 (green); the graph on the right (**g**) quantifies results (n = 50 vacuoles/dot, 3 replicates/conditions), using one-tailed Student's t-test. **h-j**, Representative images of MORC, double AP2XII-1/AP2XI-2 KD, and ΔBFD1/ DD-BFD1-Ty treated with respective inducers and stained with DBA and bradyzoite markers BAG1 (red) or BCLA (green) (**h**). Graphs show % of BAG1 (**i**) or BCLA (**j**) positive vacuoles across three experiments (n = 50 vacuoles/dot), statistical analysis by one-way ANOVA and Tukey's test. Data are presented as mean values ± s.d. **k**, IAA-induced (24 h) vacuoles of MORC KD parasites containing GRA81 tagged with mCherry (in red), co-stained with a bradyzoite marker (BCLA, green), and Hoechst DNA-specific dye.

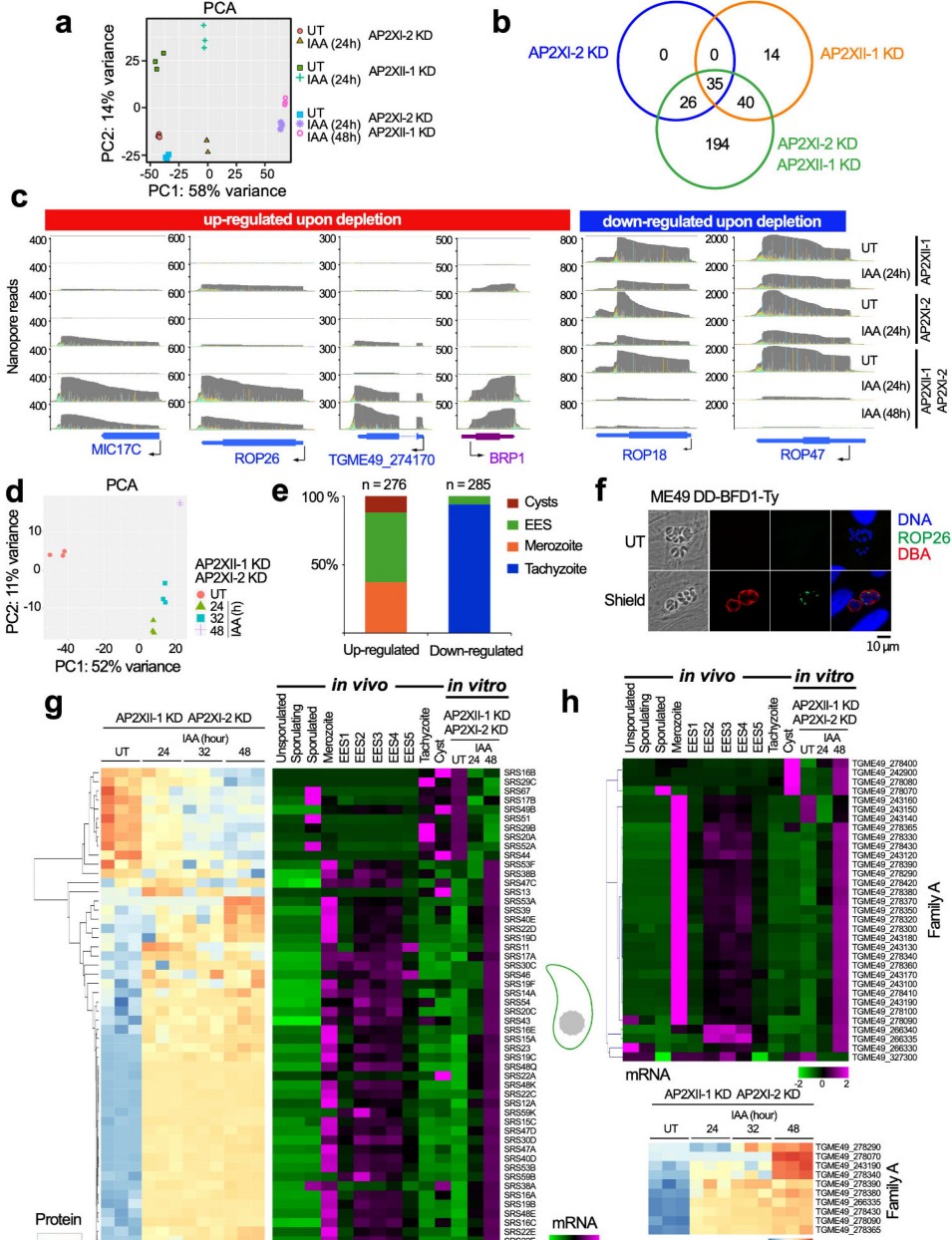

**Extended Data Fig. 4 | Co-depletion of AP2XI-2 and AP2XII-1 induces the expression of merozoite proteins, including a large repertoire of parasite surface proteins. a**, Principal Component Analysis (PCA) of mRNA sequencing data from biological triplicates of single KD or double KD parasites. Samples were collected from untreated conditions or after 24 or 48 h of IAA treatment. **b**, Venn diagram illustrating the overlapping genes that were upregulated in the three IAA-treated knockdown strains. Significant genes (FC > 8) were identified using DESeq2 with an independent-hypothesis-weighted approach and Benjamini–Hochberg false discovery rate (FDR) < 0.1. **c**, M-pileup representation of aligned Nanopore DRS reads at genes differentially expressed following IAA-induced knockdown of AP2XII-1 and AP2XI-2 individually or in combination. **d**, PCA illustrates the biological and technical variance between triplicate proteome samples extracted after 24-, 32-, and 48-hours post AP2XII-1/AP2XI-2 knockdown induction, juxtaposed with the untreated sample (UT). **e**, Histogram

delineating the distribution of up- and down-regulated proteins (n = 276 and 285, respectively) post AP2XII-1 and AP2XI-2 knockdown, categorized by their life stage association. **f**, Representative vacuoles of *ΔBFD1*/DD-BFD1-Ty parasites grown for 48 h with vehicle or 3 μM Shield-1, stained for ROP26 (green), DBA (red) and Hoechst DNA-specific dye (blue). **g-h**, Heat map showing hierarchical clustering analysis of selected SRS (**g**) and Family A (**h**) mRNA transcripts and their corresponding proteomic enrichments, which were significantly upregulated (Log2 FC > 2; *P*-value < 0.01) or downregulated (Log2 FC < −1; *P*-value < 0.01) following the simultaneous depletion of AP2XII-1 and AP2XI-2. The abundance of these transcripts is presented across different in vivo stages - merozoites, EES1-EES5 stages, tachyzoites, sporozoites, and cysts, as documented in prior studies[12,13,16]. Analysis parameters are those of Fig. 1f. **g**, Created with BioRender.com.

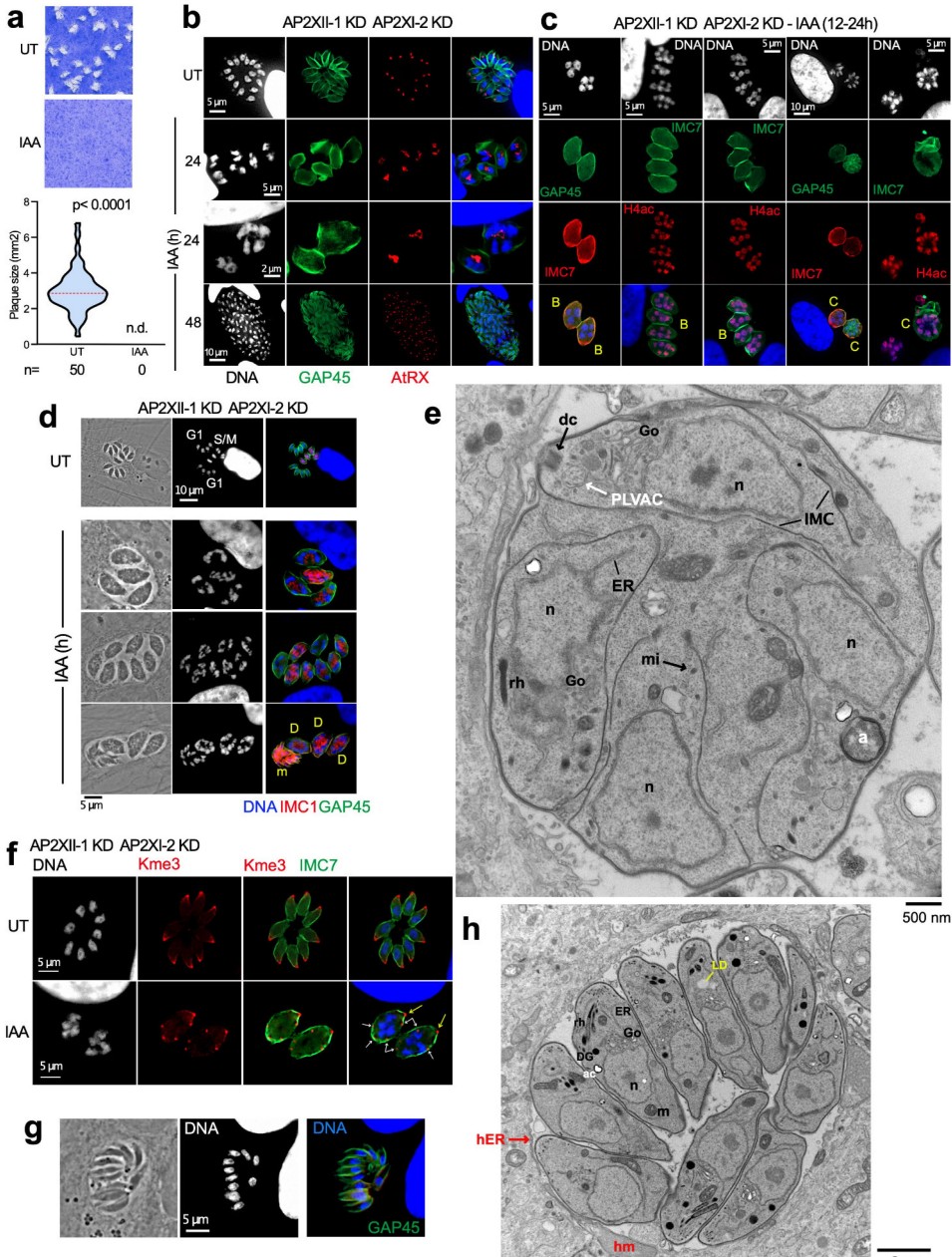

**Extended Data Fig. 5 | In vitro development of pre-gametes stages.**
**a**, Infectivity of the RH_AP2XII-1-mAID-HA/AP2XI-2-mAID-MYC strain was assessed using plaque assays, comparing untreated to IAA-treated for 7 days. Statistical significance was evaluated using Mann-Whitney. n.d. = not detected. **b-d**, IFA of tachyzoites (UT) and AP2XII-1/AP2XI-2-depleted zoites (at indicated time post-IAA) were fixed and stained with (**b**) antibodies of AtRX clone 11G8 (red) and IMC7 (green), (**c**) IMC7 or pan-acetylated histone H4 (red) and GAP45 or IMC7 (green), (**d**) IMC1 (red) and GAP45 (green). The cells were co-stained with DNA-specific Hoechst dye (white or blue). Morphotypes B, C and D meronts are highlighted in yellow. **e**, An electron micrograph of RH (AP2XII-1 KD/AP2XI-2 KD)-infected HFFs treated with IAA for 24 h shows an advanced stage of daughter individualization, characterized by polarized inner

membrane complex (IMC) and apical conoid. **f**, Untreated tachyzoites (UT) and merozoites depleted of AP2XII-1/AP2XI-2 (24 h post-IAA) were marked with H3K9me3 (red), IMC7 (green), and Hoechst DNA-specific dye. Yellow and white arrows respectively highlight mother and daughter conoids. **g**, Fully formed merozoites (24 h post-IAA) were stained with GAP45 (green) and DNA-specific Hoechst dye. **h**, An electron micrograph of IAA-treated parasites shows fully formed merozoites aligned in the PV with apex directed towards the PV membrane. Go: Golgi apparatus, rh: rhoptry, mi: microneme, n: nucleus (plus posterior that in tachyzoite), PLVAC: plant-like vacuolar compartment, dc: daughter conoid, IMC: inner membrane complex, DG: dense granule, Ac: acidocalcisome, LD: lipid droplet, ER: endoplasmic reticulum, hER: host endoplasmic reticulum, m: mitochondrion, hm: host mitochondrion.

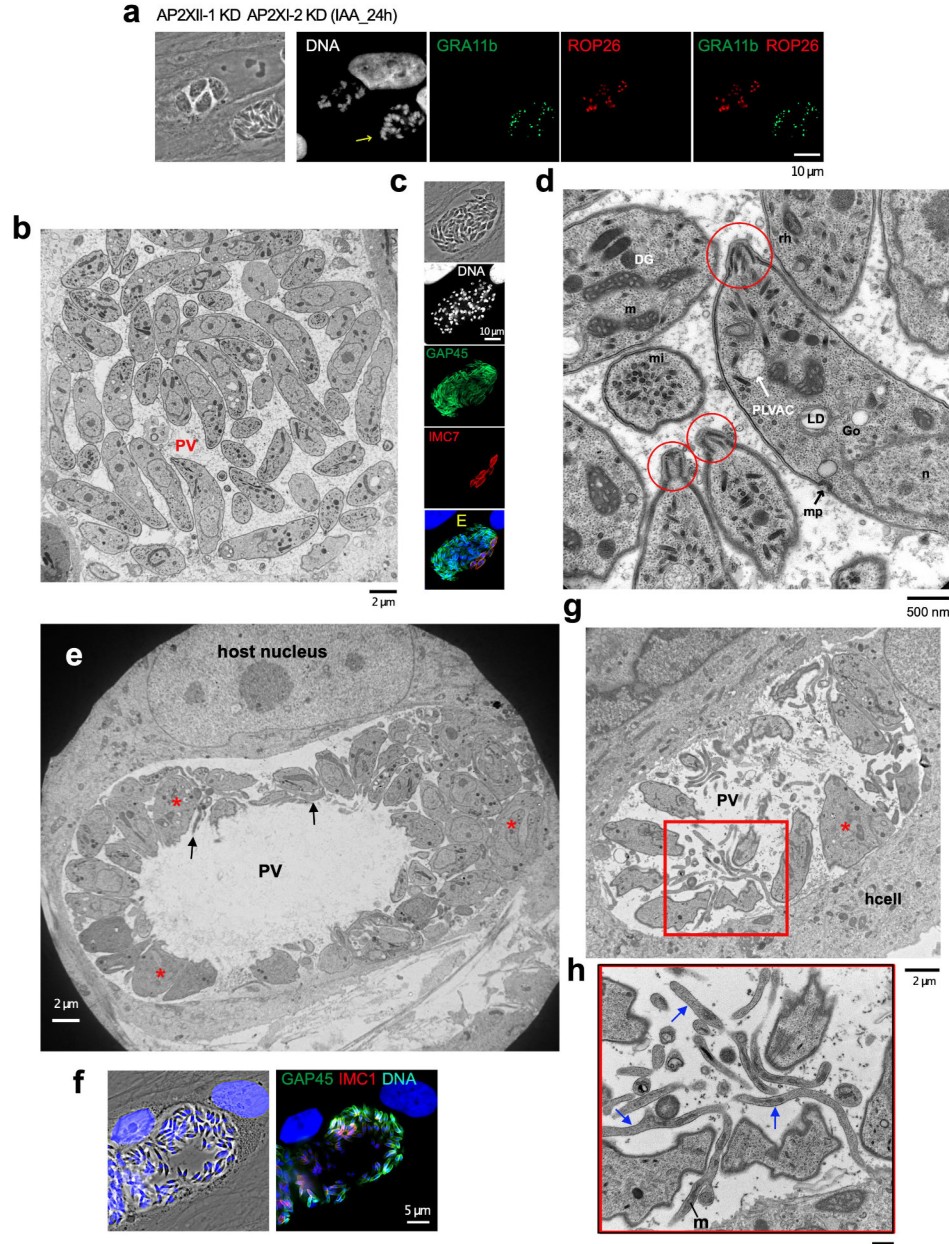

**Extended Data Fig. 6 | Mature merozoites and their special substructural features. a**, AP2XII-1/AP2XI-2-depleted meronts (24 h post-IAA) were fixed and stained with ROP26 (red), GRA11b (green) and Hoechst DNA-specific dye. Yellow arrows indicate fully developed merozoites. (**b, d, e, g, h**) Electron micrograph images of RH (AP2XII-1 KD/AP2XI-2 KD)-infected HFFs treated for 48 h with IAA. **b**, Emphasis on changes in body shape of merozoite (sausage). **c**, AP2XII-1/AP2XI-2-depleted type E meronts (48 h post-IAA) were fixed and stained with IMC7 (red), GAP45 (green) and Hoechst DNA-specific dye. **d**, Emphasis on conoid extrusion and same organelle content as in tachyzoite. n:

nucleus, Go: Golgi apparatus, rh: rhoptry, m: mitochondrion, mi: microneme, DG: dense granule, LD: lipid droplet, mp: micropore, PLVAC: plant-like vacuolar compartment. Red circles showing extruded conoid. **e**, Emphasis on two other morphological transformations of merozoites, either very large (asterisks) or tubular (arrows) in PV with large lumen. **f**, AP2XII-1/AP2XI-2-depleted type E meronts (48 h post-IAA) were fixed and stained with IMC1 (red), GAP45 (green) and Hoechst DNA-specific dye. (**g** and **h**) Emphasis on thin parasitic forms (arrows) containing mitochondria and ribosomes. m: mitochondrion.

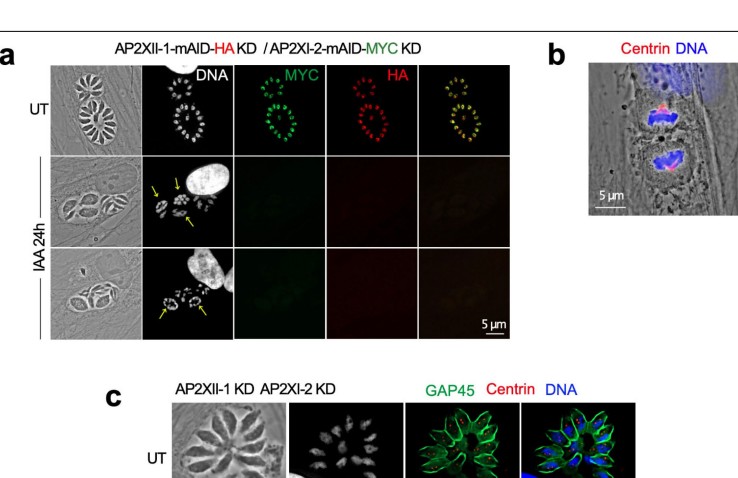

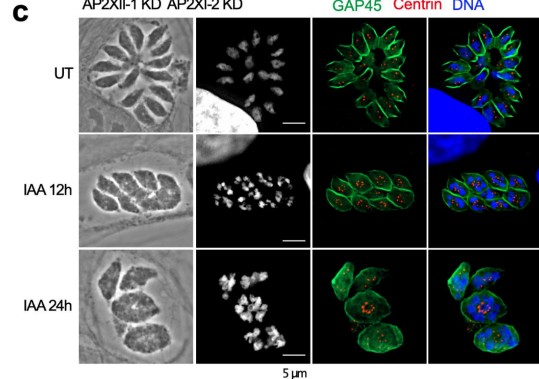

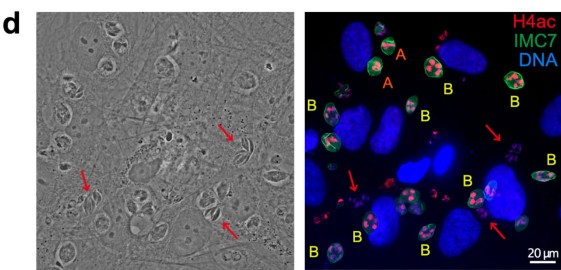

**Extended Data Fig. 7 | Microscopic analysis of endopolygeny post-AP2XII-1/ AP2XI-2 co-depletion. a**, Untreated tachyzoites (UT) and zoites depleted of AP2XII-1/AP2XI-2 (24 h post-IAA) are stained with HA (red) and MYC (green) to identify AP2XII-I-mAID-HA and AP2XI-2-mAID-MYC respectively. DNA staining was accomplished using Hoechst dye. Polyploid meronts are indicated by a yellow arrow. **b**, Centrin distribution (red) within the dividing mother fibroblast cell. **c**, Untreated tachyzoites (UT) and AP2XII-1/AP2XI-2-depleted polyploid zoites, 24 h post-IAA treatment, stained with centrin (red), GAP45 (green), and Hoechst DNA-specific dye. **d**, Detailed observation of the diversity in pre-gamete stages 16 h after simultaneous depletion of AP2XII-1 and AP2XI-2. Both zoite and host cell nuclei are marked with pan-acetylated histone H4 (red) and Hoechst DNA-specific dye (blue), whereas polyploid meronts are identified by IMC7 staining. Red arrows designate completely developed merozoites, while morphotypes A and B are highlighted in orange and red, respectively.

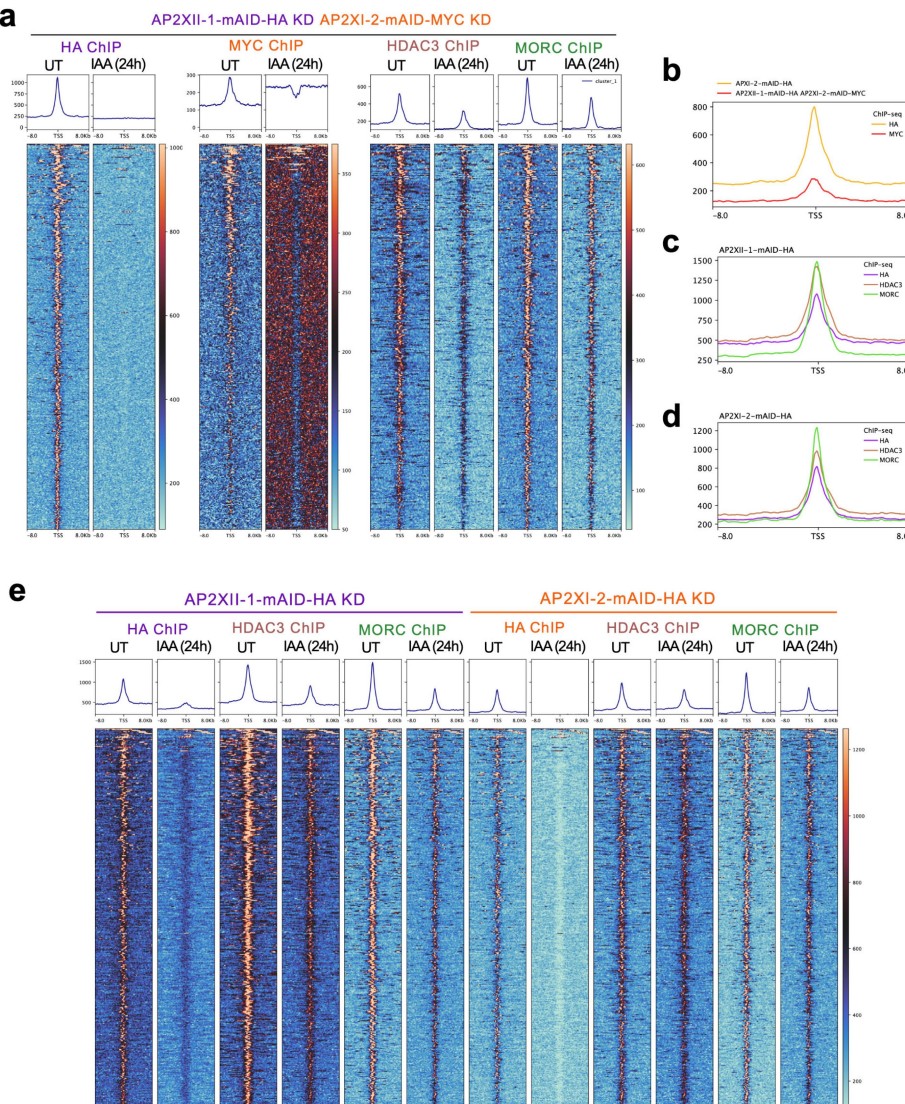

**Extended Data Fig. 8 | Contribution of AP2XII-1 and AP2XI-2 to MORC and HDAC3 recruitment on chromatin. a**, Heatmap and profile analysis of APXII-1 (HA), AP2XI-2 (MYC), HDAC3, and MORC ChIP-seq peak intensity in double KD parasites, comparing untreated (UT) to co-depleted (24 h post-IAA) conditions. Density profiles centered at TSS (± 8 kb) and heat maps of peak density are shown, with color scales indicating signal intensity. **b-d**, Comparison of chip peak profiles in cluster 2 genomic loci (as defined in Fig. 6a), centered at TSS (± 8 kb), for (**b**) AP2XI-2 (HA or MYC antibody) in single AP2XI-2 KD or double

knockdown in absence of auxin treatment; (**c**) AP2XII-1 (HA), HDAC3, and MORC in AP2XII-1 KD; (**d**) AP2XI-2 (HA), HDAC3, and MORC in AP2XI-2 KD. **e**, Heatmap and profile analysis of APXII-1 (HA), AP2XI-2 (MYC), HDAC3, and MORC ChIP-seq peak intensity in single KD parasites, comparing untreated (UT) to depleted (24 h post-IAA) conditions. Average signal profiles centered at TSS (± 8 kb) and heat maps of peak density are shown, with color scales indicating signal intensity. For all panels, Deeptools analysis used summed occupancies within a bin size of 10.

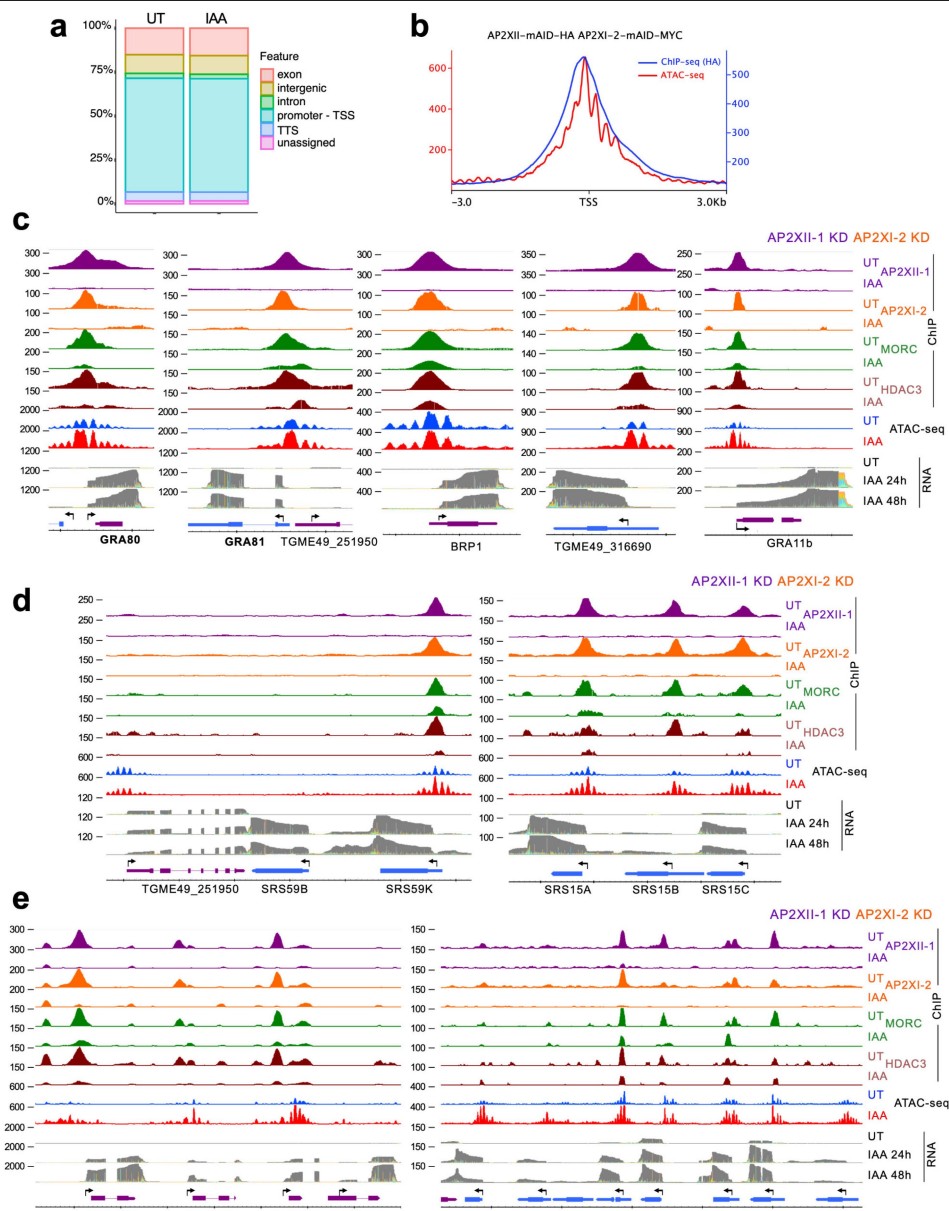

**Extended Data Fig. 9 | Representative merozoite genes and their regulation by AP2XII-1 and AP2XI-2 and their repressive partners MORC and HDAC3.** **a**, HOMER analysis reveals global distribution of significant Tn5 accessibility peaks across all genomic features in the AP2XII-1/AP2XI-1 double knockdown strain. Similar profiles were observed between untreated and treated conditions. **b**, Comparison profile of ChIP-seq summed occupancies over a bin size of 10 for AP2XI-2 (HA) and Tn5 accessibility density at all gene loci centered at TSS (± 3 kb) in the AP2XII-1/AP2XI-2 double knockdown strain without auxin treatment. **(c-e)**, IGB screenshots illustrate representative genomic regions containing merozoite genes, displaying ChIP-seq signal occupancy, ATAC-seq chromatin accessibility profiles, and nanopore DRS data, similar to Fig. 6i.

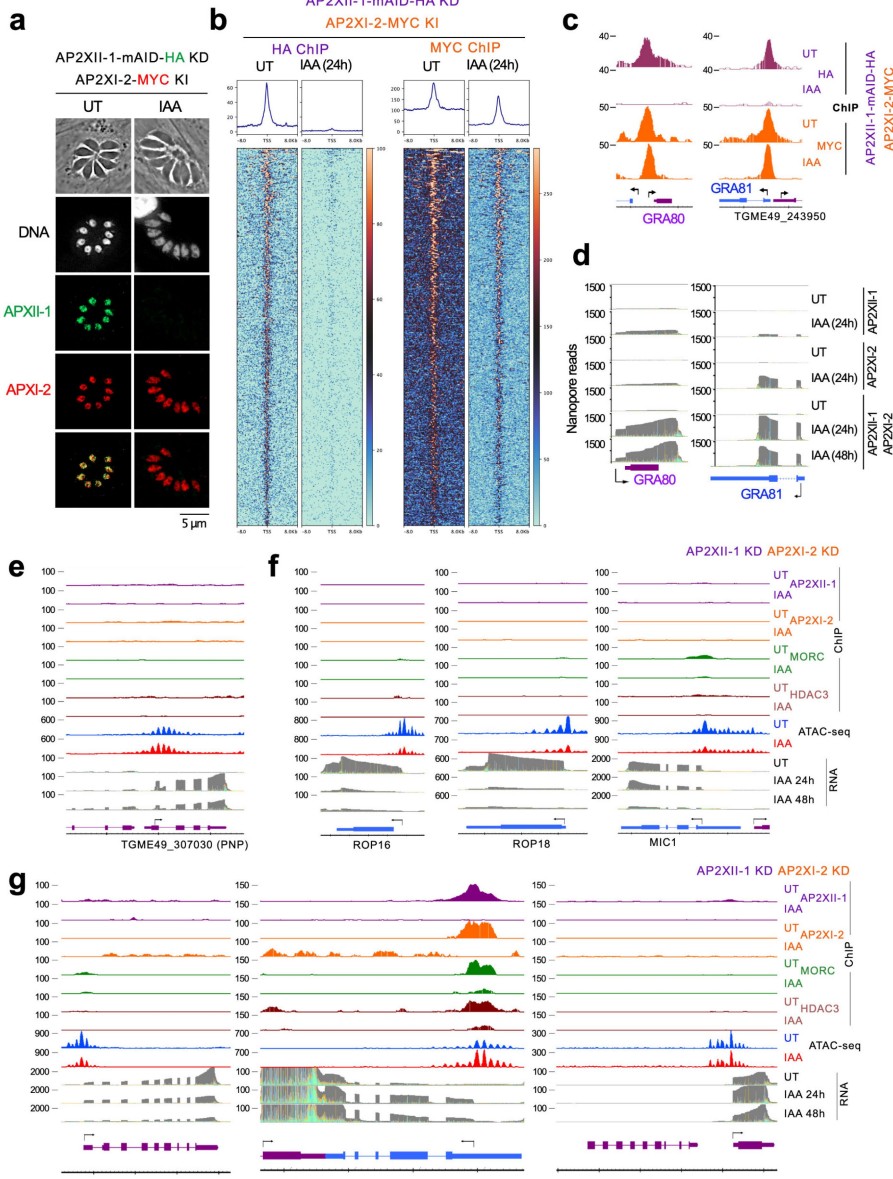

**Extended Data Fig. 10 | AP2XI-2 can bind to chromatin even in the absence of AP2XII-1. a**, AP2XII-1 (HA) and AP2XI-2 (MYC) expression levels and subcellular localization were assessed using IFA following AP2XII-1 depletion. Hoechst DNA-specific dye was used for counterstaining. **b**, Heatmap and profile analysis of APXII-1 (HA) and AP2XI-2 (MYC) ChIP-seq mean occupancy over a bin size of 10 in APXII-1 KD parasites with AP2XI-2-MYC knock-in (KI), comparing untreated (UT) and co-depleted (24 h post-IAA) conditions. Average signal profiles centered at TSS ( ± 8 kb) and heat maps of peak density display signal intensity. **c**, IGB screenshots depict the enrichment of AP2XII-1 and AP2XI-2 at the GRA80 and GRA81 loci in the context of AP2XII-1 single knockdown. **d**, M-pileup representation of aligned Nanopore DRS reads at genes up-regulated following IAA-induced knockdown of AP2XII-1 and AP2XI-2 individually or in combination. **e**, IGB screenshot highlighting the genomic region of the merozoite-specific purine nucleoside phosphorylase (PNP) gene, which is up-regulated upon IAA treatment independently of MORC and HDAC3. **f**, IGB screenshots displaying the genomic regions of the rhoptry genes (ROP16, ROP18) and the microneme gene (MIC1), demonstrating their repression in IAA-treated parasites. **g**, IGB screenshots showcasing the AMA1 gene family. (**e-g**) ChIP-seq signal occupancy, ATAC-seq chromatin accessibility profiles, and nanopore DRS are visualized in a manner consistent with Fig. 6i.

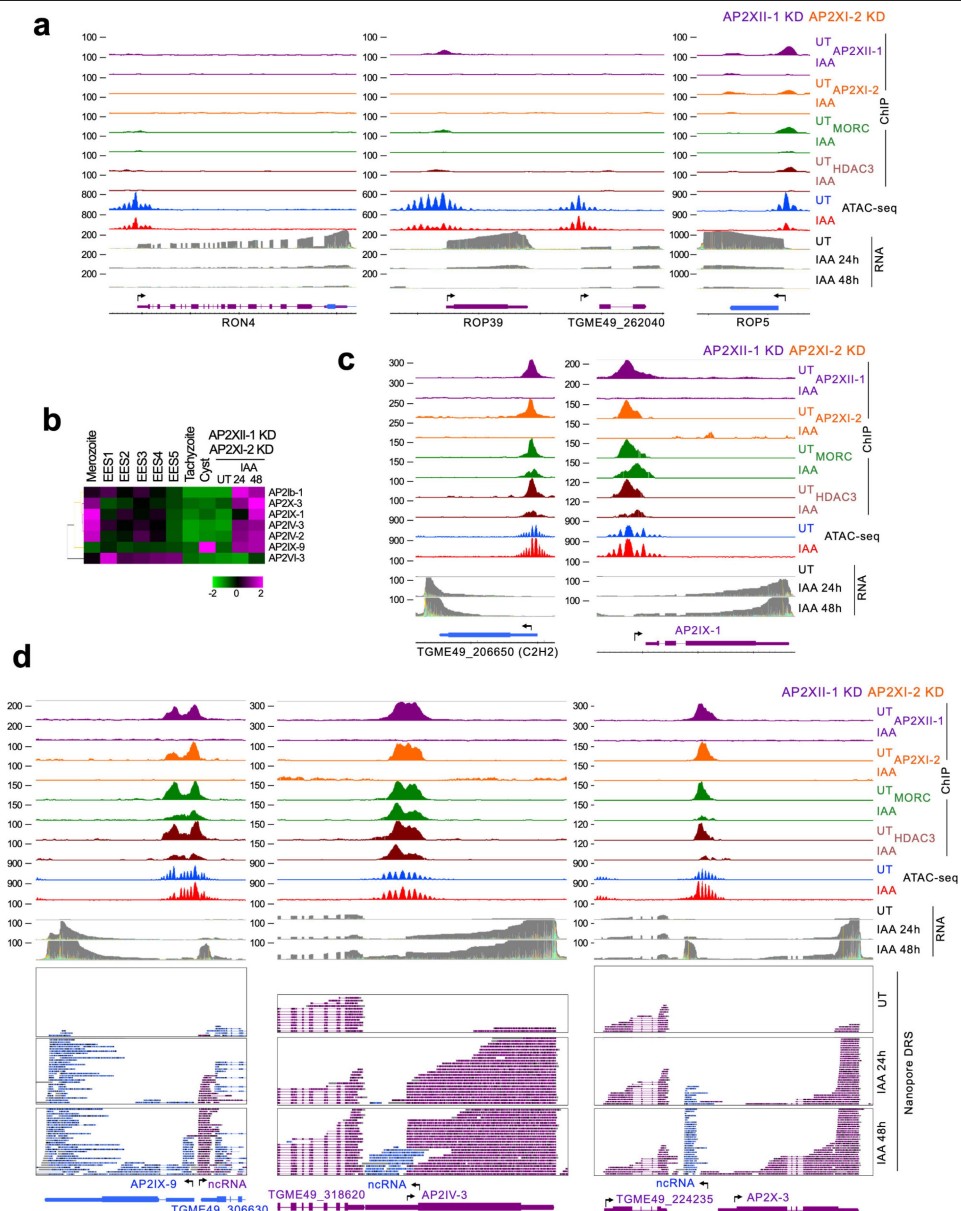

**Extended Data Fig. 11 | AP2XII-1 and AP2XI-2 co-depletion induces a downstream network of secondary transcription factors to guide merogony. a**, IGB screenshots displaying the genomic regions of the rhoptry genes (RON4, ROP39, ROP5) repressed in IAA-treated parasites in an AP2XII-1/AP2XI-2-independent manner. ChIP-seq signal occupancy, ATAC-seq chromatin accessibility profiles, and nanopore DRS are visualized in a manner consistent with Fig. 6i. **b**, Heat map showing hierarchical mRNA clustering analysis (Pearson correlation) of AP2 TFs regulated by simultaneous depletion of

AP2XII-1 and AP2XI-2. Shown is the abundance of their transcripts at different developmental stages, namely merozoites, EES, tachyzoites, and cysts. The color scale indicates the log2-transformed fold changes. **c-d**, IGB screenshots of genomic regions with secondary transcription factors, including C2H2 Zinc Finger and AP2s, exhibiting activated expression in IAA-treated parasites. Displayed are ChIP-seq signal occupancy, ATAC-seq chromatin accessibility profiles, and nanopore DRS data, following the same representation as in Fig. 6i.

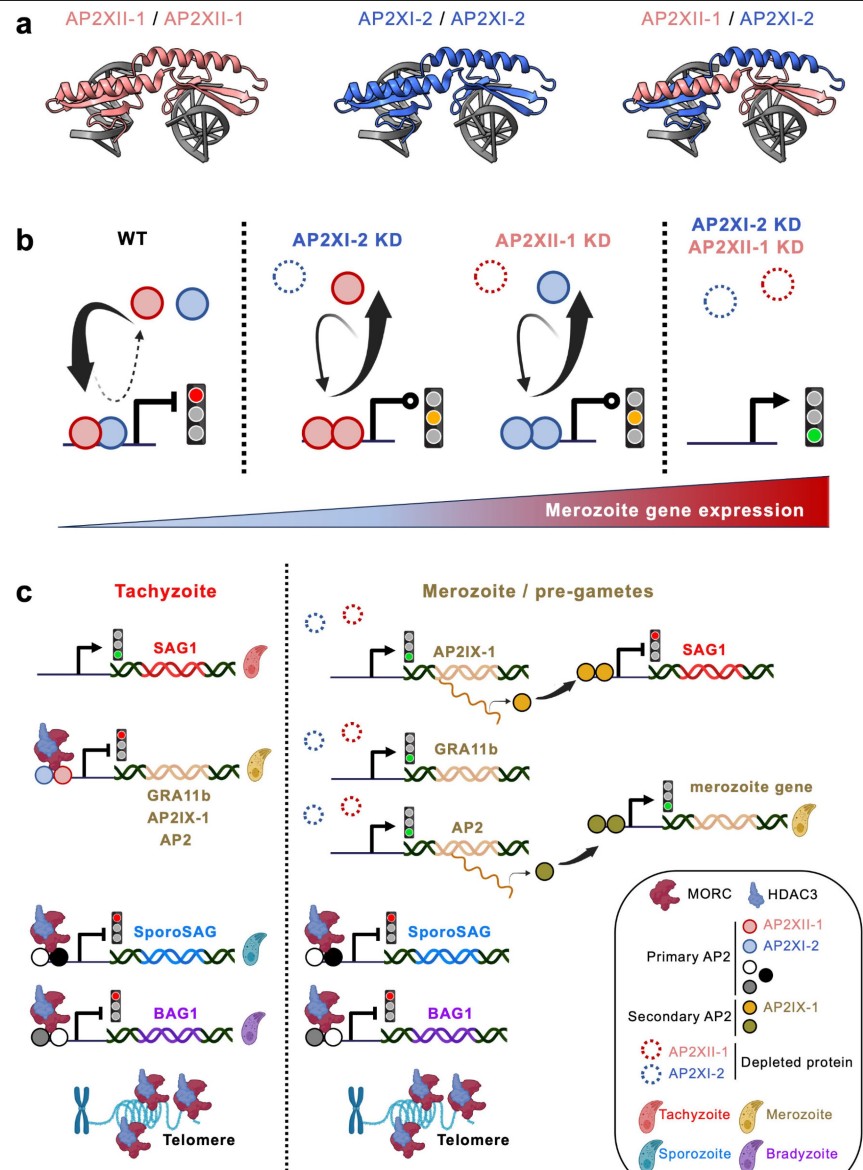

**Extended Data Fig. 12 | Modeling of the *modus operandi* of AP2XII-1/
AP2XI-2. a**, Inspired by a 2.2 A crystal structure (pdb: 3IGM), the model shows
that AP2XII-1 and AP2XI-2 can interact with DNA through either homodimeric
or heterodimeric forms. These configurations influence DNA looping by
stabilizing or forming domain-swapped dimers. **b**, AP2XI-2 and AP2XII-1
together fully silence merozoite gene expression (red light). A single KD only
partially represses gene expression (orange light). When both are knocked
down, genes are fully expressed (green light). Even with one AP2 missing,
the remaining one can form a homodimer to partially repress transcription
(Extended Data Fig. 10c, d). We hypothesize that each AP2 forms a fragile but
stable homodimer, allowing mild gene expression due to easy disassociation
from DNA. In contrast, heterodimers bind strongly to DNA, resulting in
complete gene silencing. **c**, MORC/HDAC3 forms multiple partnerships
with primary AP2, keeping chronic and sexual-stage genes in a persistently
repressed chromatin state. The repressor complex drive the hierarchical
expression of secondary AP2s, which may enforce the unidirectionality of the

life cycle by influencing the gene expression of their respective stages. In
tachyzoites, genes specific to this stage (*SAG1*) are expressed, while those
related to merozoites (*GRA11b*) are repressed by AP2XII-1/AP2XI-2. Other
primary AP2s help in recruiting MORC to other stage-specific genes, such as
those for sporozoites and bradyzoites, to silence their expression. Additionally,
MORC has a non-genic function where it binds to telomeric repeats, possibly
facilitating silencing at the chromosome ends (Extended Data Fig. 1).
Co-depletion of AP2XII-1/AP2XI-2 trigger merozoite gene activation, speeding
up merogony. Secondary AP2s like AP2IX-1 then emerge, repressing tachyzoite-
specific genes like SAG1. This second wave of TFs could also act as activators
for pre-sexual genes not directly controlled by the primary AP2s (e.g., *PNP*;
Extended Data Fig. 10e), offering an alternative route to merozoite development.
The MORC/HDAC3-containing complexes maintain stage-specific and telomeric
gene silencing, demonstrating their selective roles. **a–c**, Created with
BioRender.com.

## Reporting Summary

## Statistics

For all statistical analyses, confirm that the following items are present in the figure legend, table legend, main text, or Methods section.

| n/a | Confirmed | |
|---|---|---|
| ☐ | ☒ | The exact sample size (*n*) for each experimental group/condition, given as a discrete number and unit of measurement |
| ☒ | ☐ | A statement on whether measurements were taken from distinct samples or whether the same sample was measured repeatedly |
| ☐ | ☒ | The statistical test(s) used AND whether they are one- or two-sided<br>*Only common tests should be described solely by name; describe more complex techniques in the Methods section.* |
| ☒ | ☐ | A description of all covariates tested |
| ☒ | ☐ | A description of any assumptions or corrections, such as tests of normality and adjustment for multiple comparisons |
| ☐ | ☒ | A full description of the statistical parameters including central tendency (e.g. means) or other basic estimates (e.g. regression coefficient) AND variation (e.g. standard deviation) or associated estimates of uncertainty (e.g. confidence intervals) |
| ☐ | ☒ | For null hypothesis testing, the test statistic (e.g. *F*, *t*, *r*) with confidence intervals, effect sizes, degrees of freedom and *P* value noted<br>*Give P values as exact values whenever suitable.* |
| ☒ | ☐ | For Bayesian analysis, information on the choice of priors and Markov chain Monte Carlo settings |
| ☒ | ☐ | For hierarchical and complex designs, identification of the appropriate level for tests and full reporting of outcomes |
| ☐ | ☒ | Estimates of effect sizes (e.g. Cohen's *d*, Pearson's *r*), indicating how they were calculated |

*Our web collection on statistics for biologists contains articles on many of the points above.*

## Software and code

Policy information about availability of computer code

| | |
|---|---|
| Data collection | Librairies were sequenced on an Illumina NovaSeq platform and collected in FastQ format. MS and MS/MS data were acquired using Xcalibur software version 4.0 (Thermo Scientific). Peptides and proteins were identified using Mascot (version 2.6). |
| Data analysis | Integrated Genome Browser - by Nowlan H. Freese, David C. Norris, and Ann E. Loraine, Bioinformatics 2016 Jul 15;32(14):2089-95.<br>GraphPad - Prism 7<br>R (version 3.5.3), R Development Core Team, 2012<br>blastp (version 2.2.31+), Altschul et al., 1990<br>BOWTIE (V2.1.0)<br>Samtools v1.4<br>Bamtools v2.5.1<br>Guppy v5 or higher<br>minimap2 v2.17<br>Deeptools v3.5.1 was used for global peak intensity over gene features.<br>Basecaller software (OLB V1.8)<br>SeqMan NGen (version 14, DNASTAR. Madison, WI, USA)<br>Lasergene Genomics Suite version 14 (DNASTAR, Madison, WI, USA)<br>ArrayStar module (version 14, DNASTAR. Madison, WI, USA)<br>ClustalW software<br>RAW files were processed using MaxQuant version 1.6.2.10.<br>Proline software (http://proline.profiproteomics.fr) was used to filter the results.<br>Statistical analyses were performed using ProStaR (Wieczorek, S. et al. DAPAR & ProStaR: software to perform statistical analyses in |

quantitative discovery proteomics. Bioinformatics 33, 135–136 (2017).
nf-core ATAC-SEQ v2.0 is a nextflow pipeline for ATAQ-SEQ data processing
Integrated Genome Browser (IGB) v9.1.8 or higher

For manuscripts utilizing custom algorithms or software that are central to the research but not yet described in published literature, software must be made available to editors and reviewers. We strongly encourage code deposition in a community repository (e.g. GitHub). See the Nature Portfolio guidelines for submitting code & software for further information.

## Data

Policy information about availability of data

All manuscripts must include a data availability statement. This statement should provide the following information, where applicable:
- Accession codes, unique identifiers, or web links for publicly available datasets
- A description of any restrictions on data availability
- For clinical datasets or third party data, please ensure that the statement adheres to our policy

The ChIP-seq data have been deposited to the GEO Dataset :
https://www.ncbi.nlm.nih.gov/geo/query/acc.cgi?acc=GSE222819

The ATAC-seq data have been deposited to the GEO Dataset :
https://www.ncbi.nlm.nih.gov/geo/query/acc.cgi?acc=GSE222832

The Nanopore and Illumina RNAseq data have been deposited to the GEO Datasets under the BioProject: PRJNA921935
https://www.ncbi.nlm.nih.gov/bioproject/PRJNA921935

MS-base proteomic data have been deposited to the ProteomeXchange Consortium via the PRIDE partner repository with the dataset identifiers PXD039400 and PXD042658 for respectively proteome-wide and interactomic analyses.
https://www.ebi.ac.uk/pride/archive/projects/PXD039400
https://www.ebi.ac.uk/pride/archive/projects/PXD042658

## Human research participants

Policy information about studies involving human research participants and Sex and Gender in Research.

| | |
|---|---|
| Reporting on sex and gender | N/A |
| Population characteristics | N/A |
| Recruitment | N/A |
| Ethics oversight | N/A |

Note that full information on the approval of the study protocol must also be provided in the manuscript.

# Field-specific reporting

Please select the one below that is the best fit for your research. If you are not sure, read the appropriate sections before making your selection.

☒ Life sciences          ☐ Behavioural & social sciences          ☐ Ecological, evolutionary & environmental sciences

For a reference copy of the document with all sections, see nature.com/documents/nr-reporting-summary-flat.pdf

# Life sciences study design

All studies must disclose on these points even when the disclosure is negative.

| | |
|---|---|
| Sample size | Sample sizes were determined because differences among groups were consistent and significant.<br>All experiments were performed in biological replicates to allow for statistical analyses. |
| Data exclusions | No data were excluded from the analysis |
| Replication | All findings were reproduced successfully, and replications are described in methods and figure legends in more detail. |
| Randomization | No method of randomization was used and all experiments were performed in independent biological replicates as stated for each experiment in the main manuscript. All corresponding treatment and mock samples were processed at the same time to minimize technical variation. Random images of cells were evaluated for IFA and TEM. Randomization for other experiments were not subjective. |

| | |
|---|---|
| Blinding | Investigators were not blinded during the experiments. Experiments were performed in biological replicates and provided consistent statistically relevant results. Other experiments were not performed with blinding since no subjective evaluations were involved. |

# Reporting for specific materials, systems and methods

We require information from authors about some types of materials, experimental systems and methods used in many studies. Here, indicate whether each material, system or method listed is relevant to your study. If you are not sure if a list item applies to your research, read the appropriate section before selecting a response.

## Materials & experimental systems

| n/a | Involved in the study |
|---|---|
| ☐ | ☒ Antibodies |
| ☒ | ☐ Eukaryotic cell lines |
| ☒ | ☐ Palaeontology and archaeology |
| ☐ | ☒ Animals and other organisms |
| ☒ | ☐ Clinical data |
| ☒ | ☐ Dual use research of concern |

## Methods

| n/a | Involved in the study |
|---|---|
| ☐ | ☒ ChIP-seq |
| ☒ | ☐ Flow cytometry |
| ☒ | ☐ MRI-based neuroimaging |

## Antibodies

| | |
|---|---|
| Antibodies used | The following primary antibodies were used in the immunofluorescence, immunoblotting, and/or ChIP assays: rabbit anti-TgHDAC3 (RRID: AB_2713903), rabbit anti-TgGAP45 (gift from Pr. Dominique Soldati), mouse anti-HA tag (Roche, RRID: AB_2314622), rabbit anti-HA Tag (Cell Signaling Technology, RRID: AB_1549585), rabbit anti-mCherry (Cell Signaling Technology, RRID: AB_2799246), rabbit anti-FLAG (Cell Signaling Technology, RRID: AB_2798687), mouse anti-MYC clone 9B11 (RRID: AB_2148465), H3K9me3 (Diagenode, RRID: AB_2616044), rabbit Anti-acetyl-Histone H4, pan (Lys 5,8,12) (Millipore, RRID:AB_310270),  rat anti-IMC7 (gift from Pr. Gubbels MJ),  mouse anti-IMC1 (gift from Pr. Ward GE), mouse anti-AtRx antibody clone 11G826, mouse anti-GRA11b14. We have also raised homemade antibodies against linear peptides in rabbits corresponding to the following proteins: MORC_Peptide2 (C +SGAPIWTGERGSGA); AP2XI-2 (C+HAFKTRRTEAAT) TGME49_273980/GRA80  (C+RPPWAPGAGPEN); TGME49_243940/GRA81 (C +QKELAEVAQRALEN); TGME49_277230/GRA82 (C+SDVNTEGDATVANPE); TGME49_209985/ROP26 (CQETVQGNGETQL); SRS48 family (CKALIEVKGVPK); SRS59B/K (C+IHVPGTDSTSSGPGS); TGME49_314250/BRP1 (C+QVKEGTKNNKGLSDK); TGME49_307640/CK2 kinase (C+IRAQYHAYKGKYSHA); and TGME49_306455 (C+DGRTPVDRVFEE). They were manufactured by Eurogentec and used for immunofluorescence, immunoblotting and/or chromatin immunoprecipitation. Secondary immunofluorescent antibodies were coupled with Alexa Fluor 488 or Alexa Fluor 594 (Thermo Fisher Scientific). Secondary antibodies used in Western blotting were conjugated to alkaline phosphatase (Promega) or horseradish peroxidase. |
| Validation | All the antibodies used in this study were validated for ChIP-seq, IFA or Western blot by several groups according to their RRID numbers and sometimes by the company that own them. GAP45, IMC7, IMC1, AtRx(11G8) and GRA11b are well known antibody in Toxoplasma gondii and were validated by several publications. |

## Animals and other research organisms

Policy information about studies involving animals; ARRIVE guidelines recommended for reporting animal research, and Sex and Gender in Research

| | |
|---|---|
| Laboratory animals | Six-week-old NMRI, CD1 or Balb/C mice were obtained from Janvier Laboratories (Le Genest-Saint-Isle, France) |
| Wild animals | No wild animals were used in this study |
| Reporting on sex | Female mice were used for all studies. |
| Field-collected samples | No field-collected samples were used in this study |
| Ethics oversight | Mouse care and experimental procedures were performed under pathogen-free conditions in accordance with established institutional guidance and approved protocols from the Institutional Animal Care and Use Committee of the University Grenoble Alpes (APAFIS#4536-2016031 017075121 v5). |

Note that full information on the approval of the study protocol must also be provided in the manuscript.

## ChIP-seq

### Data deposition

☒ Confirm that both raw and final processed data have been deposited in a public database such as GEO.

☒ Confirm that you have deposited or provided access to graph files (e.g. BED files) for the called peaks.

| Data access links | The ChIP-seq data have been deposited to the GEO Dataset : |
| :--- | :--- |
| *May remain private before publication.* | Series GSE222819 |
| | To review GEO accession GSE222819: |
| | Go to https://www.ncbi.nlm.nih.gov/geo/query/acc.cgi?acc=GSE222819 |
| | Enter token wjefcuwirhybtqh into the box |

| Files in database submission | GSM6932717  AP2-XII-1-KD-HA_HA-antibody_UT |
| :--- | :--- |
| | GSM6932718  AP2-XII-1-KD-HA_HA-antibody_IAA24h |
| | GSM6932719  AP2-XII-1-KD-HA_HDAC3-antibody_UT |
| | GSM6932720  AP2-XII-1-KD-HA_HDAC3-antibody_IAA24h |
| | GSM6932721  AP2-XII-1-KD-HA_MORC-antibody_UT |
| | GSM6932722  AP2-XII-1-KD-HA_MORC-antibody_IAA24h |
| | GSM6932723  AP2-XI-2-KD-HA_HA-antibody_UT |
| | GSM6932724  AP2-XI-2-KD-HA_HA-antibody_IAA24h |
| | GSM6932725  AP2-XI-2-KD-HA_HDAC3-antibody_UT |
| | GSM6932726  AP2-XI-2-KD-HA_HDAC3-antibody_IAA24h |
| | GSM6932727  AP2-XI-2-KD-HA_MORC-antibody_UT |
| | GSM6932728  AP2-XI-2-KD-HA_MORC-antibody_IAA24h |
| | GSM6932729  AP2-XII-1-KD-HA_AP2-XI-2-KD-Myc_HA-antibody_UT |
| | GSM6932730  AP2-XII-1-KD-HA_AP2-XI-2-KD-Myc_HA-antibody_IAA24h |
| | GSM6932731  AP2-XII-1-KD-HA_AP2-XI-2-KD-Myc_Myc-antibody_UT |
| | GSM6932732  AP2-XII-1-KD-HA_AP2-XI-2-KD-Myc_Myc-antibody_IAA24h |
| | GSM6932733  AP2-XII-1-KD-HA_AP2-XI-2-KD-Myc_HDAC3-antibody_UT |
| | GSM6932734  AP2-XII-1-KD-HA_AP2-XI-2-KD-Myc_HDAC3-antibody_IAA24h |
| | GSM6932735  AP2-XII-1-KD-HA_AP2-XI-2-KD-Myc_MORC-antibody_UT |
| | GSM6932736  AP2-XII-1-KD-HA_AP2-XI-2-KD-Myc_MORC-antibody_IAA24h |

| Genome browser session | Not applicable |
| :--- | :--- |
| (e.g. UCSC) | |

## Methodology

| Replicates | MORC, HDAC3 or HA : 3 different Chips in different parental strains |
| :--- | :--- |

| Sequencing depth | Sequencing layout: 1x150bp |
| :--- | :--- |
| | Sequencing Depth for each sample ID |
| | ID: total number of reads (Pass solexa CHASTITY quality filter)/uniquely mapped (ToxoDB-13.0_TgondiiME49) |
| | GSM6932717  AP2-XII-1-KD-HA_HA-antibody_UT (47,375,300/16,108,744 reads) |
| | GSM6932718  AP2-XII-1-KD-HA_HA-antibody_IAA24h (33,607,612/11,284,027 reads) |
| | GSM6932719  AP2-XII-1-KD-HA_HDAC3-antibody_UT (41,375,218/17,389,398 reads) |
| | GSM6932720  AP2-XII-1-KD-HA_HDAC3-antibody_IAA24h (40,312,162/14,737,507 reads) |
| | GSM6932721  AP2-XII-1-KD-HA_MORC-antibody_UT (37,349,606/10,598,146 reads) |
| | GSM6932722  AP2-XII-1-KD-HA_MORC-antibody_IAA24h (34,730,526/9,808,916 reads) |
| | GSM6932723  AP2-XI-2-KD-HA_HA-antibody_UT (38,347,182/8,439,251 reads) |
| | GSM6932724  AP2-XI-2-KD-HA_HA-antibody_IAA24h (30,766,515/6,807,339 reads) |
| | GSM6932725  AP2-XI-2-KD-HA_HDAC3-antibody_UT (37,024,267/10,518,299 reads) |
| | GSM6932726  AP2-XI-2-KD-HA_HDAC3-antibody_IAA24h (35,824,007/11,383,677 reads) |
| | GSM6932727  AP2-XI-2-KD-HA_MORC-antibody_UT (32,639,068/7,847,338 reads) |
| | GSM6932728  AP2-XI-2-KD-HA_MORC-antibody_IAA24h (34,919,949/9,531,074 reads) |
| | GSM6932729  AP2-XII-1-KD-HA_AP2-XI-2-KD-Myc_HA-antibody_UT (26,209,227/6,862,246 reads) |
| | GSM6932730  AP2-XII-1-KD-HA_AP2-XI-2-KD-Myc_HA-antibody_IAA24h (26,015,064/5,812,766 reads) |
| | GSM6932731  AP2-XII-1-KD-HA_AP2-XI-2-KD-Myc_Myc-antibody_UT (17,571,565/3,432,148 reads) |
| | GSM6932732  AP2-XII-1-KD-HA_AP2-XI-2-KD-Myc_Myc-antibody_IAA24h (28,627,661/6,737,545 reads) |
| | GSM6932733  AP2-XII-1-KD-HA_AP2-XI-2-KD-Myc_HDAC3-antibody_UT (19,248,893/4,861,975 reads) |
| | GSM6932734  AP2-XII-1-KD-HA_AP2-XI-2-KD-Myc_HDAC3-antibody_IAA24h (23,609,829/3,008,898 reads) |
| | GSM6932735  AP2-XII-1-KD-HA_AP2-XI-2-KD-Myc_MORC-antibody_UT (20,981,524/4,634,579 reads) |
| | GSM6932736  AP2-XII-1-KD-HA_AP2-XI-2-KD-Myc_MORC-antibody_IAA24h (19,924,121/3,299,573 reads) |

| Antibodies | rabbit anti-TgHDAC3 (Bougdour et al., 2009; RRID: AB_2713903) |
| :--- | :--- |
| | rabbit anti-HA Tag (Cell Signaling Technology, RRID: AB_1549585) |
| | mouse anti-MYC clone 9B11 (RRID: AB_2148465) |
| | rabbit MORC_Peptide2 (C+SGAPIWTGERGSGA) |

| Peak calling parameters | After the sequencing platform generated the sequencing images, the stages of image analysis and base calling were performed using Off-Line Basecaller software (OLB V1.8). After passing Solexa CHASTITY quality filter, the clean reads were aligned to T. gondii reference genome (TGME49) using BOWTIE V2 then converted and sorted using Bamtools V2.5. Aligned reads were used for peak calling of the ChIP enriched peaks using MACS V2.2 with a cutoff p-value of 10-4. Data visualization: For IGB visualization and gene centered analysis using Deeptools, MACS2 generated bedgraph files were processed with the following command: "sort -k1,1 -k2,2n 5_treat_pileup.bdg > 5_treat_pileup-sorted.bdg" then converted using the BedGraphToBigWig program (ENCODE project). The Deeptools analysis were generated using "computeMatrix reference point" with the following parameters (--minThreshold 2, --binSize 10 and --averageTypeBins sum). Plotprofile or heatmap was then used with a k-mean clustering when applicable. Inter sample comparison were obtained using the nf-core  chip-seq V2.0.0 workflow with standard parameters. From this pipeline, HOMER |
| :--- | :--- |

(annotatePeaks) was used to analyze peak distribution relative to gene features. All these raw and processed files can be found at Series GSE222819.

Data quality

Raw_data: Contains all reads which pass Solexa CHASTITY quality filter in FASTQ format (*_sequence.fastq files).

Software

BOWTIE2 software (V2.1.0)
MACS v2.2 (Model-based Analysis of ChIP-seq) software was used to detect the peak from ChIP-seq data.
Deeptools v3.5.1 was used for global peak intensity over gene features.
nf-core Chip-Seq v2.0.0 is a nextflow pipeline for Chip-Seq analysis.
Samtools v1.4
Bamtools v2.5.1
ucsc-bedgraphtobigwig v377
Integrated Genome Browser (IGB) v9.1.8 or higher

