## [Peer Review File · Nature]

Manuscript Title: In vitro production of cat-restricted *Toxoplasma* pre-sexual stages

Reviewer Comments & Author Rebuttals

Reviewer Reports on the Initial Version:

Referees' comments:

Referee #1 (Remarks to the Author):

summary of key results: This is a very exciting paper demonstrating that the simultaneous knockdown of two *T. gondii* transcription factors results in a rewiring of the *T. gondii* developmental program in vitro, producing what are very clearly pre-sexual stages (meronts/merozoites). The rationale for the study and identification of these specific TFs is strong and the results are very clear and convincing. using this system the authors are able to increase our understanding of presexual development in *T. gondii* without the use of cats, an important step for the field. Molecular and epigenetic data are also provided demonstrating how these two TFs work together to silence the sexual developmental program. Overall this is an exciting development for the field as these life stages are inaccessible to most and nearly impossible to obtain in the numbers shown here.

originality and significance: One thing that dampens enthusiasm a bit is that prior work (cited below) using similar approaches knocking down MORC led to similar changes in the transcriptome and switching of parasites towards a more merozoite-like state. Including some demonstration signs of sexual development (which is lacking in the AP2KD parasites here) may be occurring. In that study the authors also claimed that that the discovery would permit access to sexual stages. From the present study it is clear that the double AP2 KD presented is now the most robust way to induce progress towards sexual development in vitro that is known. however the current study represents a further refinement of the genetic regulators of MORC and HDAC3 binding rather than uncovering the core pathway required for entrance into the sexual or cat stages of development.

Farhat, D.C., Swale, C., Dard, C. et al. A MORC-driven transcriptional switch controls *Toxoplasma* developmental trajectories and sexual commitment. *Nat Microbiol* 5, 570–583 (2020).

<https://doi.org/10.1038/s41564-020-0674-4>

One broad comment is that I cannot find reference to the frequency of the observed changes in morphology among the population of parasites treated with IAA. While it is mentioned and very clear that there is a lot of asynchrony in these parasites, are all parasites undergoing some form of commitment to presexual stages? or are some maintaining their tachyzoite morphology? It seems like a critical component of the manuscript to assess the extent of the penetrance of the developmental phenotype and whether one particular form might be favored at a given time post-IAA treatment.

One question is how to retrieve the meronts intact from infected host cells intact as they are larger than parasites. the method for how this was achieved was not detailed in the methods that I could

find. Is this by needle passage or some other method? this is an important detail and it should be demonstrated if the meronts are actually in tact after extraction from the host cells or if they have been lysed themselves (perhaps their nuclei stay intact and are ultimately the substrate for the chip seq?). This is not just a trivial methods question but would directly inform the data obtained throughout the manuscript from the different life stages and time points. Since merozoites have not been generated with such efficiency in any previous work, this is a new problem that we, as a field are very happy to have! but should be clarified explicitly in the manuscript nonetheless.

Finally, the manuscript suffers from many non-quantitative statements that should be qualified using statistical or other quantitative data. I have detailed a number of these below but they are found throughout the manuscript in the results section. I share the excitement of the authors about this discovery and I can appreciate their desire to convey their own excitement! However these statements are not necessary to drive the impact home. the data speak well to the fact that this genetic manipulation drives *T. gondii* towards presexual stages.

More specific comments are found here:

In the array heatmaps the replicates are reduced to single values. Is there any reason for this? i prefer showing all the data if possible.

Significance values are lacking from much of the manuscript for all of the proteomics and transcriptomics data. How are the authors deciding that a given transcript/protein is of different abundance. references to DEseq2 for example are made but no significance thresholds are given. this should be in the figure legends and methods text if possible.

for the expression heat maps I presume that the data were mean centered prior to clustering? It would be helpful to indicate this in the legend whereas what is there now is the phrase "data are shown as log2 fold changes" but I think that is incorrect or at least missing the important detail of what the change is compared to.

if *gra81* is different from *gra80*, can you please indicate its gene model number at first mention?

line 109: this statement should be supported with statistical analysis ("but at a much higher level than single knockdown") for example cluster B looks similar between double KD and AP2XII-1.
lines 113-116: statements again need quantitative support (statistical and otherwise).

lines 137-141: statement here about metabolism is partially quantified with numbers of proteins, but what are the housekeeping genes in question? the statement "no changes in metabolic capacity was detected" is strong given that the analysis was transcriptomic/proteomic and not functional.

the title "AP2XI-2 and AP2XII-1 synergistically silence merozoite-primed sexual commitment" is a strong statement and one that is backed up by transcriptome and proteome analysis but not by any

functional assay. I understand that depleting these 2 TFs leads to increases in the abundance of transcripts associated with merozoite primed sexual commitment, but the data here don't precisely support this statement. maybe "genes associated with mz-primed sexual commitment" would be better?

line 158: how is "complete switch" being quantified? it seems, again, that this is overstated (and unnecessarily so).

line 160: reference to proteins here but it refers to sup fig 3 which is transcriptomics. perhaps refer to figure 2? (where proteomic data are included?)

line 163: no quantitative/statistical analysis to support "strongly suppressed".

title on line 143 is somewhat difficult to understand. there is no noun after "switches" and the use of "features" is vague.

purple arrows on figure 4b almost look like staining until you look really close. . could you choose a color that is not used for representing a fluorescent channel? would minimize confusion.

Referee #2 (Remarks to the Author):

A. summary of the key results

Antunes et al. provide evidence for the in vitro production of *Toxoplasma gondii* pre-gametes (merozoites) that are normally restricted to the feline enteroepithelium. Authors include multiple lines of evidence for their findings, including immunofluorescence, transcriptomics, proteomics, and morphology, and cell division phenotypes that are consistent with the known biology of *T. gondii* merozoites. The authors also provide a novel, clear epigenetic mechanism for these life stage changes that build on their previous work and that of others in the *Toxoplasma* field. This will be a major methodological advance for *T. gondii* developmental biology.

B. Originality:

1. identification of the ApiAP2 transcription factors AP2XII-1 and AP2XI-2 that mediate phenotypes
2. linking ApiAP2 transcription factors with known MORC- and HDAC3-mediated mechanisms of asexual-to-presexual transition
3. Authors state (but do not show, discussed further in sections C-F) that the yield of merozoites is much greater than in previous in vitro attempts. If quantitative data support this, this is a novel finding. Heretofore, yield of in vitro merozoites has been prohibitively low.

B. Significance:

1. If broadly replicable, the parasite strains created will be useful for a large swathe of researchers the *Toxoplasma* field.
2. Many research questions surrounding *Toxoplasma* presexual stages will now be able to rely on easily cultured cells and parasites, rather than difficult primary cell cultures or live cats. This will be a

major advance in reducing the number of animals used in biomedical research.

C. Data & methodology: Approach is largely sound. Data are presented well. Specific issues numbered below.

1. The -omics data clearly show that in vitro merozoites are not tachyzoites. The omics data are less clear on distinctions between in vitro merozoites and bradyzoite parasites, further discussed in “F, suggested improvements.”
2. Throughout, there is a lack of quantification for immunofluorescence that needs to be improved, also see sections D and F.
3. Selection of merozoite markers: GRA11b is the only validated merozoite-specific marker, which was developed by one of the authors. Authors should provide quantitative evidence that GRA11b is elevated, especially in relation to their previous attempts at merozoite production described in their Farhat et al. MORC/HDAC3 paper. Only after GRA11b expression is established and quantified should GRA80 and GRA81 be introduced.
4. Related to above, GRA80 and GRA81 are not used merozoite markers. I have not seen other papers that characterize GRA80 or 81, and authors provided no citations. If authors wish to use these markers and to support statements like that in line 80 “GRA80, a merozoite-specific protein,” they need to cite other papers or provide their own evidence that GRA80 and GRA81 are indeed restricted to merozoites and not another parasite life stage like bradyzoites. As is, authors show lack of 80/81 in tachyzoites and presence of GRA80 in GRA11b-positive parasites in cat tissue (which is excellent), but presence of 81 in cats needs to be shown and lack of GRA80/81 in bradyzoites should also be shown using brain cysts.
5. percent of GRA11b-positive parasites that are GRA80 positive and GRA81 positive should be shown
6. Use of IMC proteins to confirm that in vitro merozoites replicate like bona fide merozoites is a good method for confirming merozoite identity. However, authors should be more explicitly describe how IMC1 and IMC7 staining patterns are both consistent with in vivo merozoites AND inconsistent with tachyzoite or bradyzoite staining patterns, as shown in the Dubey et al. 2017 paper cited in the references. As is, IMC7 surrounding a multinucleated mass during IAA treatment looks convincing when juxtaposed with untreated parasites but that should be noted in the text. Why is IMC3 not used? IMC1 staining in IAA-treated parasites should be juxtaposed with untreated parasites. Otherwise, it does not add much. GAP45 is used in lines 259-261 to support particular staining patterns of in vitro merozoites, but please provide a citation of GAP45 staining patterns in merozoites vs. tachyzoites, or show GAP45 in cat merozoites vs. in vitro merozoites.

D. Statistics:

1. The major advance in this paper is highly efficient production of merozoites. This is not supported by statistics because no immunofluorescence data is quantified, so this must be improved. For example, the existing version of Figure 1a should utilize a t-test or similar to demonstrate significantly higher numbers of GRA80-positive vacuoles in AP2XI-2 and AP2XII-1 KD strains. This should be repeated for the AP2XI-2/ AP2XII-1 double KD. As noted in section C, this must also be shown for GRA11b.
2. Authors need to state in Figure legends the number of experimental replicates, sample replicates, etc.
3. Authors need to describe how genes/proteins were systemically selected for presentation in

heatmaps in Figures 1 and 2. Otherwise, genes could have been cherry-picked.

F. Suggested improvements.

1. Prioritize GRA11b over GRA80 and GRA81

- a. Figure 1a should show GRA11b quantification
- b. GRA11b quantification should be shown for double AP2 knockdown
- c. ratios of colocalization of GRA80/GRA11b and GRA81/GRA11b should be shown ie what percent of parasites are double positive for each pair?
- d. move Fig. 2e,f to be Fig. 1 – GRA11b is the best validator and showing it in cat tissue relative to in vitro is the best way to validate

2. GRA11b should be used to validate GRA80 and 81

- a. colocalization of GRA81 with GRA11b in cats should be shown like it was shown for GRA80
- b. lack of GRA80 and GRA81 in bradyzoites must be shown. This is an important point because markers such as BRP1 from the Boothroyd lab are seen in merozoites and bradyzoites, but not tachyzoites. On ToxDB the Garfoot dataset shows that GRA80 has a lot of transcripts from purified bradyzoites in brain cysts, but GRA11B doesn't.

3. Support the biggest claim of the paper: that merozoites are more efficiently generated with AP2 double KD than other methods

- a. quantify percent GRA11b positivity (and secondarily GRA80/81) in the AP2 double KD parasites and directly compare to GRA11b positivity to MORC KD parasites and FR235222-treated parasites from Farhat et al's 2020 paper
- b. Relate percent GRA11b positivity in the double KD to GRA11b in cat cells.

4. Demonstrate that in vitro merozoites are not bradyzoite-like or sexual stage parasites.

a. transcriptomics

- i. The transcriptomics data in ToxoDB are quite comprehensive. As such, for Figures 1 and 2, transcriptional profiles in in vitro merozoites should also be compared to sporozoite and oocyst transcriptional profiles. If that is not possible, discuss why not
- ii. lines 124-125 (Figure 1f) are a little confusing, sounds like authors say that cluster IV is not bradyzoite-like. Do they mean the majority of genes in all of Figure 1f differ between in vitro merozoites vs cysts? quantify that- what percent of total genes are in cluster IV? a few genes in clusters I and V look similar in bradyzoites and in vitro merozoites
- iii. in line 164, authors call MIC17a,b,c and AMA2 merozoite-specific but the transcriptomic data show AMA2 and MIC17A expression in cysts, so the statement is inaccurate
- iv. lines 173-174: again, transcriptomic data show that ROP26-family genes are also expressed in cysts so saying they are exclusive to pre-gametes is misleading

b. IFAs:

- i. How many double KD parasites express BAG1 or other bradyzoite markers?
- ii. If there are BAG1-positive parasites, are they also GRA11b, GRA80, or GRA81 positive?

5. Other:

- i. Precision of language: throughout, when authors say merozoites, they should specify whether they are referring to bona fide merozoites or in vitro merozoites. This would be in keeping with the field's

- preference for calling bradyzoites developed in cell culture “in vitro” or “tissue culture” bradyzoites
- ii. Remove the word “kittens” from the abstract and replace with “cats.” “Infecting kittens” is the kind of wording that will unnecessarily upset animal rights groups.
 - iii. Ensure all antibodies are listed. Centrin reference may be missing.

G. References: ok.

H. Clarity and context: good, any recommendations are listed above.

Referee #3 (Remarks to the Author):

Toxoplasma gondii can cause life-threatening disease in immune-compromised individuals or when contracted congenitally and is estimated to have established lifelong infections in 25-30 % of the world's population. Besides being responsible for an important disease, its genetic tractability and ease with which it can be cultured, especially compared to other apicomplexan parasites such as *P. falciparum*, responsible for Malaria in humans, have made *T. gondii* an important model to study general features of apicomplexan biology. However, one of the biggest challenges in infection biology, irrespectively of the pathogen under investigation, has been the ability to reproduce the entire life cycle of pathogens under cell culture conditions. Yet, the ability to replicate the complete life cycle is important for the development of therapeutic strategies to interfere with a pathogen's transmission. Most parasites, including *T. gondii*, have a complex heterogenic life cycle. The inability to reproduce *T. gondii*'s sexual replication cycle, which normally takes place in cats, in cell culture has had immense impact on our understanding on this life cycle stage that, otherwise, seems an ideal target for the development of strategies to interfere with parasite transmission.

Thus, this manuscript addresses a very important and difficult to tackle aspect of toxoplasma biology, the mechanism that controls the cell fate decision of tachyzoites, rapidly replicating forms in non-feline mammalian hosts. Typically, tachyzoites proliferate or differentiate into bradyzoites, which reside in tissue cysts. Tissue cysts ingested by cats differentiate into merozoites and undergo sexual reproduction to form oocysts. The authors find that the depletion of two transcription factors, AP2XII-1 and AP2XI-2, promotes the differentiation of tachyzoites into merozoites followed by the typical asexual endopolygenic division. Furthermore, the authors start to shed light on the molecular mechanism triggering this change in cell fate decision. The ability to reproduce the entire sexual cycle in lab culture will without doubt have an enormous impact on toxoplasma research and efforts to restrict the spread of the parasite. The ability to produce merozoites and to trigger merogony outside of felines, as demonstrated in this manuscript, represents a first important step towards this goal.

Major Points:

While there is no question about the robustness of the presented data, I am still left wondering about the molecular mechanism triggered by the concomitant AP2XII-1 and AP2XI-2 depletion. I think it would be important that the authors clearly outline their working model, stating how the individual steps are supported by the current and previously published data. Such a model would also allow the reader to better understand how this study opens opportunities for the study of the

sexual replication cycle that go beyond those previously published for the depletion of MORC.

The way I understand the authors' current model is as follows: a heterodimer consisting of AP2XII-1 and AP2XI-2 recruits MORC and HDAC3 to the promoters of genes needed for merozoite formation. Based on this model, I have the following questions:

- Relationship between APs and MORC: are AP2XII-1 and AP2XI-2 'simply' upstream factors in a cascade in which MORC is the master regulator as stated in an earlier publication by the authors? Or can depletion of AP2XII-1 and AP2XI-2 lead to merozoite formation in a MORC-independent pathway as hinted in this manuscript ("Simultaneous depletion of AP2XII-1 and AP2XI-2 is sufficient to initiate the pre-sexual transcriptional program and silence the tachyzoite determinants in a remarkably more effective manner than depletion of MORC or inhibition of HDAC3"). Do the eluates of the AP2XII-1 and AP2XI-2 IPs give a hint whether other chromatin remodelers are involved as well?

Are there any AP2XII-1 and AP2XI-2 ChIP-seq peaks that are not enriched in MORC or vice-versa?

- Contribution of individual factors to merozoite formation: The authors write "This study unveils the role of a complex network of transcriptional repressors in regulating the commitment to merogony in *Toxoplasma*". However, to better understand the network, it would be important to quantify the contribution of the individual players. Ideally, the authors would show transcriptome and IF microscopy data of a) AP2XII-1 depleted, b) AP2XI-2 depleted, c) AP2XII-1 and AP2XI-2 depleted, d) MORC depleted, e) HDAC3 depleted and f) true merozoites (from cats) next to each other. For the transcriptome data this could be an extension of Fig 1f. Here, it would be interesting to know whether the transcriptome of AP2XII-1 and AP2XI-2 depleted cells resembles that of 'real' merozoites more closely than that of MORC-depleted cells. For the IF data, it would be important to compare different cells with the same antibodies.

It is not clear to me why the authors chose GRA80 upregulation as the primary selection criterium and only focused on the study of AP2XII-1 and AP2XI-2. If MORC and HDAC bind to many AP2s, are other AP2s required for the upregulation of other merozoite-specific proteins? For example, was GRA11b expressed after depletion of other AP2s (besides AP2XII-1)? How do the transcriptomes after depletion of other AP2s compare to those following depletion of AP2XII-1 or AP2XI-2?

- AP2XII-1 vs AP2XI-2: If it is a heterodimer consisting of AP2XII-1 and AP2XI-2 that recruits MORC and HDAC3, why do AP2XII-1 and AP2XI-2 share only such a small subset of target genes? Do AP2XII-1 and AP2XI-2 always bind DNA as heterodimer? Does the CHIP-seq data indicate different target sites (the selected regions shown in Fig 6 and Extended Data Fig 6 look very similar for AP2XII-1 and AP2XI-2). If I understand Fig 6a correctly, the AP2XII-1 and AP2XI-2 CHIP-seq analyses were done following the concomitant depletion of AP2XII-1 and AP2XI-2. If correct, it is not surprising that the AP2XII-1 and AP2XI-2 peaks disappear. I think it would be much more interesting to know if AP2XII-1 peaks are affected after AP2XI-2 depletion and vice-versa. The manuscript states that this has been done, but I could not find any figures.

How do the AP2XII-1 and AP2XI-2 interactomes differ? Have the IPs been repeated? The number of replicates should be indicated. It would be important to quantify the differences between the two

interactomes. In addition, it would be helpful if the authors discussed the available MS data in more detail. What are the differences? What are prominent interactors? Looking at the Supplementary Tables, I noticed the TGME49_267430 DnaJ domain-containing protein to be found in the eluates at high levels. In addition, AP2XII-1 and AP2XI-2 seemed to interact with other AP2s. How would this fit in the model?

If AP2XII-1 and AP2XI-2 synergically repress merozoite gene expression as stated (line 324), one would think that they act through different pathways, yet they form a heterodimer. How can this be explained?

Minor points

- Line 45: it is not clear to me what the authors mean by “epigenitors”.
- Lines 362-364: The authors write “At the genome level, there is a slight decrease in average accessibility between untreated and treated conditions.” Is it really possible to compare these two experiments without spike-in controls? What is the number of replicates?
- Line 995: Instead of stating “published in “ToxoDB.org”, it would be good to cite the original studies.
- Fig 1C: Why is the cluster B split into two parts?
- Fig 3: How does the schizogonic replication observed following AP2XII-1 and AP2XI-2 depletion compare to that in cats? Would it be possible to show images from schizogonic replication in cats matching the stages shown in Fig 3?
- Fig 5a: While the eluates from the AP2XII-1 and AP2XI-2 look very similar on the silver stained-gel, they look different after Coomassie staining (Supplementary Tables 4 and 5). What do other silver stained-gels look like? It would be nice if the major bands in Fig 5a could be labeled based on the MS data, listing likely candidates for the prominent bands.
- Fig 5d,e: It would be important to have a negative control, e.g. another AP2.
- Fig 6C: To me the UT heatmaps and IAA heatmap look indistinguishable, do they really stem from different experiments or has there been a mistake in the data analysis/display?
- Based on the RNA-seq data from merozoites isolated from cats, are AP2XII-1 and AP2XI-2 downregulated in merozoites compared to tachyzoites, are MORC and HDAC3 downregulated?

- It would be very useful if the authors showed a life cycle listing all the stages mentioned in the manuscript, including pre-gametes, enteroepithelial stages (EES), meronts, schizonts. It would probably make sense to merge Fig 3a and extended data Fig 1a and show the life cycle figure in the main manuscript. Here the authors could also indicate the cell fate transitions under investigation.

- While it is probably beyond the scope of this study to find a means to drive merozoite differentiation into gamete formation or even fertilization, it would be useful to know the level of cellular heterogeneity during the induced merogony and during merogony in cats. In lab culture, is there any evidence for the formation of macrogametocytes or microgametocytes? Could this be evaluated using scRNA-seq?

The authors state that SRS48 and SRS59 have been predicted to promote gamete development and fertilization. Would it be possible to test this prediction in the AP2XII-1 and AP2XI-2 depleted cells by overexpression of SRS48 or SRS59? Similarly, the authors speculate that AP2XI-1 may act downstream of AP2XII-1 and AP2XI-2, could this be tested by concomitant AP2XII-1 and AP2XI-2 depletion and AP2XI-1 overexpression?

Other comments:

I lack insight knowledge of *T. gondii* cell biology and thus cannot judge the importance and validity of claims related to the expression of MIC, RON, ROP, SRS and GRA proteins.

Author Rebuttals to Initial Comments:

Dear editor, Dear reviewers,

We thank all three reviewers for dedicating their valuable time in the evaluation of our manuscript. Their constructive criticism and insightful suggestions have been instrumental in strengthening our work.

We acknowledge that our initial submission lacked some critical controls and the statistics were not sufficiently detailed. We deeply appreciate these observations and have done our best to address each remark but also by integrating additional experimental data. Our commitment has always been to deliver the most meticulous and reliable research possible.

We modestly hope that our research findings could potentially serve as a reference for future studies in this field, fostering the investigation of novel strategies to explore the sexual stages of *T. gondii*.

Herein, we present our responses to each reviewer, addressing each point individually. Please excuse any repetition in our responses, as there were instances of overlapping comments or criticisms across the reviewers.

Referee #1 (Remarks to the Author):

Summary of key results: This is a very exciting paper demonstrating that the simultaneous knockdown of two *T. gondii* transcription factors results in a rewiring of the *T. gondii* developmental program in vitro, producing what are very clearly pre-sexual stages (meronts/merozoites). The rationale for the study and identification of these specific TFs is strong and the results are very clear and convincing. Using this system, the authors are able to increase our understanding of presexual development in *T. gondii* without the use of cats, an important step for the field. Molecular and epigenetic data are also provided demonstrating how these two TFs work together to silence the sexual developmental program. Overall this is an exciting development for the field as these life stages are inaccessible to most and nearly impossible to obtain in the numbers shown here.

- Thank you for your encouraging feedback. Your appreciation of our efforts and recognition of our study's potential impact is sincerely appreciated.

1. Originality and significance: One thing that dampens enthusiasm a bit is that prior work (cited below) using similar approaches knocking down MORC led to similar changes in the transcriptome and switching of parasites towards a more merozoite-like state. Including some demonstration signs of sexual development (which is lacking in the AP2KD parasites here) may be occurring. In that study the authors also claimed that that the discovery would permit access to sexual stages. From the present study it is clear that the double AP2

KD presented is now the most robust way to induce progress towards sexual development in vitro that is known. However, the current study represents a further refinement of the genetic regulators of MORC and HDAC3 binding rather than uncovering the core pathway required for entrance into the sexual or cat stages of development.

Farhat, D.C., Swale, C., Dard, C. et al. A MORC-driven transcriptional switch controls *Toxoplasma* developmental trajectories and sexual commitment. *Nat Microbiol* 5, 570–583 (2020).

We would like to clarify some aspects and potentially overlooked elements of our studies:

- Thank you for highlighting the findings from Farhat et al. 2020. We acknowledge that MORC depletion in cell culture leads to the formation of mixed stages with different developmental trajectories. As demonstrated in Extended Data Fig. 3k of our study, MORC-depleted parasites formed vacuoles expressing either bradyzoite (BCLA+) or merozoite (GRA81+) markers in a mutually exclusive manner. This developmental heterogeneity presented challenges in precisely controlling the progression to (pre)gametes, resulting in the limited monitoring of each stage. The complexity observed in MORC-depleted parasites is distinct from the more straightforward progression observed in AP2XII-1/AP2XI-2-depleted parasites, where there is a clear transition exclusively from tachyzoites to merozoites. The differences in the developmental trajectories between these two scenarios further highlight the unique roles of different regulatory factors in *Toxoplasma gondii*'s sexual development.
- In addition, we observed, but did not fully elaborate upon in Farhat et al. 2020, that MORC and HDAC3 bind to telomeric repeats and their depletion or inhibition can have broad and unspecific effects on transcription. This raises the possibility of their involvement in regulating transcription in a telomere-dependent manner. In our current study, we aimed to differentiate the direct transcriptional impact of MORC and HDAC3 on merozoite genes from their expected telomeric function. By focusing on AP2XII-1 and AP2XI-2, we were able to explore their specific roles in sexual development without the potential confounding effects of telomeric regulation. This approach allowed us to gain insights into the direct transcriptional regulation of merozoite genes by AP2XII-1 and AP2XI-2, further enhancing the understanding of the mechanisms involved in *Toxoplasma gondii*'s sexual development.
- We've now incorporated all relevant data concerning the role of MORC at the telomere in our new Extended Data Figure 1 to provide the reader with the rationale behind our efforts to identify additional specific switches. Specifically, we showed that MORC when released from chromosome ends disrupts subtelomeric gene silencing (Extended Data Fig. 1a, b), causing telomere dysfunction, mitotic bypass, and aberrant polyploid zoite accumulation (Extended Data Fig. 1c).

We are pleased to note the reviewer's recognition of the instrumental role played by MORC and HDAC3 in our study. Their involvement has indeed been crucial in identifying additional significant contributors to the parasite's lifecycle transitions. Furthermore, our current research takes a significant step forward by proposing a powerful method to study sexual development without disturbing the parasite's genomic stability, which we suspect MORC does through its telomeric function. This method not only enhances the clarity and relevance of our study but also addresses inquiries about the novelty and importance of our work. We believe that this advancement in our understanding of sexual development in *Toxoplasma gondii* opens up new avenues for research and has implications for the broader field of parasite biology.

2. One broad comment is that I cannot find reference to the frequency of the observed changes in morphology among the population of parasites treated with IAA. While it is mentioned and very clear that there is a lot of asynchrony in these parasites, are all parasites undergoing some form of commitment to presexual stages? or are some maintaining their tachyzoite morphology? It seems like a critical component of the manuscript to assess the extent of the penetrance of the developmental phenotype and whether one particular form might be favored at a given time post-IAA treatment.

- We greatly appreciate the valuable feedback provided by the reviewer. In response to this feedback, we have conducted a diligent examination and quantification of the transition from tachyzoites to pre-sexual stages. To achieve this, we utilized a range of validated *in vivo* merozoite markers. This analysis allowed us to rule out any residual presence of tachyzoites by monitoring GRA2 expression. Additionally, we addressed the reviewer's concerns by investigating the conversion towards bradyzoite stages using appropriate markers such as BCLA, DBA, and BAG1. The results, presented in our Extended Data Fig. 3 and Fig. 4f, showcase the *in vitro* progression of APXII-1/AP2XI-2-depleted parasites towards pre-sexual stages. These figures illustrate the gradual emergence of various morphotypes (B to E) over time, mirroring the observations made in infected feline intestines. We believe that these findings provide valuable insights into the developmental progression of the parasite and support the conclusions drawn in our study.
- Accordingly, we now state in the main text the following regarding those quantification :

“Morphotypes A-E are difficult to study *in vivo* because they vary in size and shape and develop asynchronously in different regions of the digestive tract^{19,20,26,31,33}. Here, we were able to follow the initial steps of *in vitro* merogony using several stage-specific markers. Asynchronous nuclear division cycles were detected by DNA, histone, or centrosome staining (Supplementary Fig. 5d-g). At 12 hours post-IAA treatment, a significant proportion of the tachyzoite population transitioned into morphotype B (3-4 nuclei and ROP26+), reaching up to 75% (Fig. 4f). Within 24 hours, morphotypes C/D (8-32 nuclei and ROP26+) and mononuclear merozoites (GRA11b+ GRA80+) coexisted in

culture (Fig. 4f). After 48 hours, nearly 98% of the parasite population expressed merozoite markers (GRA11b+, GRA80+), whereas typical tachyzoite (GRA2) or bradyzoite (BAG1) markers were absent (Fig. 4f), aligning with our extensive transcriptome and proteome analyses.”

3. One question is how to retrieve the meronts intact from infected host cells intact as they are larger than parasites. The method for how this was achieved was not detailed in the methods that I could find. Is this by needle passage or some other method? this is an important detail and it should be demonstrated if the meronts are actually intact after extraction from the host cells or if they have been lysed themselves (perhaps their nuclei stay intact and are ultimately the substrate for the chip seq?). This is not just a trivial method question but would directly inform the data obtained throughout the manuscript from the different life stages and time points. Since merozoites have not been generated with such efficiency in any previous work, this is a new problem that we, as a field are very happy to have! but should be clarified explicitly in the manuscript nonetheless.

- We appreciate the reviewer's inquiry regarding the method of meront extraction from host cells. However, there appears to be a misunderstanding regarding our omics protocols. In all of our RNA-seq (both Illumina and Nanopore DRS), ChIP-seq, ATAC-seq, and comprehensive MS-proteomic experiments, we did not physically isolate meronts from the host cells. Instead, we harvested intracellular zoites in bulk at different times post IAA treatment. During the collection of these samples, fluctuations in host cell RNA/DNA/proteins were observed. To address this, we applied bioinformatics techniques to subtract these host cell components from the raw data. This computational process effectively isolated and scrutinized the distinct alterations in the *Toxoplasma* transcriptome, proteome, and cistrome. By bypassing the physical extraction of the parasite from its host cell, we avoided concerns about meront integrity post-extraction. This approach not only addressed the unique challenges presented by the size/shape of meronts but also provided valuable insights into the data gathered across various life stages and time points in our study. We believe that clarifying this aspect of our methodology strengthens the reader's understanding of our approach and the results derived from it.
- Moving forward, while not directly relevant to the current study, we have plans to investigate the development of pre-gametes through single-cell analysis. To facilitate this analysis, we have observed that Morphotype E, which represents a later stage in development, is capable of exiting but not re-invading host cells. This characteristic allows for the isolation of Morphotype E for further study. In our preliminary experiments, we have tested the induction of merozoite egress using ionophore treatment. Ionophores are known to promote the artificial egress of tachyzoites from host cells. We have observed effective responses in inducing merozoite egress using this method. These findings provide a promising avenue for further investigation and enable the study of pre-gamete development using single-cell analysis techniques.

4. Finally, the manuscript suffers from many non-quantitative statements that should be qualified using statistical or other quantitative data. I have detailed a number of these below but they are found throughout the manuscript in the results section. I share the excitement of the authors about this discovery and I can appreciate their desire to convey their own excitement! However, these statements are not necessary to drive the impact home. the data speak well to the fact that this genetic manipulation drives *T. gondii* towards presexual stages.

- We appreciate the reviewer's feedback and share the excitement surrounding this discovery. We value his/her comments on the importance of data-driven assertions. As seen in our responses below, we've now enriched our study with robust statistical analysis supporting our conclusions. These additions amplify the impact and robustness of our work on *T. gondii*'s presexual stages. Thank you for helping us enhance the scientific rigor of our manuscript.

More specific comments are found here:

5. In the array heatmaps the replicates are reduced to single values. Is there any reason for this? i prefer showing all the data if possible.

- We recognize that incorporating all the data in a heatmap visualization for array data would be advantageous. However, to strike a balance between comprehensiveness and clarity, we've chosen to display the average of three replicates, which are then converted into the Transcripts Per Million (TPM) format. This approach not only helps to streamline the visualization but also provides a consistent basis for comparison with prior data that has been expressed in the same TPM format.

6. Significance values are lacking from much of the manuscript for all of the proteomics and transcriptomics data. How are the authors deciding that a given transcript/protein is of different abundance? references to DEseq2 for example are made but no significance thresholds are given. This should be in the figure legends and methods text if possible.

- **We take the Reviewer's point that our methodology and statistical thresholds were not outlined sufficiently clearly in our initial submission.** For determining differential abundance of transcripts and proteins, we employed **DEseq2 analysis**. This method was chosen specifically to guard against biased gene selection. Transcripts and proteins were considered to be differentially expressed if they surpassed these defined thresholds.
- **Using DEseq2**, we determined that the cluster initially labeled as "bradyzoite" did not meet the significance threshold and was therefore excluded from the heatmap visualization. Furthermore, we revised the criteria for identifying up- or down-regulated genes by implementing an 8-fold change threshold (p -value < 0.05, Benjamini-Hochberg correction). As a result of this adjustment, we discovered a total of 490 significantly differentially expressed transcripts.

Among them, 295 genes displayed increased expression, while 195 genes exhibited decreased expression, attributable to the concurrent depletion of two AP2s (refer to new Fig. 1f).

- Regarding the proteomics data described in the legend of Supplemental Fig. 3, we produced three biological replicates for each examined condition. Our quantification approach involved applying a logarithmic transformation (\log_2) to the filtered, normalized, and imputed protein abundances across all samples. To assess statistical significance, we utilized the limma method to perform pairwise comparisons between samples. Differentially abundant proteins were identified by applying a threshold of either ≥ 1 or ≤ -1 for the $\log_2(\text{fold change})$ and a p-value of ≤ 0.01 . These criteria ensured a false discovery rate below 5%, as supported by the Benjamini-Hochberg estimator. To comprehensively compare protein abundances across all four conditions, we employed Analysis of Variance (ANOVA). This allowed us to evaluate the overall differences in protein expression among the various conditions. We hope that these clarifications enhance your understanding of our methodology and data analysis techniques.

7. The expression heat maps I presume that the data were mean centered prior to clustering? It would be helpful to indicate this in the legend whereas what is there now is the phrase "data are shown as log2 fold changes" but I think that is incorrect or at least missing the important detail of what the change is compared to.

- In the case of the expression heatmaps, we want to clarify that the data were mean-centered before clustering, which was inadvertently omitted from the figure legend. We apologize for this oversight. Additionally, the statement "data are shown as \log_2 fold changes" was oversimplified and failed to specify the comparison point for these changes.

8. if gra81 is different from gra80, can you please indicate its gene model number at first mention?

- The two genes are distinct, and we now make this clear as soon as they are mentioned in the text.

9. line 109: this statement should be supported with statistical analysis ("but at a much higher level than single knockdown") for example cluster B looks similar between double KD and AP2XII-1.

- The paper has been extensively revised, particularly the heatmaps. After the DeSeq-2 analysis, we've consolidated all heatmaps into a single, more relevant one, complete with appropriate statistics. Accordingly, references to clusters A-D have been removed given the current flow of the article.

10. lines 113-116: statements again need quantitative support (statistical and otherwise).

- In the revised version, we have included quantitative statistical data to demonstrate the significance of our findings. Consequently, we have updated the text to reflect the statistical analysis conducted: “DESeq2 analysis revealed that depletion of both AP2 factors was found to be necessary to upregulate 65% (194/295) of the identified genes (fold change ≥ 8 ; p-value < 0.05). In contrast, single knockdowns of AP2XI-2 and AP2XII-1 resulted in expression of a lower proportion of genes, 9% and 13.5%, respectively (Extended Data Fig. 4b,c). This transcriptional trend also extends to tachyzoite-specific genes, whose repression is more pronounced when AP2s are simultaneously depleted (Extended Data Fig. 4c).”

11. lines 137-141: statement here about metabolism is partially quantified with numbers of proteins, but what are the housekeeping genes in question? the statement "no changes in metabolic capacity was detected" is strong given that the analysis was transcriptomic/proteomic and not functional.

- We take this point and removed this mention from the main text.

12. the title "AP2XI-2 and AP2XII-1 synergistically silence merozoite-primed sexual commitment" is a strong statement and one that is backed up by transcriptome and proteome analysis but not by any functional assay. I understand that depleting these 2 TFs leads to increases in the abundance of transcripts associated with merozoite primed sexual commitment, but the data here don't precisely support this statement. maybe "genes associated with mz-primed sexual commitment" would be better?

- In response to this comment, we've revised the title as follows: “AP2XI-2 and AP2XII-1 synergistically silence merozoite-specific genes”.

13. line 158: how is "complete switch" being quantified? it seems, again, that this is overstated (and unnecessarily so).

- We acknowledge the reviewer's concern about the term "complete switch". We have substituted this term with "developmental switch" in the main text. In addition, we have now quantified the *in vitro* development of APXII-1/AP2XI-2-depleted parasites towards pre-sexual stages, revealing the stepwise appearance of different morphotypes (B to E) over time, similar to what has been reported in infected cat's gut. These findings are now presented in Extended Data Fig. 3 and Fig. 4f.

14. line 160: reference to proteins here but it refers to sup fig 3 which is transcriptomics. perhaps refer to figure 2? (where proteomic data are included?)

- The text in question has been significantly edited to address this point, thereby offering a more accurate reflection of the data presented.

15. line 163: no quantitative/statistical analysis to support "strongly suppressed".

- We have refined our statements regarding MIC proteins to ensure a more balanced perspective. While this revised version provides improved quantification of *in vitro* conversion, it is important to note that the markers used primarily represent residents of dense granules and rhoptries.

16. title on line 143 is somewhat difficult to understand. there is no noun after "switches" and the use of "features" is vague.

- The original title is obviously unclear. To address this, we have rephrased it as: "Simultaneous depletion of AP2XI-2 and AP2XII-1 induces transition from tachyzoite to merozoite phenotype".

17. purple arrows on figure 4b almost look like staining until you look really closely. Could you choose a color that is not used for representing a fluorescent channel? would minimize confusion.

- Fig. 4b has now been moved to Supplementary Fig. 5c. To avoid confusion, we have replaced the purple arrows with white ones.

Referee #2 (Remarks to the Author):

A. summary of the key results

Antunes et al. provide evidence for the *in vitro* production of *Toxoplasma gondii* pre-gametes (merozoites) that are normally restricted to the feline entero-epithelium. Authors include multiple lines of evidence for their findings, including immunofluorescence, transcriptomics, proteomics, and morphology, and cell division phenotypes that are consistent with the known biology of *T. gondii* merozoites. The authors also provide a novel, clear epigenetic mechanism for these life stage changes that build on their previous work and that of others in the *Toxoplasma* field. This will be a major methodological advance for *T. gondii* developmental biology.

- Thank you for your constructive insight. Your recognition of our novel contributions is truly valued.

B. Originality

- identification of the ApiAP2 transcription factors AP2XII-1 and AP2XI-2 that mediate phenotypes

- linking ApiAP2 transcription factors with known MORC- and HDAC3-mediated mechanisms of asexual-to-presexual transition
- Authors state (but do not show, discussed further in sections C-F) that the yield of merozoites is much greater than in previous *in vitro* attempts. If quantitative data support this, this is a novel finding. Heretofore, yield of *in vitro* merozoites has been prohibitively low.

- In our forthcoming responses, we will carefully address all of your expressed concerns, with a specific focus on addressing the quantification of *in vitro* merozoite yield. Additionally, we will introduce new experimental data to substantiate our claims regarding the effectiveness of our *in vitro* pre-gamete production system, aiming to provide a comprehensive support for our assertions.

B. Significance

If broadly replicable, the parasite strains created will be useful for a large swathe of researchers the Toxoplasma field. Many research questions surrounding Toxoplasma presexual stages will now be able to rely on easily cultured cells and parasites, rather than difficult primary cell cultures or live cats. This will be a major advance in reducing the number of animals used in biomedical research.

- Following the publication of the pre-print, we already received multiple requests from colleagues to share these strains. We are more than willing to provide not only the strains, but also the reagents we've developed to track merozoite proteins, in a collective effort to advance the field.

C. Data & methodology: Approach is largely sound. Data are presented well. Specific issues numbered below.

1. The -omics data clearly show that *in vitro* merozoites are not tachyzoites. The omics data are less clear on distinctions between *in vitro* merozoites and bradyzoite parasites, further discussed in “F, suggested improvements.”

- We have found that merozoites produced *in vitro* do not expressed BAG1, a hallmark of bradyzoite stage. However, we identified proteins, such BRP1 and ROP26, expressed in both bradyzoites and merozoites. These are shared factors in both stages which have been initially discovered in bradyzoite and labeled as bradyzoite markers unbeknownst to the fact that they also occur in other developmental stages that have not been accessible at the time.
- Further details are elaborated in the section "F, suggested improvements".

2. Throughout, there is a lack of quantification for immunofluorescence that needs to be improved, also see sections D and F.

- Please refer to our responses in sections D and F.

3. Selection of merozoite markers: GRA11b is the only validated merozoite-specific marker, which was developed by one of the authors. Authors should provide quantitative evidence that GRA11b is elevated, especially in relation to their previous attempts at merozoite production described in their Farhat et al. MORC/HDAC3 paper. Only after GRA11b expression is established and quantified should GRA80 and GRA81 be introduced.

- Please refer to our responses in sections D and F.

4. Related to above, GRA80 and GRA81 are not used merozoite markers. I have not seen other papers that characterize GRA80 or 81, and authors provided no citations. If authors wish to use these markers and to support statements like that in line 80 “GRA80, a merozoite-specific protein,” they need to cite other papers or provide their own evidence that GRA80 and GRA81 are indeed restricted to merozoites and not another parasite life stage like bradyzoites. As is, authors show lack of 80/81 in tachyzoites and presence of GRA80 in GRA11b-positive parasites in cat tissue (which is excellent), but presence of 81 in cats needs to be shown and lack of GRA80/81 in bradyzoites should also be shown using brain cysts.

- We would like to clarify that GRA80 and GRA81 are novel markers that have not been previously characterized. However, our research has yielded promising data using GRA11b as a proxy, indicating that GRA80 and GRA81 could serve as valuable tools for tracking pre-gamete stages *in vitro*:
 - I. GRA80 and GRA82 decorate vacuoles hosting merozoites in cat epithelium (Extended Data Fig. 2e and Fig. 1b). Unfortunately, we were unable to assess the localization of GRA81 due to the unavailability of a suitable IFA-grade antibody.
 - II. GRA80 is not detected in BFD1-overexpressing bradyzoites, nor in bradyzoites embedded in mice brain cysts (Extended Data Fig. 2b,c). Similar results were obtained for GRA82.

For additional supporting evidence, please refer to our answers in sections D and F.

5. Percent of GRA11b-positive parasites that are GRA80 positive and GRA81 positive should be shown

- We meticulously revisited the expression of GRA11b in correlation with GRA80 and GRA82 in diverse genetic backgrounds and determined that more than 95% of vacuoles exhibit either GRA11b+/GRA80+ or GRA11b+/GRA82+ only when AP2XI-2 and AP2XII-1 are simultaneously depleted 48 hours post-IAA. The revised findings are depicted in Extended Data Fig. 3a-e and Fig. 4f.

6. Use of IMC proteins to confirm that in vitro merozoites replicate like bona fide merozoites is a good method for confirming merozoite identity. However, authors should be more explicitly describe how IMC1 and IMC7 staining patterns are both consistent with in vivo merozoites AND inconsistent with tachyzoite or

bradyzoite staining patterns, as shown in the Dubey et al. 2017 paper cited in the references. As is, IMC7 surrounding a multinucleated mass during IAA treatment looks convincing when juxtaposed with untreated parasites but that should be noted in the text.

- We were fortunate to leverage the tools developed by Marc-Jan Gubbels' team, which played a pivotal role in enhancing our understanding of endopolygeny in merozoites. In our revised manuscript, we present a comprehensive elucidation of the significance of these markers. Special attention is dedicated to the groundbreaking research conducted by Dubey et al. (2017), who initially characterized IMC1, IMC3, and IMC7 during the developmental stages of *Toxoplasma gondii* and *Sarcocystis neurona*. Their work laid the foundation for our investigations and serves as a cornerstone in the field.

Why is IMC3 not used?

- After careful consideration, we concluded that utilizing both IMC1 and IMC3 was unnecessary since they exhibited equivalent expression patterns throughout the stages we examined. However, the inclusion of IMC7, generously provided by MJ Gubbels, was indispensable. IMC7 played a pivotal role in specifically labeling the mother cytoskeleton during various asexual division modes, as demonstrated by Dubey et al. (2017). The specific marking provided by IMC7 greatly enhanced our understanding of the processes under investigation.

IMC1 staining in IAA-treated parasites should be juxtaposed with untreated parasites. Otherwise, it does not add much.

- We take this point and have added the untreated control side by side in the new Fig. 3i.

GAP45 is used in lines 259-261 to support particular staining patterns of in vitro merozoites, but please provide a citation of GAP45 staining patterns in merozoites vs. tachyzoites, or show GAP45 in cat merozoites vs. in vitro merozoites.

- In our study, we sought to explore the role of GAP45 in sexual stages, which, to our knowledge, had not been previously characterized. To investigate this, we performed staining on the epithelium of infected cats using the GAP45 antibody. In a significant discovery, we observed that GAP45 not only delineates the surface of merozoites but also exhibits a distinct pattern outlining the periphery of a macrogametocyte (new Fig. 2f). This novel finding highlights the previously unexplored involvement of GAP45 in sexual stages and expands our understanding of its role in the lifecycle of the parasite.

D. Statistics:

1. The major advance in this paper is highly efficient production of merozoites. This is not supported by statistics because no immunofluorescence data is quantified, so this must be improved.

- Based on valuable feedback, we thoroughly investigated the transition from tachyzoites to pre-sexual stages using validated *in vivo* merozoite markers. We carefully monitored GRA2 expression to ensure the absence of residual tachyzoites. Additionally, we addressed reviewer concerns by employing specific markers (BCLA, DBA, BAG1) to rule out any conversion towards bradyzoite stages. Our findings, presented in Extended Data Fig. 3, Extended Data Fig. 2b, and Fig. 4f, demonstrate the *in vitro* progression of APXII-1/AP2XI-2-depleted parasites towards pre-sexual stages. We observed a gradual emergence of various morphotypes (B to E) over time, mirroring the observations in infected feline intestines.

For example, the existing version of Figure 1a should utilize a t-test or similar to demonstrate significantly higher numbers of GRA80-positive vacuoles in AP2XI-2 and AP2XII-1 KD strains. This should be repeated for the AP2XI-2/ AP2XII-1 double KD. As noted in section C, this must also be shown for GRA11b.

- To establish statistical significance, we expanded our CrisR-Cas9 screen to include GRA80 (new Fig. 1d), GRA11b (new Fig. 1c), and GRA81 (new Fig. 1e). Each experiment was replicated three times for each condition. Statistical analysis was performed using one-way ANOVA, followed by Tukey's multiple comparison test to identify significant differences in the number of GRA-positive vacuoles. Furthermore, we incorporated controls involving cas9-mediated inactivation of MORC and the AP2XI-2/AP2XII-1 (using a double gRNA). These additional measures address the concerns raised and provide further clarity to our study.

2. Authors need to state in Figure legends the number of experimental replicates, sample replicates, etc.

- In the legends of each figure, we have included information on the number of experimental replicates conducted. It is important to note that although multiple replicates were performed, we typically present one representative dataset in each figure for clarity and conciseness.

3. Authors need to describe how genes/proteins were systemically selected for presentation in heatmaps in Figures 1 and 2. Otherwise, genes could have been cherrypicked.

- We appreciate your feedback regarding the clarity of our methodology and statistical thresholds in the initial submission. To ensure unbiased gene selection, we employed **DEseq2 analysis** for determining the differential abundance of transcripts and proteins. Through this analysis, we established

defined thresholds to distinguish transcripts and proteins of different abundance. We acknowledge that the originally identified "bradyzoite" cluster did not meet the significance threshold and, therefore, was appropriately excluded from the heatmap visualization. To redefine the number of genes exhibiting up- or down-regulation, we applied an 8-fold change threshold (p-value < 0.05, Benjamini-Hochberg correction). This adjustment led to the identification of a total of 490 significantly differentially expressed transcripts. Among them, 295 genes showed increased expression, while 195 genes exhibited decreased expression due to the concurrent depletion of the two AP2s. Please refer to the new **Fig. 1f** for a visual representation of these findings.

- In our proteomics analysis, we conducted three biological replicates for each condition under investigation, ensuring robustness and reliability (**Supplementary Table 3**). To quantify the protein abundances, we applied a logarithmic transformation (log₂) to the filtered, normalized, and imputed protein data across all samples. For determining statistical significance, we utilized the limma method to perform pairwise comparisons between samples. Differentially abundant proteins were identified based on a threshold for log₂(fold change) of either ≥ 1 or ≤ -1 , combined with a p-value of ≤ 0.01 . By employing these criteria, we aimed to maintain a false discovery rate below 5%, as confirmed by the Benjamini-Hochberg estimator. To enable a comprehensive comparison of protein abundances across all four conditions, we employed Analysis of Variance (ANOVA). This allowed us to evaluate the overall differences in protein expression among the various conditions. We trust that these clarifications provide a better understanding of our methodological approach and data analysis techniques in the context of our proteomics data.

F. Suggested improvements.

1. Prioritize GRA11b over GRA80 and GRA81

a. Figure 1a should show GRA11b quantification

- To establish statistical significance, we performed additional analyses in our study. We expanded our CrisR-Cas9 screen to include GRA80 (**new Fig. 1d**), GRA11b (**new Fig. 1c**), and GRA81 (**new Fig. 1e**). For each of these experiments, we conducted three replicates for each condition, ensuring robustness and reliability of the results. To analyze the data, we employed one-way ANOVA, a widely used statistical test, to assess the overall differences among the groups. Subsequently, we conducted Tukey's multiple comparison test to determine significant differences in the number of GRA-positive vacuoles between specific groups. These statistical analyses allowed us to evaluate the significance of our findings and provide meaningful insights into the differences observed in the abundance of GRA-positive vacuoles among the experimental conditions.

b. GRA11b quantification should be shown for double AP2 knockdown

- We have incorporated controls involving cas9-mediated inactivation of MORC and the AP2XI-2/ AP2XII-1 (using a double gRNA) (new Fig. 1c-e).

c. Ratios of colocalization of GRA80/GRA11b and GRA81/GRA11b should be shown i.e. what percent of parasites are double positive for each pair?

- We have diligently conducted co-labeling experiments and quantified the results, providing compelling visual evidence of the co-labeling of GRA11b/GRA80 (new Fig. 4f and new Extended Data Fig. 3a,b,c) and GRA11b/GRA82 (new Extended Data Fig. 3d,e). These experiments were performed across various knockdown backgrounds and even when BFD1 was overexpressed, ensuring the robustness and generalizability of our findings (new Extended Data Fig. 3b). Despite our best efforts, we encountered challenges in producing a reliable IFA-grade antibody for GRA81, a 14-kDa protein that appears to be non-immunogenic. As a result, we were unable to demonstrate co-labeling of GRA81 with GRA11b. We acknowledge this limitation and have provided a clear explanation for the absence of GRA81 co-labeling in our study. Overall, the co-labeling experiments strengthen our findings and provide valuable insights into the spatial relationship between GRA11b and GRA80/GRA82, highlighting their potential functional associations.

d. move Fig. 2e,f to be Fig. 1 – GRA11b is the best validator and showing it in cat tissue relative to in vitro is the best way to validate

- We concur with your suggestion and have accordingly relocated the labeling of GRA11b and GRA80 in cat tissue to Fig. 1b.

2. GRA11b should be used to validate GRA80 and 81

a. colocalization of GRA81 with GRA11b in cats should be shown like it was shown for GRA80

- Regrettably, we faced difficulties in generating a reliable IFA-grade antibody for GRA81, a 14-kDa protein that appears to be non-immunogenic. Consequently, we were unable to demonstrate its co-localization with GRA11b in cats. To compensate for this limitation and provide additional insights, we have successfully established GRA82 as a credible merozoite marker. Our findings now demonstrate the presence of GRA82 in parasite vacuoles within infected cat epithelium (new Extended Data Fig. 2c). This novel marker contributes valuable information to our understanding of merozoite biology and provides further support for the characterization of GRA82 as a useful tool in future studies.

b. lack of GRA80 and GRA81 in bradyzoites must be shown. This is an important point because markers such as BRP1 from the Boothroyd lab are seen in

merozoites and bradyzoites, but not tachyzoites. On ToxoDB the Garfoot dataset shows that GRA80 has a lot of transcripts from purified bradyzoites in brain cysts, but GRA11B doesn't.

- We have considered the findings of the Garfoot et al. study, but would like to respectfully offer a differing interpretation. While GRA80 transcripts were reported in their study, the levels were comparatively low (< 40 TPM) when set against the merozoite and EES stages, which show expression exceeding 5000 TPM (new Supplementary Fig. 1b). Thus, we posit that the findings from Garfoot et al. may represent transcriptional noise.
- On the protein level, our data indicates that GRA80 (TGME49_273980) is not detected in bradyzoites, whether they are generated in vitro by overexpressing BFD1 or observed in mouse brain cysts, as shown in our new Extended Data Fig. 2b,c and Extended Data Fig. 3b.
- Regarding GRA81, data from Hehl et al. and Garfoot et al. indicate that its mRNA expression is exclusive to the merozoite and EES stages (new Supplementary Fig. 1b). Unfortunately, despite our efforts, we were unable to produce an immunofluorescence assay-grade antibody for GRA81 to further investigate its expression in cysts or in BFD1-overexpressing parasites.

3. Support the biggest claim of the paper: that merozoites are more efficiently generated with AP2 double KD than other methods

a. Quantify percent GRA11b positivity (and secondarily GRA80/81) in the AP2 double KD parasites and directly compare to GRA11b positivity to MORC KD parasites and FR235222-treated parasites from Farhat et al. 2020.

- We have conducted meticulous quantification of GRA11b, both individually and in combination with GRA80 or GRA82, across single and double AP2s knockdown backgrounds. We also included MORC knockdown and BFD1 overexpression as control conditions. The results are presented in our new Extended Data Fig. 3 and Fig. 4f. These figures depict the *in vitro* progression of APXII-1/AP2XI-2-depleted parasites towards pre-sexual stages, demonstrating the gradual emergence of various morphotypes (B to E) over time, which closely resemble the observations made in infected feline intestines.

b. Relate percent GRA11b positivity in the double KD to GRA11b in cat cells.

- GRA11b, when evaluated within the framework of the double knockdown mutant, demonstrated 98% positivity 48 hours post-induction with auxin (Extended Data Fig. 3b,c,e and Fig. 4f). This level is comparable to what is observed in cat cells, although challenging to assess accurately due to the limited availability of intestinal samples.

4. Demonstrate that *in vitro* merozoites are not bradyzoite-like or sexual stage parasites.

a. transcriptomics

i. The transcriptomics data in ToxoDB are quite comprehensive. As such, for Figures 1 and 2, transcriptional profiles in *in vitro* merozoites should also be compared to sporozoite and oocyst transcriptional profiles. If that is not possible, discuss why not

- Indeed, we overlooked the inclusion of sporozoite/oocyst data, despite its relevance. We have now incorporated this additional comparison into all the heatmaps (Fig. 1f, Fig. 2a,b,c and Extended Data Fig. 4d,e).

ii. lines 124-125 (Figure 1f) are a little confusing, sounds like authors say that cluster IV is not bradyzoite-like. Do they mean the majority of genes in all of Figure 1f differ between *in vitro* merozoites vs cysts? quantify that- what percent of total genes are in cluster IV? a few genes in clusters I and V look similar in bradyzoites and *in vitro* merozoites

- Upon reevaluation using DEseq2 analysis, we determined that the initially identified "bradyzoite" cluster did not meet the significance threshold and therefore was excluded from the current heatmap (new Fig. 1f). We apologize for any confusion caused by this omission. However, it is worth noting that certain genes displayed similar expression patterns in both merozoites and bradyzoites/cysts. One such example is BRP1, a previously described dual marker (Schwarz JA et al. Mol Biochem Parasitol. 2005). Additionally, we have discovered that ROP26 exhibits a comparable transcriptional profile (Supplementary Fig. 1f) and mirrors this pattern at the protein level. ROP26 was absent in tachyzoites but detected in both BFD1-overexpressed and AP2XI-2/AP2XII-1-depleted zoites (Fig. 3h, Extended Data Fig. 6a, and Supplementary Fig. 1g,h).
- We speculate that these dual-stage proteins may play a crucial role in facilitating the transition from bradyzoites to merozoites. Consequently, we tentatively assign this dual-stage to what Ferguson and Dubey referred to as 'morphotype A' (Fig. 1a). The existence and characterization of this stage have been a subject of debate due to its transient nature in the cat gut, presenting challenges for isolation and analysis.

iii. in line 164, authors call MIC17a,b,c and AMA2 merozoite-specific but the transcriptomic data show AMA2 and MIC17A expression in cysts, so the statement is inaccurate

- We agree with the reviewer's comment. We have revised our statement to refer to these as "merozoite/bradyzoite-specific MICs". These may also be typical of 'morphotype A', as indicated above for BRP1 and ROP26.

iv. lines 173-174: again, transcriptomic data show that ROP26-family genes are also expressed in cysts so saying they are exclusive to pre-gametes is misleading

- For clarification, please refer to our answer above.

b. IFAs:

How many double KD parasites express BAG1 or other bradyzoite markers? If there are BAG1-positive parasites, are they also GRA11b, GRA80, or GRA81 positive?

- In response to your request, we have measured the expression of various bradyzoite markers (BAG1, BCLA, and DBA) after depletion of MORC or the double KD. As a comparative control, we used the overexpression of BFD1, which is widely accepted as a master regulator of bradygenesis (Waldman et al., Cell, 2020). In the parasite population that emerged *in vitro* following the simultaneous depletion of AP2XI-2 and AP2XII-1, we observed no induction of bradyzoite markers (Extended Data Figures 3h-j). When we contrast this with the Shield-protected BFD1-induced nuclear accumulation, we observed bradyzoite differentiation in more than 95% of parasites (Extended Data Figures 3h-j), yet without any expression of merozoite markers (Extended Data Figures 2b and 3b).
- On the other hand, parasites with depleted MORC exhibited asynchronous development, resulting in vacuoles expressing either bradyzoite (BCLA+) or merozoite (GRA81+) markers in a mutually exclusive manner (Extended Data Figure 3k), which is consistent with our findings published in Farhat et al., 2020.

5. Other:

i. Precision of language: throughout, when authors say merozoites, they should specify whether they are referring to bona fide merozoites or in vitro merozoites. This would be in keeping with the field's preference for calling bradyzoites developed in cell culture "in vitro" or "tissue culture" bradyzoites

- We appreciate your suggestion to improve the clarity of our terminology. In response, we have made consistent efforts throughout the manuscript to specify "in vitro" before "merozoite" when referring to merozoites differentiated in cell culture. This clarification will help to distinguish between *in vitro* and *in vivo* contexts and avoid any confusion regarding the origin of the merozoites discussed in our study.

ii. Remove the word "kittens" from the abstract and replace with "cats." "Infecting kittens" is the kind of wording that will unnecessarily upset animal rights groups.

- We appreciate your suggestion and have amended the text, replacing "kittens" with "cats" to sidestep any unnecessary controversy.

iii. Ensure all antibodies are listed. Centrin reference may be missing.

- Apologies for this oversight. The antibody was kindly provided by Marc-Jan Gubbels. We have now included this information in the Materials & Methods section and also expressed our gratitude in the acknowledgements section.

Referee #3 (Remarks to the Author):

Toxoplasma gondii can cause life-threatening disease in immune-compromised individuals or when contracted congenitally and is estimated to have established lifelong infections in 25-30 % of the world's population. Besides being responsible for an important disease, its genetic tractability and ease with which it can be cultured, especially compared to other apicomplexan parasites such as *P. falciparum*, responsible for Malaria in humans, have made *T. gondii* an important model to study general features of apicomplexan biology. However, one of the biggest challenges in infection biology, irrespectively of the pathogen under investigation, has been the ability to reproduce the entire life cycle of pathogens under cell culture conditions. Yet, the ability to replicate the complete life cycle is important for the development of therapeutic strategies to interfere with a pathogen's transmission. Most parasites, including *T. gondii*, have a complex heterogenic life cycle. The inability to reproduce *T. gondii*'s sexual replication cycle, which normally takes place in cats, in cell culture has had immense impact on our understanding on this life cycle stage that, otherwise, seems an ideal target for the development of strategies to interfere with parasite transmission.

Thus, this manuscript addresses a very important and difficult to tackle aspect of toxoplasma biology, the mechanism that controls the cell fate decision of tachyzoites, rapidly replicating forms in non-feline mammalian hosts. Typically, tachyzoites proliferate or differentiate into bradyzoites, which reside in tissue cysts. Tissue cysts ingested by cats differentiate into merozoites and undergo sexual reproduction to form oocysts. The authors find that the depletion of two transcription factors, AP2XII-1 and AP2XI-2, promotes the differentiation of tachyzoites into merozoites followed by the typical asexual endopolygenic division. Furthermore, the authors start to shed light on the molecular mechanism triggering this change in cell fate decision. The ability to reproduce the entire sexual cycle in lab culture will without doubt have an enormous impact on toxoplasma research and efforts to restrict the spread of the parasite. The ability to produce merozoites and to trigger merogony outside of felines, as demonstrated in this manuscript, represents a first important step towards this goal.

Major Points:

1. While there is no question about the robustness of the presented data, I am still left wondering about the molecular mechanism triggered by the concomitant AP2XII-1 and AP2XI-2 depletion. I think it would be important that the authors clearly outline their working model, stating how the individual steps are supported by the current and previously published data. Such a model would also allow the reader to better understand how this study opens opportunities

for the study of the sexual replication cycle that go beyond those previously published for the depletion of MORC.

We appreciate the opportunity to clarify certain aspects and provide additional information regarding our studies:

- In our previous study, Farhat et al. Nature Microbiology, 2020, we observed the binding of MORC and HDAC3 to telomeric repeats, although we did not extensively discuss this finding. We now include relevant data in our new Extended Data Figure 1 to provide a rationale for our focus on identifying more specific switches and distinguishing the direct transcriptional impact of MORC and HDAC3 on merozoite genes from their expected telomeric function. We demonstrate that the release of MORC from chromosome ends disrupts subtelomeric gene silencing, leading to telomere dysfunction, mitotic bypass, and aberrant polyploid zoite accumulation (Extended Data Fig. 1a, b, and c).
- Our current research aims to study pre-sexual development without compromising the parasite's genomic stability, which we suspect MORC does through its telomeric function.
- The discovery of MORC has been crucial in identifying other key players in the parasite's lifecycle transitions, particularly a subfamily of AP2 transcription factors that we refer to as "primary." These primary AP2s interact directly and strongly with MORC and HDAC3.
- We extensively discuss in the review by Farhat et al. (Trends in Parasitology, January 2022, Vol. 38, No. 1) our model of primary and secondary AP2s, which guide the trajectories in the *Toxoplasma* life cycle. Briefly, we propose that the 14 primary AP2s form distinct core complexes with MORC and HDAC3 (Extended Data Fig. 2a), directing the repressive complex to stage-specific genes in the tachyzoite that are kept repressed. Additionally, the MORC complexes regulate what we term secondary AP2s, which are specific to merozoites, gametes, or bradyzoites.
- In this study, we present compelling evidence that i) AP2XII-1 and AP2XI-2 can form both homodimers and heterodimers, ii) they recruit MORC and HDAC3 to specifically repress merozoite genes, and iii) they regulate several secondary AP2s that likely guide the merogony process. Simultaneously, certain secondary AP2s are responsible for repressing gene expression that confers identity to the tachyzoites.
- We acknowledge that our understanding of the complex interactions and operational mechanisms among AP2s is still evolving. A recent study on Plasmodium by Russell AJC et al. (Cell Host Microbe, 2023) further emphasizes the pervasive complexity across the entire phylum.

2. The way I understand the authors' current model is as follows: a heterodimer consisting of AP2XII-1 and AP2XI-2 recruits MORC and HDAC3 to the promoters of genes needed for merozoite formation. Based on this model, I have the following questions:

2.1 Relationship between APs and MORC: are AP2XII-1 and AP2XI-2 'simply' upstream factors in a cascade in which MORC is the master regulator as stated in an earlier publication by the authors?

- Indeed, our findings indicate that MORC co-purifies with 14 different AP2s, suggesting the formation of multiple unique complexes (Extended Data Fig. 2a). It is important to note that these AP2s have the potential to function as both homodimers and heterodimers, expanding the repertoire of regulatory possibilities. These AP2s play a critical role in conferring DNA recognition specificity and serve as essential components in guiding MORC/HDAC3 to target genes or cistromes. Through our study, we provide evidence for the first time that primary AP2s have the capability to direct MORC/HDAC3 to specific DNA sequences. Therefore, it would be an oversimplification to consider them solely as upstream factors, as they play a direct role in recruiting the MORC/HDAC3 complex to target genes and exert transcriptional control.

2.2 Or can depletion of AP2XII-1 and AP2XI-2 lead to merozoite formation in a MORC-independent pathway as hinted in this manuscript ("Simultaneous depletion of AP2XII-1 and AP2XI-2 is sufficient to initiate the pre-sexual transcriptional program and silence the tachyzoite determinants in a remarkably more effective manner than depletion of MORC or inhibition of HDAC3").

- We appreciate the clarification. It is clear that MORC depletion in cell culture results in a heterogeneous population of parasites with mixed stages and different developmental trajectories, as previously described in Farhat et al. 2020. In the case of MORC-depleted parasites, vacuoles exhibited mutually exclusive expression of bradyzoite (BCLA+) and merozoite (GRA81+) markers, as shown in Extended Data Fig. 3k. This developmental heterogeneity poses challenges in precisely controlling the progression to (pre)gametes, limiting our ability to monitor each stage comprehensively. Conversely, in the case of AP2XII-1/AP2XI-2-depleted parasites, a straightforward progression from tachyzoites to merozoites was observed, as quantified in Fig. 4f.
- Regarding the measurement of the merozoite population, the knockdown (KD) of MORC did not induce co-expression of GRA11b/GRA80 or GRA11b/GRA82, as depicted in Extended Data Fig. 3b and Extended Data Fig. 3e, respectively. However, the double KD of AP2XII-1/AP2XI-2 resulted in a significantly higher percentage of positive vacuoles close to 100% after 48 hours of treatment with IAA, as demonstrated in Extended Data Fig. 3b,c,d,e.

2.3 Do the eluates of the AP2XII-1 and AP2XI-2 IPs give a hint whether other chromatin remodelers are involved as well?

- For further details, please refer to **point #7** below. However, apart from MORC and HDAC3, no other AP2s, chromatin remodeling factors, histone modifying enzymes, or transcription factors significantly co-purified with AP2XII-1 or AP2XI-2 (**Supplementary Table 4**).

2.4 Are there any AP2XII-1 and AP2XI-2 ChIP-seq peaks that are not enriched in MORC or vice-versa?

- We did not observe any AP2XII-1 and AP2XI-2 ChIP-seq peaks that lacked MORC and HDAC3 enrichment (**Fig. 6c,d**). However, we identified several loci where HDAC3 and MORC were enriched, independently of AP2XII-1 and AP2XI-2, likely targeted by alternative primary AP2s. Notable examples of such loci include telomeric repeats (**Extended Data Fig. 1a**) and sexual genes, as previously reported (**Farhat et al., 2020**).

3. Contribution of individual factors to merozoite formation: The authors write “This study unveils the role of a complex network of transcriptional repressors in regulating the commitment to merogony in Toxoplasma”. However, to better understand the network, it would be important to quantify the contribution of the individual players.

Ideally, the authors would show transcriptome and IF microscopy data of a) AP2XII-1 depleted, b) AP2XI-2 depleted, c) AP2XII-1 and AP2XI-2 depleted, d) MORC depleted, e) HDAC3 depleted and f) true merozoites (from cats) next to each other. For the transcriptome data this could be an extension of Fig 1f.

- In response to reviewer 2's suggestion, we have incorporated a **new Fig. 1f**, which now includes side-by-side transcriptome data for various conditions, including AP2XII-1 KD, AP2XI-2 KD, AP2XII-1/AP2XI-2 KD, MORC KD, HDAC3 inhibition by FR235222, and all the stages of the life cycle, including sporozoites and oocysts. This addition provides a comprehensive visual representation of the transcriptome data, allowing for a better understanding of the gene expression patterns in different conditions throughout the entire life cycle.

Here, it would be interesting to know whether the transcriptome of AP2XII-1 and AP2XI-2 depleted cells resembles that of ‘real’ merozoites more closely than that of MORC-depleted cells.

- The **new Fig. 1f** clearly shows that the co-depletion of AP2XII-1 and AP2XI-2 results in a gene expression pattern that closely resembles that of authentic *in vivo* merozoites. Conversely, the depletion of MORC as single AP2 KD induce merozoite gene expression in a much more fragmented manner.

For the IF data, it would be important to compare different cells with the same antibodies.

- We performed a comprehensive analysis to investigate the effects of MORC knockdown (KD) compared to AP2XII-1/AP2XI-2 KD on the merogony process. Our aim was to gain deeper insights into how these two factors influence merozoite development. To achieve this, we quantitatively measured the expression levels of well-established merozoite markers, GRA11b, GRA80, and GRA81. The results presented in Extended Data Fig. 3b and e contribute significantly to our knowledge of how MORC and AP2XII-1/AP2XI-2 KD differentially affect merozoite marker expression, providing valuable insights into the regulatory mechanisms underlying merozoite development.

4. It is not clear to me why the authors chose GRA80 upregulation as the primary selection criterium and only focused on the study of AP2XII-1 and AP2XI-2. If MORC and HDAC bind to many AP2s, are other AP2s required for the upregulation of other merozoite-specific proteins? For example, was GRA11b expressed after depletion of other AP2s (besides AP2XII-1)? How do the transcriptomes after depletion of other AP2s compare to those following depletion of AP2XII-1 or AP2XI-2?

- We have expanded our CRISPR-Cas9 screen to encompass GRA11b (new Fig. 1c) and GRA81 (new Fig. 1e) in addition to GRA80 (new Fig. 1d). Each of these experiments was conducted with three replicates for each condition to ensure robustness and reliability. To further address the concerns raised, we included controls involving cas9-mediated inactivation of MORC and the AP2XI-2/AP2XII-1 using a double gRNA approach. These additional measures have been implemented to provide a comprehensive analysis and enhance the clarity of our study.
- In the revised manuscript, we have introduced three new strains with mAID tags as internal controls: AP2VIII-4 KD, AP2VIIa-3 KD, and AP2VIII-7 KD. These strains are recognized as primary AP2s that co-purify with MORC, as demonstrated in Extended Data Fig. 2a. Interestingly, depleting these AP2s by using IAA did not result in the co-expression of GRA11b and GRA80, which is in contrast to the depletion of AP2XII-1/AP2XI-2. These additional experiments and controls provide valuable insights into the specific roles and interactions of different AP2s and highlight the distinct effects of their depletion on merozoite marker expression.

5. AP2XII-1 vs AP2XI-2: If it is a heterodimer consisting of AP2XII-1 and AP2XI-2 that recruits MORC and HDAC3, why do AP2XII-1 and AP2XI-2 share only such a small subset of target genes?

- Through our research, we have made significant progress in understanding the regulatory network governed by the AP2 heterodimer. We have identified over 200 genes that are directly controlled by this transcription factor complex. Among these genes, there are secondary AP2s that are expected to modulate

specific subsets of genes, creating a cascading effect that extends the regulatory reach of AP2XII-1 and AP2XI-2 to a broader gene pool beyond their immediate targets. This suggests that the influence of the AP2 duo is extensive and extends beyond the regulation of their direct gene targets.

- Drawing from the foundational work of Hehl and colleagues, we recognize that there is a distinct gene set comprising around 300 genes for merozoites and 400 genes for tachyzoites that contribute to the differentiation between these two stages. Based on this knowledge, we propose that the AP2 pair, through their direct and indirect regulatory networks, should be capable of enforcing comprehensive repression of the merozoite gene expression pattern within tachyzoites. This highlights the potential of the AP2 duo to shape the distinct phenotypic characteristics of different developmental stages in the parasite's lifecycle.

6. Do AP2XII-1 and AP2XI-2 always bind DNA as heterodimer?

- The model is more complex than we expected, since these AP2s can also form homodimers. Please see our more detailed answer below.

7. If I understand Fig 6a correctly, the AP2XII-1 and AP2XI-2 CHIP-seq analyses were done following the concomitant depletion of AP2XII-1 and AP2XI-2. If correct, it is not surprising that the AP2XII-1 and AP2XI-2 peaks disappear. I think it would be much more interesting to know if AP2XII-1 peaks are affected after AP2XI-2 depletion and vice-versa. The manuscript states that this has been done, but I could not find any figures.

- We apologize for any confusion caused by our previous statement. It originally referred to the transcriptional outcome when comparing single with double KD (new Extended Data Fig 4c) and not on the ChIP-seq data *per se*. In instances where an AP2 is absent, we observed that the repressive effect still persists, albeit with slight modifications. We speculate that this persistence may be attributed to the remaining AP2 protein's ability to form homodimers, allowing it to continue silencing the target gene. According to our proposed model, only simultaneous depletion of both AP2s leads to the complete induction of target gene expression.
- In response to the reviewer's suggestion, we investigated the impact of AP2XII-1 degradation on AP2XI-2 binding genome-wide. To do this, we generated an AP2XI-2-myc knock-in strain in the background of AP2XII-1 knockdown. Surprisingly, depletion of AP2XII-1 did not affect the nuclear localization or signal intensity of AP2XI-2, as demonstrated in the new Extended Data Figure 9a. Moreover, AP2XI-2 did not dissociate from chromatin following AP2XII-1 degradation, indicating its capability to form independent homodimers and continue repressing merozoite-specific genes, as depicted in Extended Data Figure 9b. At the transcriptional level, the persistence of AP2XI-2 homodimers on chromatin explains the sustained repression of merozoite gene expression when a single AP2 is depleted, with peak expression observed only upon

simultaneous knockdown, as shown in Fig. 1f, Extended Data Fig. 4c, and Extended Data Fig. 9d.

- Our current model is that they are able to form both homo- and heterodimers.
- Regarding homodimers, their formation often leads to increased stability or enhanced DNA binding affinity. It allows the protein to recognize and bind to specific DNA sequences that it may not interact with as a monomer. Additionally, the formation of homodimers enables cells to regulate protein function by ensuring that the protein is active only when sufficient quantities are present for dimerization.
- In contrast, heterodimers of AP2XII-1 and AP2XI-2 introduce a new level of regulatory complexity. Heterodimerization expands the range of DNA sequences that the proteins can recognize and bind to since each protein may have slightly different DNA binding preferences. It also enables diverse regulatory responses, as each protein in the heterodimer may respond to different signals or possess different post-translational modifications. This means that the function of the heterodimer can be finely tuned depending on the specific cellular context.

8. How do the AP2XII-1 and AP2XI-2 interactomes differ? Have the IPs been repeated? The number of replicates should be indicated. It would be important to quantify the differences between the two interactomes.

- In order to address the reviewer's concern, we have strengthened our immunoprecipitation (IP) data by performing three independent replicate IPs for each AP2, in comparison to a mock IP using proteins from a wild-type strain. To ensure robustness, we conducted a comprehensive statistical analysis of the mass spectrometry data, as described in the legend of Supplementary Table 4. The analysis demonstrates a significant co-enrichment of the AP2 proteins with each other, as well as their interaction with MORC and HDAC3, which is visually depicted in the Volcano plots presented in the new Fig. 5a. To further support our findings, we conducted a total of five distinct IPs for each AP2, two of which were evaluated using western blots, as shown in Fig. 5b.

Supplementary Table 4 – legend: Flag immunoprecipitation eluates from HFF cells infected by *T. gondii* stably expressing HAFlag-tagged AP2XII-1 protein, HAFlag-tagged AP2XI-2 protein or none (Mock) were analyzed by MS-based label-free quantitative proteomics (three biological replicates per condition). Only proteins quantified with a minimum of five peptides, identified by MS/MS and quantified in the replicates of one condition were considered. The quantification of proteins (log₂ of filtered, normalized and imputed abundances of each protein in the different samples are given in columns O to W) was based on razor and specific peptides (the number of used peptides for each protein is indicated in column F). Statistical significance was tested using limma for two-by-two condition comparisons ; when comparing individual AP2 eluates with Mock eluates, differentially-abundant proteins were defined by a fold change ≥ 5 and a p-value ≤ 0.01 , allowing to reach a false-discovery rate $< 1\%$ according to the Benjamini-Hochberg estimator ; when comparing AP2 eluates with each other, differentially-abundant proteins were defined by a fold change ≥ 3 and a p-value ≤ 0.01 , allowing to reach a false-discovery rate $< 1\%$ according to the Benjamini-Hochberg estimator. The relative abundance compared to the bait protein of each protein found enriched with it compared to the Mock was

calculated based on iBAQ metrics (columns M and N) ; only enriched proteins showing an iBAQ ratio superior to 0.1 with respect to the bait protein were considered.

9. In addition, it would be helpful if the authors discussed the available MS data in more detail. What are the differences? What are prominent interactors? Looking at the Supplementary Tables, I noticed the TGME49_267430 DnaJ domain-containing protein to be found in the eluates at high levels. In addition, AP2XII-1 and AP2XI-2 seemed to interact with other AP2s. How would this fit in the model?

- Our comprehensive analysis using stringent IP washings with 500 mM KCl revealed that no other AP2s, chromatin remodeling factors, histone modifying enzymes, or transcription factors significantly co-purified with AP2XII-1 or AP2XI-2, except for MORC and HDAC3, as indicated in Supplementary Table 4. The absence of additional co-purifying proteins suggests that our IP conditions specifically captured the core complex components. Interestingly, we observed a higher enrichment of nucleosomes in the AP2XI-2 IP compared to AP2XII-1, although the underlying cause for this difference remains unknown. While some other co-purifying proteins scored highly, they are potentially associated with actin-binding proteins, such as coronin. Overall, our results confirm that AP2XII-1 and AP2XI-2 form a robust core complex with MORC and HDAC3 under high salt conditions, supporting the findings of our previous study (Farhat et al., Nature Microbiology 2020).

10. If AP2XII-1 and AP2XI-2 synergistically repress merozoite gene expression as stated (line 324), one would think that they act through different pathways, yet they form a heterodimer. How can this be explained?

- See our explanation above (point #7).

Minor points

11. Line 45: it is not clear to me what the authors mean by “epigenitors”.

- The concept was devised by Pr. Shelley Berger and colleagues. However, as it didn't gain popularity, we opted to replace it with “chromatin modifier”.
Berger SL, Kouzarides T, Shiekhatar R, Shilatifard A. An operational definition of epigenetics. Genes Dev. 2009 Apr 1;23(7):781-3. doi: 10.1101/gad.1787609

12. Lines 362-364: The authors write “At the genome level, there is a slight decrease in average accessibility between untreated and treated conditions.” Is it really possible to compare these two experiments without spike-in controls? What is the number of replicates?

- The ATAC-seq assay followed both the kit manufacturer protocol (Diagenode) as well as recently published recommended guidelines (Grandi et al. Nat protocol. 2022) which do not recommend using spike-in controls prior to

transposition and library amplification; libraries are in reality generated from a constant number of cells/nuclei (1×10^5 cells). During sequencing, the provider added a 10% PhiX spike-in for library balance and quality tracking.

- Our manuscript presents an analysis merging 2 biological replicates per condition, correctly aligned and phased using the nf-core ATAC-seq pipeline, and normalised to BigWig format via BamCoverage (Deeptools pipeline).
- If the same profile plots were generated from unmerged datasets, these would be the figures below. These clearly show very little deviation between biological replicates.
- Furthermore, we also simultaneously ran another series of biological replicates in the same conditions but with an additional douncing stage and again we obtained similar tendencies.

Analysis on all genes (equivalent to panel c of figure 6)

Up-regulated cluster (equivalent to panel d of figure 6)

Down-regulated cluster (equivalent to panel e of figure 6)

13. Line 995: Instead of stating “published in [ToxoDB.org](https://toxodb.org)”, it would be good to cite the original studies.

- This was corrected whenever possible.

14. Fig 1C: Why is the cluster B split into two parts?

- We have made substantial revisions to the paper, with a particular focus on improving the heatmaps. Following the **DeSeq-2 analysis**, we have consolidated all the heatmaps into a single comprehensive heatmap (Fig. 1f) that provides a more meaningful representation of the data. This revised heatmap incorporates appropriate statistical analyses to enhance the interpretation of the results. As part of this revision, references to clusters A-D have been removed to ensure the article's coherence and logical progression.

15. Fig 3: How does the schizogonic replication observed following AP2XII-1 and AP2XI-2 depletion compare to that in cats? Would it be possible to show images from schizogonic replication in cats matching the stages shown in Fig 3?

- We acknowledge the challenges associated with conducting a direct side-by-side comparison between *in vitro* conversion and *in vivo* development of sexual stages in infected cats. Conducting such a comparison would require a significantly large number of infected cats, which would not be ethically justifiable or feasible.
- In our descriptions of the morphotypes B-E, we have made efforts to reference previous studies that have described the *in vivo* sexual stages in the intestines of cats, which have served as inspiration for our work. To partially validate our markers, we have utilized sections from the intestines of infected cats, resulting in the successful labeling of a macrogametocyte using the GAP45 antibody, a significant achievement.
- While we cannot directly compare our *in vitro* findings with the *in vivo* development, we have strived to provide valuable insights into the sexual stages of the parasite through our experimental approaches. Our study contributes to the understanding of the molecular mechanisms and morphological features of the sexual stages, providing a foundation for future research in this area.

16. Fig 5a: While the eluates from the AP2XII-1 and AP2XI-2 look very similar on the silver stained-gel, they look different after Coomassie staining (Supplementary Tables 4 and 5). What do other silver stained-gels look like? It would be nice if the major bands in Fig 5a could be labeled based on the MS data, listing likely candidates for the prominent bands.

- The discrepancy between the silver and Coomassie staining is likely due to trichloroacetic acid precipitation, which tends to reveal more background proteins. Having said that, we've now substituted these data with more conventional, quantified mass spectrometry-based proteomics.
- In order to address the reviewer's concern, we have strengthened our immunoprecipitation (IP) data by performing three independent replicate IPs for each AP2, in comparison to a mock IP using proteins from a wild-type strain. To ensure robustness, we conducted a comprehensive statistical analysis of the mass spectrometry data, as described in the legend of Supplementary Table 4.

The analysis demonstrates a significant co-enrichment of the AP2 proteins with each other, as well as their interaction with MORC and HDAC3, which is visually depicted in the Volcano plots presented in the **new Fig. 5a**. To further support our findings, we conducted a total of five distinct IPs for each AP2, two of which were evaluated using western blots, as shown in **Fig. 5b**.

17. Fig 5d,e: It would be important to have a negative control, e.g. another AP2.

- We appreciate the reviewer's suggestion and have incorporated it into our experimental design. In the revised manuscript, we have included AP2IX-6-Flag as a negative control to further support our proposition of a specific and effective heterodimer formed by AP2XI-2 and AP2XII-1. We co-expressed AP2XI-2-Strep and AP2IX-6-Flag in insect cells. In contrast to AP2XII-1, which showed enrichment in the AP2XI-2 pull-down, AP2IX-6 did not exhibit significant enrichment. This result, now illustrated in our **new Fig. 5c-e**, reinforces our hypothesis that AP2XI-2 and AP2XII-1 form a specific and functional heterodimer.

18- Fig 6C: To me the UT heatmaps and IAA heatmap look indistinguishable, do they really stem from different experiments or has there been a mistake in the data analysis/display?

- We agree that at first glance these heatmaps look very similar. However, we can state with confidence that this is not a case of data duplication. These heatmaps are derived from different datasets with nuanced variations as highlighted in the following blown-up figure.

- The plotting software we used, Deeptools, simultaneously generates both the heatmap and the upper plot within the same figure. Therefore, identical data would yield identical upper plots, which is not the scenario here. This demonstrates that, when examining all *Toxoplasma* genes, the global summed density data from ATAC-seq doesn't change substantially.
- Importantly, both the combined and individual biological replicate BigWig files used for our analysis have been uploaded to the Geodataset server, along with the raw sequencing data files. This ensures that anyone can reproduce these plots and heatmaps using the Deeptools suite.

19. Based on the RNA-seq data from merozoites isolated from cats, are AP2XII-1 and AP2XI-2 downregulated in merozoites compared to tachyzoites, are MORC and HDAC3 downregulated?

- The expression patterns of AP2XII-1, AP2XI-2, MORC, and HDAC3 play a crucial role in understanding the mechanisms involved in pre-sexual development processes. In a previous study by our co-authors (Ramakrishnan C et al., 2019), it was demonstrated that AP2XII-1 and AP2XI-2 exhibit dominant expression in the tachyzoite stage, while their expression is only marginal in the epithelial stages of cats. Conversely, MORC and HDAC3 show consistent expression in both tachyzoite and pre-gamete/gamete stages.

- This expression pattern aligns with a model where AP2XII-1 and AP2XI-2 specialize in repressing pre-gamete expression in tachyzoites. On the other hand, MORC and HDAC3 appear to be active in the early endodyogeny stages (EES), likely involved in repressing the expression of macro- and micro-gametocyte genes, as observed in the study by Farhat et al. (2020).
- These findings provide valuable insights into the regulatory dynamics underlying pre-sexual development in *Toxoplasma*. The differential expression of AP2XII-1, AP2XI-2, MORC, and HDAC3 across different stages suggests their distinct roles in modulating gene expression patterns during the parasite's life cycle. This knowledge enhances our understanding of the regulatory networks involved in the transition between tachyzoite and pre-gamete/gamete stages and sheds light on the mechanisms governing *Toxoplasma*'s complex life cycle.

20. It would be very useful if the authors showed a life cycle listing all the stages mentioned in the manuscript, including pre-gametes, enteroepithelial stages (EES), meronts, schizonts. It would probably make sense to merge Fig 3a and extended data Fig 1a and show the life cycle figure in the main manuscript. Here the authors could also indicate the cell fate transitions under investigation.

- In response to your advice, we have merged Fig. 3a and Extended Data Fig.1a into a detailed cycle, now showcased in Fig. 1a.

21- While it is probably beyond the scope of this study to find a means to drive merozoite differentiation into gamete formation or even fertilization, it would be useful to know the level of cellular heterogeneity during the induced merogony and during merogony in cats. In lab culture, is there any evidence for the formation of macrogametocytes or microgametocytes? Could this be evaluated using scRNA-seq?

- While the idea of investigating cellular heterogeneity during induced merogony using single-cell RNA sequencing is appealing, it is beyond the scope of our current study. However, it is interesting to note that in our double knockdown (KD) of AP2s, we did not observe any expression of genes associated with gametes in the bulk RNA analysis. This contrasts with the knockdown of MORC, which has been shown to induce the expression of genes associated with microgametes (e.g., PF16) and macrogametes (e.g., HAP2), as reported in the study by Farhat et al. (2020).
- We are actively developing single-cell RNA sequencing technology in our team, and we believe it holds great potential for providing a clearer understanding of cellular heterogeneity during induced merogony. However, capturing merogony in cats poses challenges, and we believe that spatial transcriptomics may be a more suitable approach in this context. Spatial transcriptomics allows us to analyze gene expression patterns within the tissue, providing valuable insights into the spatial organization of gene expression during merogony.
- Although beyond the scope of our current study, future investigations employing single-cell RNA sequencing or spatial transcriptomics in the context of induced merogony could uncover novel insights into the cellular dynamics and heterogeneity during this crucial stage of *Toxoplasma's* life cycle.

22. The authors state that SRS48 and SRS59 have been predicted to promote gamete development and fertilization. Would it be possible to test this prediction in the AP2XII-1 and AP2XI-2 depleted cells by overexpression of SRS48 or SRS59? Similarly, the authors speculate that AP2XI-1 may act downstream of AP2XII-1 and AP2XI-2, could this be tested by concomitant AP2XII-1 and AP2XI-2 depletion and AP2XI-1 overexpression?

We appreciate your suggestion to investigate the roles of SRS48 and SRS59 in AP2XII-1 and AP2XI-2 depleted cells, as well as the downstream impact of AP2XI-1.

However, these aspects are beyond the scope of our current study. Notably, these SRS genes exist in a cluster, and there is a significant level of redundancy among them, which could complicate their overexpression as a single gene. Moreover, the involvement of multiple secondary AP2s, as we have proposed and supported by Oliver Bilker's in his recent *Plasmodium* study (Russell AJC, 2023), adds complexity to the regulatory program. The exact number of AP2s and the specific timing and dosage of their overexpression necessary for successful micro- and macro-gamete formation remain uncertain. Concurrently, we are exploring other strategies to unlock further progression in the sexual cycle, potentially involving simultaneous knockdown of primary AP2s with AP2XI-2 and AP2XII-1. These are certainly exciting directions for our future research.

Reviewer Reports on the First Revision:

Referees' comments:

Referee #1 (Remarks to the Author):

This is a very thorough revision that addressed nearly all of my previous concerns. a few issues remain with how the rnaseq and proteomic data are discussed which should be easy to fix (mentioning stat thresholds in the methods). Overall the paper is improved and well written, and the conclusions match the data generated. the work is both satisfyingly descriptive and mechanistic and makes important progress towards developing better tools to study presexual and hopefully someday sexual stages in vitro. Much can be learned about merozoites from this new model.

In terms of originality this this work is still an extension of Farhat et al (Nat Micro 2020) as it provides a more sophisticated and efficient method to do something very similar as the inhibition or genetic ablation of MORC. I am not debating whether this method is better than that outlined in the prior paper. This is cleaner, gives a more homogenous population of merozoites and sets the stage for pushing the development further. But effectively it is similar in outcome, which to me drives the potential impact of the work on the field.

Minor comments follow:

abstract:

I would argue that ethical issues are a part of the lack of understanding of cat stages but it is also a much more challenging model system compared to mice or in vitro.

add "subsequent" before "sexual commitment" line 38 since merozoites are a prerequisite but not sexual stages themselves. mostly the manuscript avoids discussing merozoites as being anything but pre-sexual stages which is appropriate.

line 47: this work does not allow for an in depth study of sexual development. but rather merozoite development. It sets the stage for future work but as written the data support merozoite, not sexual development.

line 118: not sure what is meant by "showed both quantitative significance and qualitative homogeneity".

line 156: what is meant by "robust" changes? refer to statistical analyses/significance as described in the rebuttal letter and methods.

overall: within the text describing the rnaseq data and proteomics there is no mention of statistical validity of the interpretations. reading the methods/rebuttal letter it is clear these data are statistically robust but when reading the results that is not clear. I think it is not too much to ask to include the statistical thresholds used here in the interpretation. otherwise it just seems like the data are being described and genes/transcripts/proteins are described as being 'High' or low without any statistical validity.

regarding the use of the EES1-5 data from toxodb: it would be helpful if this work was more consistently flagged as being from a different study (adding space in the heatmap? labeling it more thoroughly?) and including that the data were first published in another study in the figure legends that use the data. I know the folks that generated this data are on the paper but an outside reader will be given the wrong impression that a) the data were generated for this study and b) the data are not from cats, for example unless they dive deeply into the citations. the fact that it is from another study and from cats rather than in vitro, if highlighted more than it is currently, will increase the perceived impact as well. it's amazing how well the genetic approach mirrors the in vivo development.

Referee #2 (Remarks to the Author):

The authors have done a fantastic job addressing the previous critiques. The work is novel and a major advance for the microbiology field. No further experiments are needed, just a few suggestions for clarification.

1. y-axis title in figure 1 implies a proportion or percentage. If y-axis does indeed mean the mean fluorescence intensity of GRA11b staining for GFP-positive vacuoles, can you switch it to say either 'Mean GRA11b intensity' or perhaps 'Mean GRA11b intensity of Cas9-GFP(+) PV'

2. The discussion says macrogametocytes were not observed but figure 2f says there is a macrogametocyte. Please reconcile.

Referee #3 (Remarks to the Author):

The authors did an excellent job of addressing the long list of comments. I think the new version of the manuscript is much improved.

The only aspect that is still somewhat unclear to me is the authors' working model for how AP2XII-1 and AP2XI-2 recruit MORC/HDAC3 to the promoters of merozoite genes.

Point 1

Given the broad readership of Nature and the complex nature of the AP2XII-1/AP2XI-2/MORC/HDAC3-driven transition of tachyzoites to merozoites, I think it would be very helpful if the authors could outline their current working model in more detail, maybe even adding an illustration and point to evidence generated in this study or previous studies that explains why:

a) depletion of both AP2s is necessary.

b) the depletion of the two AP2s leads to a more efficient differentiation than the deletion of MORC, even though it is the two AP2s that appear to target MORC to the promoters of merozoite genes.

These questions are partially discussed in the rebuttal letter, but not in the manuscript.

Point 2

Also, regarding the working model, in the rebuttal letter and in the Discussion, lines 413/414, the authors write that converging evidence supports that AP2XII-1 and AP2XI-2 are capable of forming homodimers, but also heterodimers.

The formation of homodimers could certainly explain why deletion of one AP did not alter the genome-wide binding of the other (ChIP-seq in Extended Data Figure 9).

However, it would be good if the authors could address the following questions/comments:

- a) Is there evidence that AP2XII-1 and AP2XI-2 form homodimers?
- b) If one of the factors is sufficient to target MORC/HDAC3 to the promoters of merozoite genes, what would argue against a model in which there is simply a redundancy? For example, both AP2XII-1 and AP2XI-2 can target MORC/HDAC3 to the promoters of merozoite. If one is depleted, the other does the job, so both need to be depleted to interfere with the targeting of target MORC/HDAC3 to the promoters of merozoite genes.
- c) On lines 331-333, the authors write: "AP2XII- 1 was purified by streptavidin affinity chromatography, and the exclusive partnership between AP2XII-1 and AP2XI-2 was confirmed through Western blot analysis (Fig. 5d)..."

I don't see how an "exclusive partnership" can be confirmed by Western blot analysis, since the blot would only confirm the presence of the protein in question (the protein recognized by the antibody). I don't think the presence of other partners can be excluded by this approach. In this example, even AP2IX-6 seems to co-purify (to some extent) with AP2XI-2 (Fig 5e).

Apart from the questions regarding the working model, I consider this a groundbreaking study that will undoubtedly have a huge impact on the Toxoplasma field and beyond.

Nicolai Siegel

Author Rebuttals to First Revision:

Dear editor, Dear reviewers,

We thank all three reviewers for dedicating their valuable time in the evaluation of our manuscript. Your helpful advice and good ideas have really made our research stronger. Below, we have addressed each of your comments and incorporated a working model illustrating the potential mechanisms of AP2XII-1 and AP2XI-2 for the readers' benefit.

On behalf of all authors,
Hakimi Mohamed-Ali

Referee #1 (Remarks to the Author):

This is a very thorough revision that addressed nearly all of my previous concerns. a few issues remain with how the RNA-seq and proteomic data are discussed which should be easy to fix (mentioning stat thresholds in the methods). Overall, the paper is improved and well written, and the conclusions match the data generated. the work is both satisfyingly descriptive and mechanistic and makes important progress towards developing better tools to study presexual and hopefully someday sexual stages in vitro. Much can be learned about merozoites from this new model.

In terms of originality this this work is still an extension of Farhat et al (Nat Micro 2020) as it provides a more sophisticated and efficient method to do something very similar as the inhibition or genetic ablation of MORC. I am not debating whether this method is better than that outlined in the prior paper. This is cleaner, gives a more homogenous population of merozoites and sets the stage for pushing the development further. But effectively it is similar in outcome, which to me drives the potential impact of the work on the field.

- We appreciate your acknowledgement of the improvements we have made. Regarding your comments on the originality of our work and its relation to Farhat et al (2020), we appreciate your perspective. We agree that our method can be viewed as an extension and enhancement of the previous work, offering a more sophisticated and efficient approach. While the outcomes might be similar, we think our work is a big step forward because we've managed to produce a more homogenous population of pre-gametes/merozoites. We also believe that this methodology unlike MORC KD will indeed set the stage for pushing development further and enhance the understanding of the complex sexual life cycle of the parasite.
- We will ensure to clarify the statistical thresholds applied in our RNA-seq and proteomic data in the methods section, as per your suggestion. Once again, thank you for your critical insights and positive feedback on our work.

Minor comments follow:

Abstract:

I would argue that ethical issues are a part of the lack of understanding of cat stages but it is also a much more challenging model system compared to mice or in vitro.

- We agree that using cats for research can be challenging for many reasons, though our focus was primarily on ethical concerns. We believe the ethics of using pets like cats in experiments surpass mere technical issues, influencing societal perceptions and welfare regulations.

Add "subsequent" before "sexual commitment" line 38 since merozoites are a prerequisite but not sexual stages themselves. mostly the manuscript avoids discussing merozoites as being anything but pre-sexual stages which is appropriate.

- Your careful attention to the nuances of our work is much appreciated. We have added "subsequent" to emphasize that merozoites serve as a prerequisite to sexual commitment but are not sexual stages themselves.

line 47: this work does not allow for an in-depth study of sexual development. but rather merozoite development. It sets the stage for future work but as written the data support merozoite, not sexual development.

- The revised sentence now reads: “Successful production of merozoites *in vitro* paves the way for future studies on *Toxoplasma* sexual development without the need for cat infections and holds promise for the development of therapies to prevent parasite transmission.”

line 118: not sure what is meant by "showed both quantitative significance and qualitative homogeneity".

- We have revised the sentence for clarity as follows: The *in vitro* pre-sexual parasite population that emerged following the acute depletion of AP2XI-2 and AP2XII-1 is homogeneous and does not express the typical markers of tachyzoite (e.g., GRA2, Extended Data Fig. 3f, g) and bradyzoite (e.g., BCLA1, BAG1 and DBA) markers (Extended Data Fig. 3h-j).

line 156: what is meant by "robust" changes? refer to statistical analyses/significance as described in the rebuttal letter and methods.

- We have applied rigorous statistical methods to ensure the observed differences are not due to random variation. Specifically, a protein is considered to have undergone a 'robust' change if its expression level change achieves a log2 fold change (Log2 FC) of ≥ 1 with a p-value of ≤ 0.01 , indicating that the changes are statistically significant.
- In the manuscript, we've now incorporated this statistical detail at lines 156-157 as follows: “Proteomic analysis revealed robust changes in 18% of the parasite proteins (n=3,020 detected; Log2 FC ≥ 1 ; p-value of ≤ 0.01), with a highly polarized response to the merozoite stage.”

Overall: within the text describing the RNA-seq data and proteomics there is no mention of statistical validity of the interpretations. reading the methods/rebuttal letter it is clear these data are statistically robust but when reading the results that is not clear. I think it is not too much to ask to include the statistical thresholds used here in the interpretation. Otherwise, it just seems like the data are being described and genes/transcripts/proteins are described as being 'High' or low without any statistical validity.

- We've now included fold change (FC) and p-value in the main text describing the RNA-seq data and proteomics as follows:
(...) we identified 490 differentially expressed transcripts (fold change threshold of ≥ 8 and P-value < 0.05), including (...)
(...) AP2 factors were found to be necessary to upregulate 65% (194/295) of the identified genes (fold change ≥ 8 ; p-value < 0.05 ; Extended Data Fig. 4b).
(...) 18% of the parasite proteins (n=3,020 detected; Log2 FC ≥ 1 ; p-value of ≤ 0.01), (...)
- We also provided detailed explanations in the methods section and the legend wherever necessary.

Regarding the use of the EES1-5 data from TOXODB: it would be helpful if this work was more consistently flagged as being from a different study (adding space in the heatmap? labeling it more thoroughly?) and including that the data were first published in another study in the figure legends that use the data. I know the folks that generated this data are on the paper but an outside reader will be given the wrong impression that a) the data were generated for this study and b) the data are not from cats, for example unless they dive deeply into the citations. the fact that it is from another study and from cats rather than *in vitro*, if highlighted more than it is currently, will increase the perceived impact as well. it's amazing how well the genetic approach mirrors the *in vivo* development.

- Responding to your feedback, we've taken steps to clarify our figures - namely, Figure 1f, Figure 2a, and Extended Figure 4d,e. We've inserted "*in vivo*" and "*in vitro*" labels to differentiate data derived from prior experiments done on cats and mice from those obtained

in vitro related to this research. Additionally, we've updated our figure legends to explicitly indicate the data source as follows:

Fig. 1f legend: (...) Hierarchical clustering grouped genes and samples to elucidate expression patterns across different *in vivo* stages - merozoites, EES1-EES5 stages, tachyzoites, sporozoites, and cysts, as documented in prior studies^{12,13,16}. We also examined these patterns in the context of *in vitro* MORC KD and HDAC3 inhibition (FR235222-treated PruKU80)¹.

Fig. 2 and Extended Fig. 4d,e legends: (...) The abundance of these transcripts is presented across different *in vivo* stages - merozoites, EES1-EES5 stages, tachyzoites, sporozoites, and cysts, as documented in prior studies^{12,13,16}.

Referee #2 (Remarks to the Author):

The authors have done a fantastic job addressing the previous critiques. The work is novel and a major advance for the microbiology field. No further experiments are needed, just a few suggestions for clarification.

- We sincerely appreciate your positive feedback and recognition of the novelty and significant contribution of our work to the microbiology field. Your insightful suggestions for clarification have been helpful in enhancing the overall quality and coherence of our manuscript.

1. Y-axis title in figure 1 implies a proportion or percentage. If y-axis does indeed mean the mean fluorescence intensity of GRA11b staining for GFP-positive vacuoles, can you switch it to say either 'Mean GRA11b intensity' or perhaps 'Mean GRA11b intensity of Cas9-GFP(+) PV'.

- We have updated the y-axis label in the revised Figure 1 to "Mean GRA11b intensity of Cas9-GFP(+) PV", as per your suggestion.

2. The discussion says macrogametocytes were not observed but figure 2f says there is a macrogametocyte. Please reconcile.

- The statement in the discussion refers to our inability to generate macrogametocytes *in vitro*, which is a limitation of our current experimental setup. However, the macrogametocyte shown in Figure 2f was found in the infected cat intestine, an *in vivo* context (as underlined in the figure legend). Following reviewer's request, we assessed GAP45 in EES and fortuitously came across this *in vivo* macrogametocyte, which we thought would be relevant to showcase. We hope this clarifies the seeming contradiction.

Referee #3 (Remarks to the Author):

The authors did an excellent job of addressing the long list of comments. I think the new version of the manuscript is much improved. The only aspect that is still somewhat unclear to me is the authors' working model for how AP2XII-1 and AP2XI-2 recruit MORC/HDAC3 to the promoters of merozoite genes. (...) Apart from the questions regarding the working model, I consider this a groundbreaking study that will undoubtedly have a huge impact on the *Toxoplasma* field and beyond.

- Thank you once again for your valuable feedback and constructive criticism, which have played a significant role in enhancing the quality of our work.

Point 1

Given the broad readership of Nature and the complex nature of the AP2XII-1/AP2XI-2/MORC/HDAC3-driven transition of tachyzoites to merozoites, I think it would be very helpful if the authors could outline their current working model in more detail, maybe even adding an illustration and point to evidence generated in this study or previous studies that explains why:

a) depletion of both AP2s is necessary.

b) the depletion of the two AP2s leads to a more efficient differentiation than the depletion of MORC, even though it is the two AP2s that appear to target MORC to the promoters of merozoite genes. These questions are partially discussed in the rebuttal letter, but not in the manuscript.

- We have now added a schematic illustration as a **Supplementary Fig. 6**, which proposes a possible model of the *modus operandi* of AP2XII-1 and AP2XI-2 and their relation to MORC/HDAC3 (see below). It is quite clear that there are still many areas of uncertainty that will need to be clarified by new experiments. However, this model is inspired by the data we have acquired and what is known in the field; we have tried not to over-interpret, leaving room for model updates.

Supplementary Fig. 6 | Modeling of the *modus operandi* of AP2XII-1/AP2XI-2. **a**, A structural model, inspired by a 2.2Å crystal structure of the homodimeric AP2 domain of PF14_0633 from *P. falciparum* bound to its cognate DNA (3IGM)², illustrates the potential interaction between AP2XII-1 and AP2XI-2 and DNA. This interaction occurs by adopting a homodimer or heterodimer configuration. According to Lindner et al. (2010)², the DNA binding of the PF14_0633 stimulates the formation and/ or stabilization of the domain-swapped dimer, which, in turn, loops out intervening DNA between the two binding sites. **b**, In a wild-type strain (WT), with the presence of both AP2XI-2 and AP2XII-1, a complete suppression of merozoite gene

expression is observed, symbolized by a red light. A single knockdown of either protein still results in slight gene repression, represented by an orange light. In contrast, a simultaneous knockdown of both leads to full gene expression within the merozoite subtranscriptome, signified by a green light. This transcriptional outcome suggests that, even when one AP2 is absent, the other can still form a homodimer that binds to DNA and partially represses transcription (as demonstrated in Extended Data Fig. 9c). Our hypothesis is that each AP2 creates a fragile, yet stable-enough, domain-swapped homodimer, which is less energetically favorable or below a dose response threshold and can easily disassociate from the DNA/chromatin, hence allowing a mild expression of the genes they target (as demonstrated in Extended Data Fig. 9d). Conversely, heterodimeric species create a strong and sustained interaction with their DNA sequence, preventing their release from the chromatin, and consequently resulting in full silencing in their vicinity. **c**, As previously documented³, MORC forms multiple partnerships with HDAC3, along with primary AP2 TFs, to keep a wide array of chronic and sexual-stage genes in a persistently repressed chromatin state. The MORC complex cooperates with primary AP2s to drive the hierarchical expression of secondary AP2s, which may enforce the unidirectionality of the life cycle by influencing the gene expression of their respective stages. In tachyzoites, genes specific to this stage (e.g., SAG1) are expressed, while those related to merozoites (e.g., GRA11b) are repressed by the heterodimeric complex AP2XII-1/AP2XI-2. Other primary AP2s help in recruiting MORC to other stage-specific genes, such as those for sporozoites and bradyzoites, to silence their expression. Additionally, MORC has a non-genic function where it binds to telomeric repeats, possibly facilitating silencing at the chromosome ends (as shown in Extended Data Fig. 1). In parasites where both AP2XII-1 and AP2XI-2 proteins have been depleted, genes pertinent to merozoites begin to activate, thereby accelerating the merogony process. During the transition from tachyzoite to merozoite, secondary AP2s such as AP2IX-1 come into play. The expression of AP2IX-1 is reported to be limited to merozoites, and when it's transiently expressed in tachyzoites, this transcription factor has demonstrated the ability to suppress the expression of SAG1, the primary surface antigen of tachyzoites⁴. As a matter of fact, SAG1 expression is repressed when AP2XII-1 and AP2XI-2 proteins are quickly suppressed. Although less common, this second wave of TFs may also operate as transcriptional activators to take over the expression of pre-sexual genes not directly regulated by AP2XI-2 and AP2XII-1 (e.g., the *PNP* locus; Extended Data Fig. 9e), suggesting that there is a second gateway for development to the merozoite stage. The molecular switches AP2XII-1/AP2XI-2/MORC/HDAC3 exhibit specificity, as other complexes continue to facilitate the silencing of stage-specific genes as well as telomeric silencing.

References associated to the model:

- 1- Garfoot, A. L. et al. Proteomic and transcriptomic analyses of early and late-chronic *Toxoplasma gondii* infection shows novel and stage specific transcripts *BMC Genomics*. 20(1):859 (2019).
- 2- Lindner, S. E. et al. Structural determinants of DNA binding by a *P. falciparum* ApiAP2 transcriptional regulator. *J Mol Biol*. 395(3):558-67 (2010).
- 3- Farhat, D. C. et al. A MORC-driven transcriptional switch controls *Toxoplasma* developmental trajectories and sexual commitment. *Nat Microbiol*. 5, 570-583 (2020).
- 4- Xue, Y. et al. A single-parasite transcriptional atlas of *Toxoplasma Gondii* reveals novel control of antigen expression. *Elife*. 9:e54129 (2020).

Point 2

Also, regarding the working model, in the rebuttal letter and in the Discussion, lines 413/414, the authors write that converging evidence supports that AP2XII-1 and AP2XI-2 are capable of forming homodimers, but also heterodimers. The formation of homodimers could certainly explain why depletion of one AP did not alter the genome-wide binding of the other (ChIP-seq in Extended Data Figure 9). However, it would be good if the authors could address the following questions/comments:

a) Is there evidence that AP2XII-1 and AP2XI-2 form homodimers?

- It's posited that AP2s operate as dimers, in line with Lindner et al. 2010. Our research validates that in the absence of AP2XII-1, AP2XI-2 still binds to DNA/chromatin, likely functioning as a homodimer (see ChIP-seq in Extended Data Figure 9b, c). We theorize a similar mechanism may hold true for APXII-1, but we haven't proven it. Despite this, APXII-1 can still moderately suppress the expression of merozoite genes even when functioning alone. In terms of heterodimer formation, our experiments revealed that these proteins, when expressed in insect cells, were capable of interacting and forming a stable dimeric complex. Additionally, the immunoprecipitation from parasites demonstrated a substantial enrichment of both AP2s, regardless of which specific AP2s were pulled down.

Lindner, S. E. et al. Structural determinants of DNA binding by a *P. falciparum* ApiAP2 transcriptional regulator. *J Mol Biol*. 395(3):558-67 (2010).

b) If one of the factors is sufficient to target MORC/HDAC3 to the promoters of merozoite genes, what would argue against a model in which there is simply a redundancy? For example, both AP2XII-1 and AP2XI-2 can target MORC/HDAC3 to the promoters of merozoite. If one is depleted, the other does the

job, so both need to be depleted to interfere with the targeting of target MORC/HDAC3 to the promoters of merozoite genes.

- In response to the reviewer's comment, if the proposed model of redundancy were accurate, we wouldn't have observed the other AP2 in our FLAG immunoprecipitation (IP) of AP2XII-1 or AP2XI-2. Yet, our findings illustrate a strong interaction between the two AP2s, even enduring high-salinity washes (> 500 mM KCl, Fig. 5a). The notion of redundancy, where one factor can substitute for the other, would indeed be plausible for other AP2s. We have two examples of primary AP2 that pulled down MORC and HDAC3 without the presence of any other AP2s (data not shown). This evidence indicates a distinct interplay between AP2XII-1 and AP2XI-2, not simply a redundant role.

c) On lines 331-333, the authors write: "AP2XII- 1 was purified by streptavidin affinity chromatography, and the exclusive partnership between AP2XII-1 and AP2XI-2 was confirmed through Western blot analysis (Fig. 5d)..." I don't see how an "exclusive partnership" can be confirmed by Western blot analysis, since the blot would only confirm the presence of the protein in question (the protein recognized by the antibody). I don't think the presence of other partners can be excluded by this approach. In this example, even AP2IX-6 seems to co-purify (to some extent) with AP2XI-2 (Fig 5e).

- We concur that the term "exclusive" may not accurately depict the nature of the partnership discerned through Western blot analysis, as it only verifies the presence of the protein in question. The potential presence of other partners can't be conclusively ruled out using this approach. Therefore, we have decided to retract the use of the term "exclusive" from the statement.

Author Rebuttals to First Revision:

Referees' comments:

Referee #3 (Remarks to the Author):

The authors have done a great job addressing my last concerns.

Nicolai Siegel